# Probiotics (*Lactobacillus plantarum* HNU082) Supplementation Relieves Ulcerative Colitis by Affecting Intestinal Barrier Functions, Immunity-Related Gene Expression, Gut Microbiota, and Metabolic Pathways in Mice

Yuqing Wu,[a] Rajesh Jha,[b] Ao Li,[a] Huanwei Liu,[a] Zeng Zhang,[a] Chengcheng Zhang,[c] Qixiao Zhai,[c] Jiachao Zhang[a,d]

[a]Key Laboratory of Food Nutrition and Functional Food of Hainan Province, College of Food Science and Engineering, Hainan University, Haikou, China
[b]Department of Human Nutrition, Food and Animal Sciences, College of Tropical Agriculture and Human Resources, University of Hawaii at Manoa, Honolulu, Hawaii, USA
[c]School of Food Science and Technology, Jiangnan University, Wuxi, China
[d]One Health Institute, Hainan University, Haikou, China

Yuqing Wu and Rajesh Jha contributed equally to this work. Author order is determined according to the specific amount of workload.

**ABSTRACT** Probiotics can effectively improve ulcerative colitis (UC), but the mechanism is still unclear. Here, shotgun metagenome and transcriptome analyses were performed to explore the therapeutic effect and the mechanism of the probiotic *Lactobacillus plantarum* HNU082 (Lp082) on UC. The results showed that Lp082 treatment significantly ameliorated dextran sulfate sodium (DSS)-induced UC in mice, which was manifested as increases in body weight, water intake, food intake, and colon length and decreases in disease activity index (DAI), immune organ index, inflammatory factors, and histopathological scores after Lp082 intake. An in-depth study discovered that Lp082 could improve the intestinal mucosal barrier and relieve inflammation by cooptimizing the biological barrier, chemical barrier, mechanical barrier, and immune barrier. Specifically, Lp082 rebuilt the biological barrier by regulating the intestinal microbiome and increasing the production of short-chain fatty acids (SCFAs). Lp082 improved the chemical barrier by reducing intercellular cell adhesion molecule-1 (ICAM-1) and vascular cell adhesion molecule (VCAM) and increasing goblet cells and mucin2. Lp082 ameliorated the mechanical barrier by increasing zonula occludens-1 (ZO-1), zonula occludens-2 (ZO-2), and occludin while decreasing claudin-1 and claudin-2. Lp082 optimized the immune barrier by reducing the content of interleukin-1$\beta$ (IL-1$\beta$), IL-6, tumor necrosis factor-$\alpha$ (TNF-$\alpha$), myeloperoxidase (MPO), and interferon-$\gamma$ (IFN-$\gamma$) and increasing IL-10, transforming growth factor-$\beta$1 (TGF-$\beta$1), and TGF-$\beta$2, inhibiting the NF-$\kappa$B signaling pathway. Taken together, probiotic Lp082 can play a protective role in a DSS-induced colitis mouse model by protecting the intestinal mucosal barrier, attenuating the inflammatory response, and regulating microbial imbalance. This study provides support for the development of probiotic-based microbial products as an alternative treatment strategy for UC.

**IMPORTANCE** Many studies have focused on the therapeutic effect of probiotics on ulcerative colitis (UC), but few studies have paid attention to the mechanism of probiotics, especially the therapeutic effect. This study suggests that Lp082 has a therapeutic effect on colitis in mice. Its mechanisms of action include protecting the mucosal barrier and actively modulating the gut microbiome, modulating inflammatory pathways, and reducing neutrophil infiltration. Our study enriches the mechanism and provides a new prospect for probiotics in the treatment of colitis, helps to deepen the understanding of the intestinal mucosal barrier, and provides guidance for the future probiotic treatment of human colitis.

Address correspondence to Qixiao Zhai, zhaiqixiao@sina.com, or Jiachao Zhang, zhjch321123@163.com.

The authors declare no conflict of interest.

**KEYWORDS** *Lactobacillus plantarum* HNU082, ulcerative colitis, intestinal mucosal barrier, short-chain fatty acid, transcriptome, shotgun metagenome, cytokine, Lp082, SCFAs

Inflammatory bowel disease (IBD) is a chronic nonspecific inflammatory disease occurring in the gastrointestinal tract and includes mainly ulcerative colitis (UC) and Crohn's disease (CD) (1). The clinical manifestations of UC patients are diarrhea, blood in the stool, weight loss, and diffuse inflammation of the colonic mucosa (2). UC has become a major health problem worldwide due to its chronicity, recurrence, and high morbidity (3) and high risk of developing colorectal cancer (CRC) (4). Furthermore, due to the disadvantages of traditional surgery and drug therapy of UC, such as postoperative complications, side effects, and high cost (5), there is an urgent need to develop a new UC treatment method.

There is no consensus on the specific pathogenesis of UC, and evidence suggests that the pathogenesis of UC is multifactorial, involving genetic susceptibility, epithelial barrier defects, immune response disorders, and environmental factors (6).

Differences in gut microbiota (type and amount) between colitis patients and healthy people are thought to be one of the key factors in disease progression (7). In UC patients, the immune response is activated, the intestinal permeability is increased, the intestinal mucosal barrier structure is destroyed, the homeostasis of gut microbiota is disturbed, and the intestinal symbiotic bacteria are destroyed, thus activating a more serious immune response, leading to the recurrence of the disease (8).

Due to the shortcomings of traditional treatments, it is urgent to develop new treatments for UC, among which probiotics, as a substitute for antibiotics, have attracted much attention for regulating gut microbiota to effectively alleviate UC (9). As one of the main probiotics, *Lactobacillus plantarum* has the characteristics of regulating the balance of gut microbiota, increasing the adhesion of beneficial bacteria to the intestinal mucosa, inhibiting the adhesion of pathogenic bacteria, and inhibiting the inflammatory reaction (10). Both animal (11) and clinical trials (12) have reported that *Lactobacillus plantarum* can reduce chronic mucosal inflammation in patients with UC and relieve the occurrence of experimental colitis induced by dextran sulfate sodium (DSS). In addition, Bibiloni et al. evaluated the efficacy of *Lactobacillus* VSL3 in 20 patients with IBD and VSL3 in newly diagnosed children with IBD and found that the *Lactobacillus* strain was effective in adult patients with mild-to-moderate IBD (13). Yin et al. (14) suggested that *Lactobacillus plantarum* can restore the damaged mucosal barrier function, regulate the imbalance of intestinal microbiota, inhibit pathogenic bacteria, enhance intestinal system immunity, and have a good effect on relieving IBD symptoms and maintaining remission. However, there are few studies on the specific mechanism of action of *Lactobacillus plantarum* in UC treatment, and there is no unified argument (15).

The strain *Lactobacillus plantarum* HNU082 (Lp082) was originally isolated from a traditional fermented food fish tea of the Li people in Hainan Province, China, which has a good safety profile and tolerance to acids and bile salts (16). The results of Lp082 whole-genome sequencing showed that this bacterium has great potential for development as a probiotic in terms of physiology and function (17). In our previous study, Lp082 not only enhanced the ecological and genetic stability of the intestinal microbiota (18) but also inhibited the growth of *Fusobacterium nucleatum* and reduced the inflammatory response (19). Previous studies have also shown that Lp082 exerts a preventive effect on hyperlipidemia through the modulation of metabolism (20). In addition, ingestion of Lp082 and supplementation with prebiotics improved the stability of the intestinal microbiota and reduced the occurrence of disorders associated with disease. These results invariably demonstrate the probiotic potential of Lp082. However, the treatment effect of Lp082 on UC has not been studied.

*Lactobacillus* has been reported to have potential benefits for IBD and CRC symptoms due to its ability to promote the formation of short-chain fatty acids (SCFAs) (21). SCFAs are important metabolites of gut microbiota, and the main components in the

intestinal tract are acetate, propionate, and butyrate. Many studies have shown that SCFAs have immunomodulatory effects (22), can reduce the expression of proinflammatory factors, can reduce the inflammatory response, and play an important role in the treatment of UC (23). Studies have shown that SCFAs can act on immune cells, such as monocyte macrophages and lymphocytes, change their gene expression, affect differentiation, chemotaxis, proliferation, and apoptosis and, thus, participate in immune regulation (24). In inflammatory responses, SCFAs can reduce the expression of C5aR, thus regulating the aggregation of macrophages and neutrophils (25). In addition, SCFAs can maintain the integrity and permeability of intestinal epithelial cells, promote the secretion of mucin in goblet cells, and protect the intestinal epithelial barrier to alleviate UC (26).

Currently, salazosulfasalazine (SASP) is a commonly used medicine to treat UC (27). Sulfasalazine is hydrolyzed into 5'-aminosalicylic acid and sulfamyridine by intestinal bacteria when it enters the human intestine. The decomposed 5'-aminosalicylic acid not only has good anti-inflammatory and antibacterial effects but also can effectively suppress the outbreak of UC through immunosuppression (28). Gu et al. (29) used SASP as the positive-control group examining tilapia head sugar lipids in the treatment of colitis.

This study aimed to compare the therapeutic effects of Lp082 and SASP on DSS-induced UC and explore the specific mechanism of Lp082 and the role of SCFAs in UC by combining shotgun metagenomic and transcriptomic techniques. The results showed that Lp082 has a better therapeutic effect than SASP on DSS-induced UC. Lp082 treats UC by regulating the gut microbiota and its metabolites (SCFAs) to maintain the intestinal mucosal barrier, influence the expression of inflammatory genes and inflammatory pathways, and influence neutrophil infiltration. Ultimately, these results will help provide new clues for revealing the complex pathogenesis of UC and the therapy of *Lactobacillus plantarum* combined with SCFAs for UC.

## RESULTS

**Intake of Lp082 alleviates physiological lesions in DSS-induced colitis mice.** People with UC have a disorder of colon function, poor absorption, loss of appetite, weight loss, diarrhea, and bloody stools (8). Therefore, the lower the body weight, the lower the amount of water and food intake and the higher the disease activity index (DAI) score (the scoring criteria are shown in Table S1 in the supplemental material), indicating more severe enteritis.

From days 1 to 7, the water intake, food intake, and body weight of the DSS group, the Lp082 group, and the SASP group all showed a similar degree of gradual decrease, which may be because these three groups were all under the same DSS modeling conditions on days 0 to 7. Then, on days 8 to 15, the water intake, food intake, and body weight of the DSS group were still decreasing, but the water intake, food intake, and body weight of Lp082 and SASP groups gradually increased. However, the water and food intake of the Lp082 combined with SASP group increased significantly from day 9 ($P < 0.05$), and body weight increased significantly from day 12 ($P < 0.05$).

The DAI index of the DSS group, Lp082 group, and SASP group increased significantly ($P < 0.05$) since the third day compared with the control group. After stopping DSS gavage on day 8, the DAI index of the DSS self-healing group still increased, while the DAI indexes of the Lp082 group and SASP group gradually decreased from day 10. Additionally, the degree of decrease in the Lp082 group was greater than that in the SASP group, indicating that Lp082 had a better improvement effect on DAI index (Fig. 1b). In addition, we observed that the feces of the mice in the DSS group were blood red, but there was no blood in the feces after Lp082 and SASP ingestion (Fig. S1a). This phenomenon is consistent with the measurement results of the DAI index.

An increase in immune organ index and a decrease in colon length indicate an increase in inflammation (23). Our results showed that the immune organ index of the DSS group was significantly increased ($P < 0.05$) but was significantly decreased after

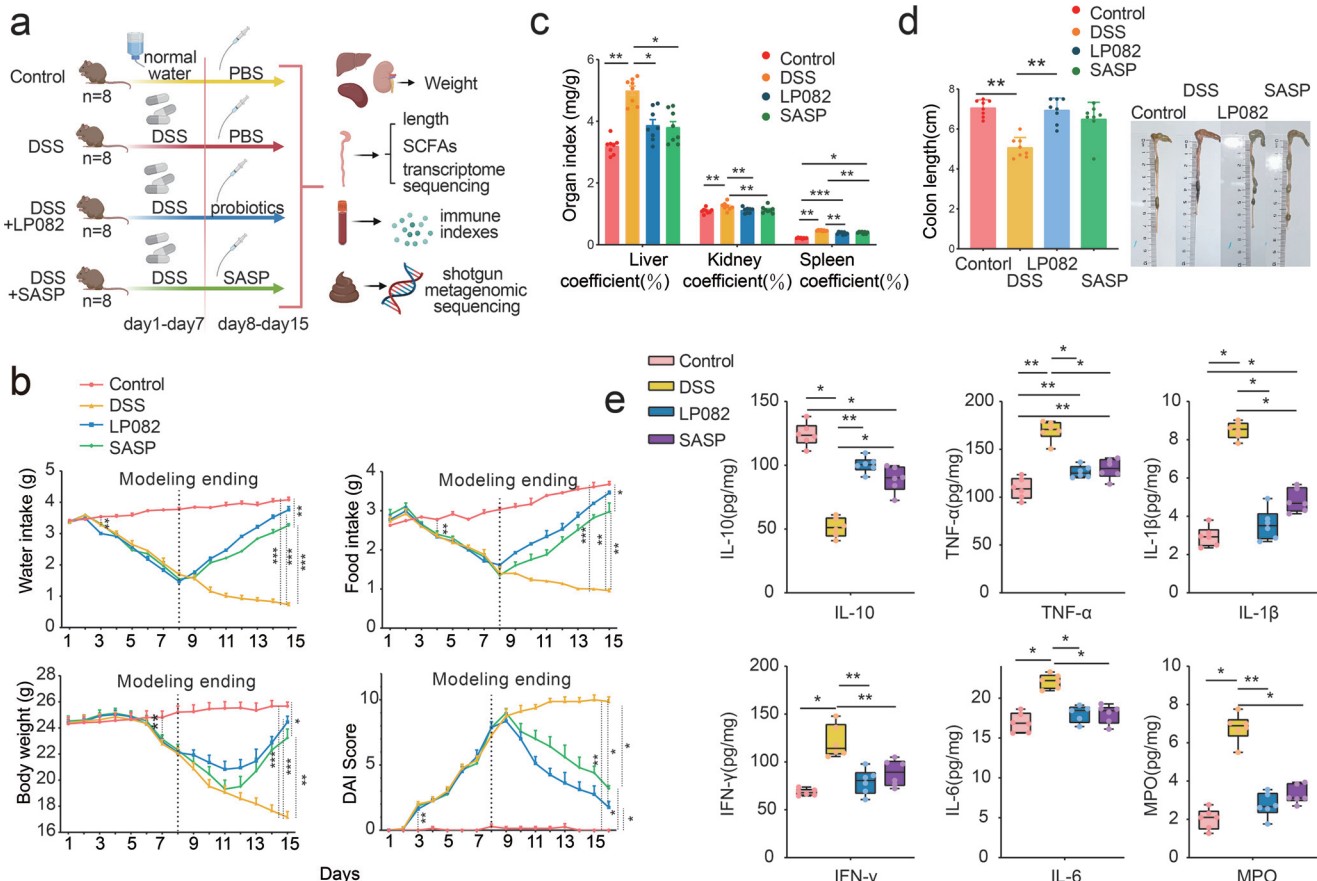

**FIG 1** Effects of Lp082 on DSS-induced UC mice. (a) Experimental design and grouping. (b) Water intake, food intake, body weight, and disease activity index ("Modeling ending" refers to the end date of modeling UC with DSS on days 1 to 7; no DSS water was administered to mice beginning with day 8). The DAI scoring system was modified from previous studies (Table S1 in the supplemental material) (70) in mice. (c) The immune organ index (mg/g) of mouse spleen, liver, and kidney. The immune organ index is calculated as immune organ index = immune organ weight (mg)/body weight (g). Increased coefficient of immune organs indicates congestion and edema of organs and increased inflammation. (d) Colon length (cm). (e) Effects of Lp082 on cytokines, including IL-10, TNF-$\alpha$, IL-1$\beta$, IFN-$\gamma$, IL-6, and MPO. Data are shown as means ± SD. A Wilcoxon signed-rank test was used. The significant difference was considered at *, $P < 0.05$; **, $P < 0.01$; and ***, $P < 0.001$. Each group had at least 6 biological replicates.

Lp082 intake ($P < 0.05$) (Fig. 1c). The colon length of the mice in the DSS group was significantly decreased ($P < 0.05$) but was significantly increased after Lp082 intake ($P < 0.05$) (Fig. 1d).

Studies have shown that DSS-induced UC mice will have a worse mental state, abdominal pain, arched back, panic, and other symptoms with the increase of disease degree, and the spleen will also increase hyperemia and infection blackening (30). After successful modeling of UC, we observed that the mice in the control group were in a normal state with normal urine and feces, shiny hair, active spirit, sensitive reaction, and increased body size. However, mice in the DSS, Lp082, and SASP groups had yellow and smelly urine, difficult defecation, bloody stool, dark and fried hair, slow reaction and easy panic, arched back, and reduced body size (Fig. S1b). On the last day of treatment (day 15), the mental state of the DSS mice was still poor, but the mental state of mice in the Lp082 and SASP groups gradually returned to normal, with an active spirit, no arched back, no hematochezia, and shiny hair (Fig. S1b). In addition, we found that the spleens of mice in the DSS group were significantly larger and darker than those of mice in the normal group, but the spleen gradually returned to normal in size and color after Lp082 and SASP intake (Fig. S1c).

The above pathological indexes in the Lp082 group were better than those in the SASP group, suggesting that Lp082 has a better remission effect on UC.

**Intake of Lp082 upregulates anti-inflammatory cytokines and downregulates proinflammatory cytokines in DSS-induced colitis mice.** To further evaluate colon injury, we quantified the proinflammatory cytokines interleukin-1beta (IL-1$\beta$),

interleukin-6 (IL-6), interferon gamma (IFN-$\gamma$), tumor necrosis factor-alpha (TNF-$\alpha$), and myeloperoxidase (MPO) and the anti-inflammatory cytokine interleukin-10 (IL-10) in the sera of 6 mice in each group. The results showed that compared with the control group, the proinflammatory cytokines TNF-$\alpha$, IL-1$\beta$, IFN-$\gamma$, IL-6, and MPO in the DSS group were significantly increased ($P < 0.05$), while the anti-inflammatory cytokine IL-10 was significantly decreased ($P < 0.05$); the opposite effect was observed in Lp082 and SASP groups (Fig. 1e).

**Intake of Lp082 alleviates pathological lesions in DSS-induced colitis mice.** In DSS-induced UC, the higher the histopathological scores, the thicker the intestinal mucosal wall, indicating more severe disease and more severe inflammation. Our results showed that ingestion of Lp082 significantly decreased the colon histopathology score and intestinal wall thickness ($P < 0.05$). The result of hematoxylin and eosin (H&E)-stained paraffin sections indicated that colonic tissue underwent severe damage in the DSS group, including neutrophil infiltration deep into the serosal layer (green arrow), crypt disappearance (black arrows), goblet cell loss (red arrow), and inflammatory cell aggregation (blue arrow) (Fig. 2a). These findings were consistent with the significantly increased colon histopathology score in the DSS group ($P < 0.05$) (Fig. 2b). However, intake of Lp082 and SASP significantly improved the above situation and increased the number of crypts (orange arrow) and goblet cells (yellow arrow), alleviated inflammatory cell foci, alleviated neutrophil infiltration, and promoted the tight junctions (TJs) of intestinal glands (gray arrow) (Fig. 2a). Moreover, a better recovery in the Lp082 group than in the SASP group was consistent with a lower histopathological score in the Lp082 group (Fig. 2b). In addition, the intestinal wall was thicker in the DSS group, thinner in the SASP group, and much thinner in the Lp082 group (Fig. 2c).

Mucin-2 (MUC-2) is the mucin secreted by goblet cells, which can form the protective layer of the intestinal mucosa epithelium (31). The tight junction protein zonula occludens-1 (ZO-1) is an important physical barrier located in the gap between intestinal epithelial cells (10). Studies have shown that the content of ZO-1 and MUC-2 is reduced in UC, and its structure and function are destroyed, resulting in increased intestinal permeability and harmful substances entering the body, aggravating inflammation. Therefore, the levels of MUC-2 and ZO-1 in the colon were determined by an immunofluorescence protein assay. The results showed that the MUC-2 protein (green fluorescence) and ZO-1 protein (red fluorescence) contents were higher in the control group, were almost absent in the DSS group, and were significantly recovered in the Lp082 and SASP groups ($P < 0.05$) and even increased more than SASP in the Lp082 group (Fig. 2d and e). These results were consistent with the surface density results of the two proteins (Fig. 2f and g). This suggests that Lp082 can reduce the decrease in the amount of ZO-1 and MUC-2 caused by DSS and maintain the normal structure and function of the intestinal mucus protein layer and intestinal epithelial cells.

**Intake of Lp082 regulates the gut microbiota in DSS-induced colitis mice.** The results of shotgun metagenomic data diversity analysis demonstrated the effect of Lp082 on the diversity of intestinal microbiota in mice. The results of the $\alpha$-diversity analysis showed that on days 1 to 7 of the study, the Shannon indexes in DSS, Lp082, and SASP groups were all significantly decreased (Fig. 3a), but the Shannon index was significantly increased after intake of Lp082 ($P < 0.05$) (Fig. 3a). The results of the $\beta$-diversity analysis showed that the DSS group, LP082 group, and SASP group (M_B, M_C, and M_D) and control group (M_A) were significantly separated on day 7 ($P < 0.05$) (Fig. 3b). However, on day 15, the DSS group was still significantly separated from the control group (T_B), while the distance between the Lp082 group (T_C), SASP group (T_D), and control group (T_A) was significantly reduced ($P$ values of <0.05), and the distance between the Lp082 group and control group was closer; the above results are consistent with the principal coordinates analysis (PCoA) distance results (Fig. 3c). The above diversity analysis results showed that Lp082 increased the $\alpha$-diversity and optimized the $\beta$-diversity of cecal microbiota in mice.

Results of the metagenomic species analysis showed that potential colitis pathogenic bacteria such as *Helicobacter hepaticus* increased significantly in the DSS group but decreased in the Lp082 group ($P < 0.05$), while the potential beneficial bacteria, such as *Lactobacillus*

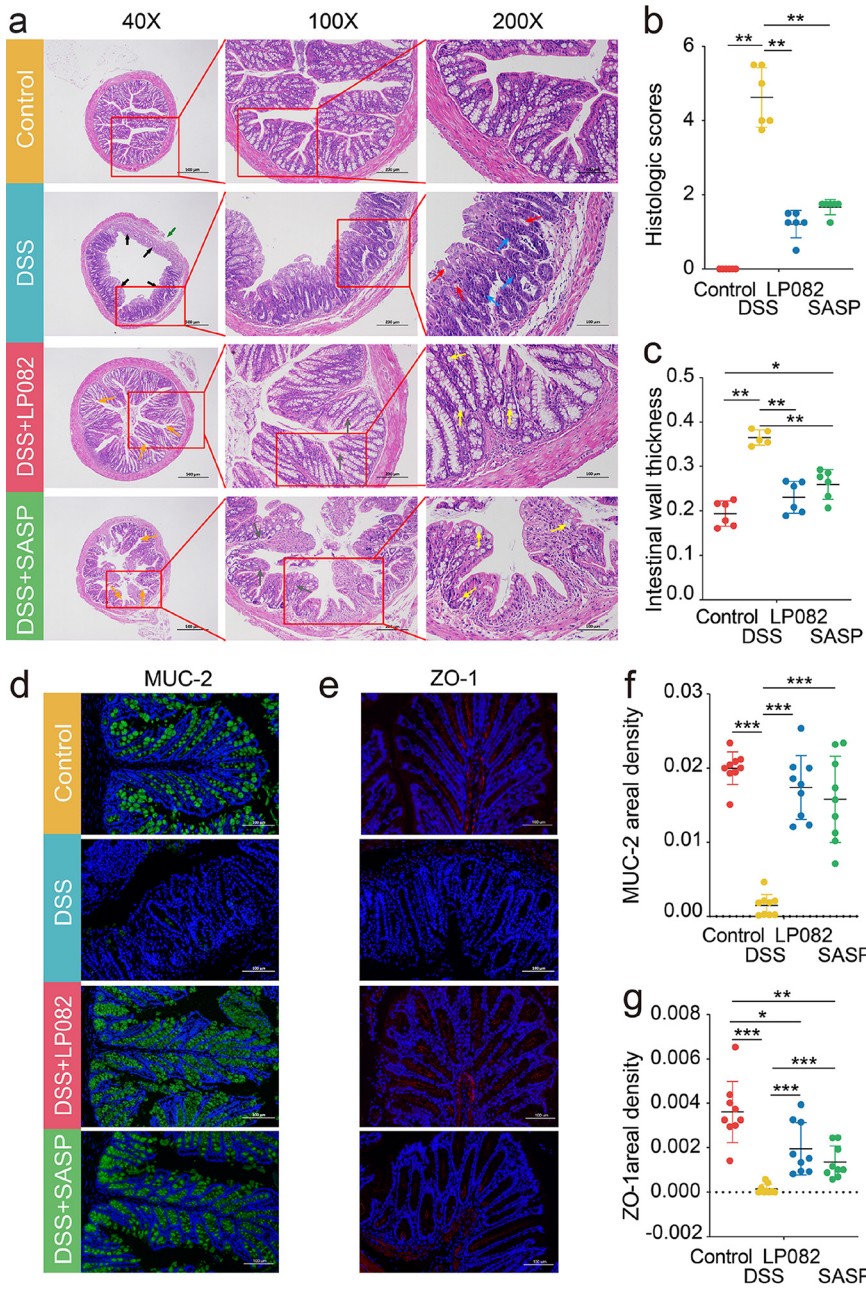

**FIG 2** Effects of Lp082 on histological parameters and immunofluorescent proteins. (a) Representative histological observation of hematoxylin & eosin-stained mouse colon at magnifications of ×40, ×100, and ×200. (b) Histopathological scoring of colon tissue. (c) Intestinal wall thickness. (d) Immunofluorescence staining of MUC-2 (green fluorescence); scale bars, 100 $\mu$m. Blue is the color of the negative of the photograph (colon tissue without antigenic markers). (e) Immunofluorescence staining of ZO-1 (red fluorescence); scale bar, 100 $\mu$m. Blue is the color of the negative of the photograph (colon tissue without antigenic markers). (f) Areal density of MUC-2 immunofluorescence protein. (g) Areal density of ZO-1 immunofluorescent protein. Data are shown as means ± SD. A Wilcoxon signed-rank test was used. The significant difference was considered at *, $P < 0.05$; **, $P < 0.01$; and ***, $P < 0.001$. Each group had at least 6 biological replicates.

plantarum, *Bifidobacterium pseudolongum*, *Akkermansia muciniphila*, *Parabacteroides distasonis*, *Lactobacillus reuteri*, and *Anaerotruncus* sp. G3 2012 (highlighted in red in Fig. 3d), were significantly decreased in the DSS group but were significantly increased in the Lp082 group ($P <$ 0.05) (Fig. 3d).

The above results show that Lp082 treatment remarkably increased the gut

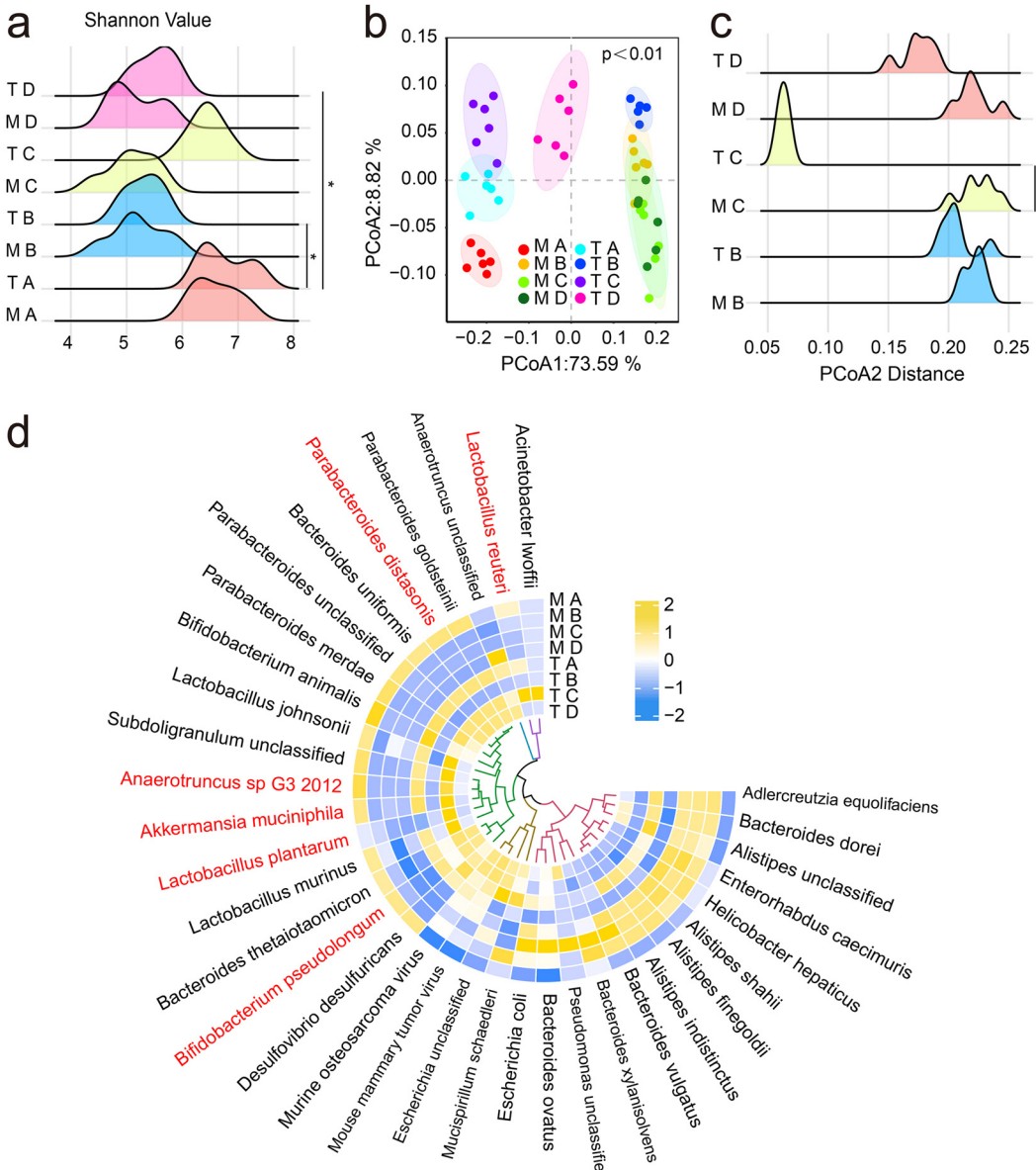

**FIG 3** Effects of Lp082 strains on the gut microbiota in mice. (a) The Shannon index. (b) Principal coordinates analysis (PCoA) based on weighted UniFrac distance at the species level. (c) PCoA distance at the species level. (d) Relative abundance of gut microbiota at the species level. The red highlight refers to the significantly increased bacteria that can produce SCFAs in the Lp082 group. The tree represents the phylogenetic tree, which is obtained by clustering the abundance of each color block based on the UniFrac distance after taking $\log_2$ ($\times$100) for the relative abundance at the species level. The clustering does not reflect any evolutionary relationship and shows the abundance of bacterial species in the sample. Zero has no special meaning in it (it is only used to facilitate the differentiation of overall abundance). The darker the yellow in the color block (the value closer to 2), the higher the relative abundance. Darker blue (values closer to −2) indicates lower relative abundance. Data are shown as means ± SD. A Wilcoxon signed-rank test was used. The significant difference was considered at *, $P < 0.05$; **, $P < 0.01$; and ***, $P < 0.001$. Each group had at least 6 biological replicates.

microbiota diversity and reduced gut microbiota structural differences, as shown by the cluster analysis and PCoA analysis; species composition was also optimized.

**Intake of Lp082 regulates short-chain fatty acids in DSS-induced colitis mice.** Next, we conducted a correlation analysis between Lp082 (*Lactobacillus plantarum*) and SCFAs and found that Lp082 (*Lactobacillus plantarum*) was strongly positively correlated with SCFAs (acetic acid, propionic acid, and butyric acid) (Fig. 4c). This correlation suggests that Lp082 can increase the content of SCFAs. The above results inspired

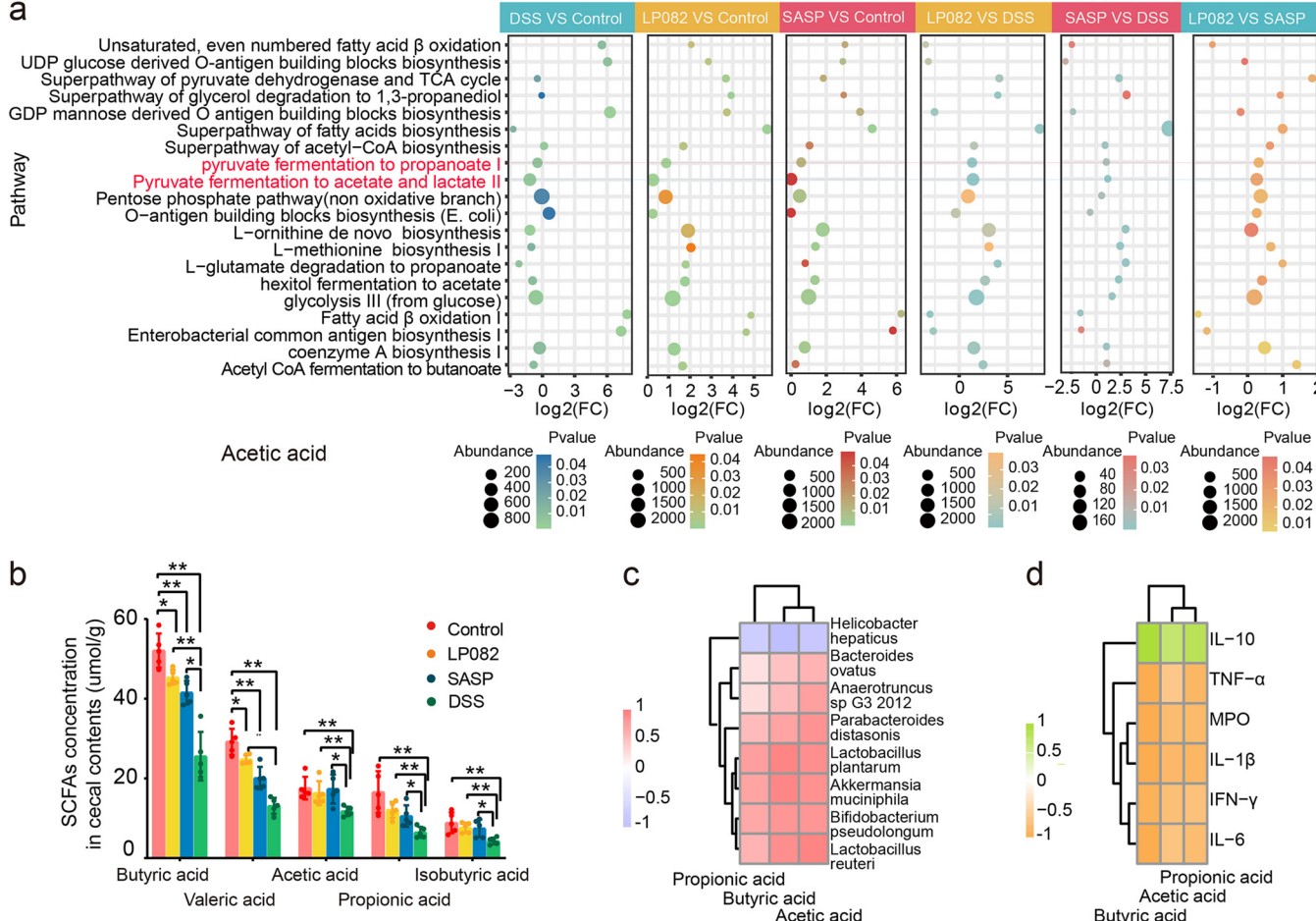

**FIG 4** The important role of SCFAs in alleviation of DSS-induced UC. (a) Gut microbial metabolic pathways associated with SCFAs. (b) SCFA content determined by gas chromatography-mass spectrometry. (c) Relationship between SCFAs and gut microbiota. The tree represents the phylogenetic tree, which was obtained by clustering the data. This clustering does not reflect any evolutionary relationships but rather shows the abundance of the samples. Shown is a correlation heat map drawn by Pearson correlation analysis based on bacterial abundance and SCFA abundance. The correlation range is from −1 to +1. The closer to −1 or +1, the stronger the correlation between bacterial species and SCFAs. Zero means no correlation, a negative value means negative correlation, and a positive value means positive correlation. (d) Relationship between SCFAs and inflammatory cytokines. The tree represents the phylogenetic tree, which was obtained by clustering the data. This clustering does not reflect any evolutionary relationships but rather shows the abundance of the samples. Shown is a correlation heat map drawn by Pearson correlation analysis based on the content of inflammatory cytokines and the abundance of SCFAs. The horizontal axis is the clustering based on the abundance of SCFAs, and the vertical axis is based on the abundance of inflammatory cytokines. Zero means no correlation, a negative value means negative correlation, and a positive value means positive correlation. Data are shown as means ± SD. A Wilcoxon signed-rank test was used. The significant difference was considered at *, $P < 0.05$; **, $P < 0.01$; and ***, $P < 0.001$. Each group had at least 6 biological replicates.

us to further explore the relationship between Lp082 and SCFAs, and we further analyzed the bacterial species and metabolic pathways associated with SCFAs. Further metagenomic data provided support for our above speculation. Combined with the metagenomic data, the species composition of the gut microbiota of the mice was further analyzed.

The results showed that the relative abundance of some special bacteria increased in the Lp082 group, such as, *Lactobacillus plantarum*, *Bifidobacterium pseudolongum*, *Akkermansia muciniphila*, *Bacteroides ovatus*, *Parabacteroides distasonis*, *Lactobacillus reuteri*, and *Anaerotruncus* sp. G3 2012 (these bacteria are highlighted in red in Fig. 3d), all of which can produce the SCFAs (32).

Subsequently, we further analyzed the metabolic pathways of the gut microbiota in mice. Results of the differential metabolic pathway analysis showed that the abundance of gut microbiota metabolic pathways related to SCFA production decreased in the DSS group but increased in the Lp082 group (Fig. 4a). We infer that Lp082 can promote the content of SCFAs (acetate, propionate, and butyrate) by adjusting three metabolic

pathways, including pyruvate fermentation to propanoate I, pyruvate fermentation to acetate and lactate II, and acetyl coenzyme A (CoA) fermentation to butanoate (Fig. 4a).

To prove the above findings, we further used gas chromatography-mass spectrometry (GC-MS) to detect the content of SCFAs in the cecal contents of 6 mice in each group. Compared with the control group, the contents of butyric acid, valeric acid, acetic acid, propionic acid, and isobutyric acid were significantly decreased after ingestion of DSS ($P < 0.01$). Compared with the DSS group, the contents of butyric acid, acetic acid, propionic acid, and isobutyric acid were extremely significantly increased after ingestion of Lp082 ($P < 0.01$). This confirmed our previous hypothesis based on the correlation that Lp082 intake would increase SCFA levels (Fig. 4b). Based on the above results, we speculate that Lp082 increases the content of SCFAs by affecting the abundance of SCFA-producing microbes as well as the metabolic pathways of SCFA-producing microbes.

To further understand the role of SCFAs, we performed a Pearson correlation analysis. The results showed that *Helicobacter hepatica*, which was significantly increased in the DSS group, was strongly negatively correlated with acetic acid, propionic acid, and butyric acid (Fig. 4c). *Lactobacillus plantarum*, *Bifidobacterium pseudolongum*, *Akkermansia muciniphila*, *Parabacteroides distasonis*, and *Lactobacillus reuteri*, which were significantly increased in the Lp082 group, showed a strong positive correlation with acetic acid, propionic acid, and butyric acid. *Anaerotruncus* sp. G3 2012 and *Bacteroides ovatus* showed a strong positive correlation with butyric acid and acetic acid and a weak positive correlation with propionic acid (Fig. 4c). These SCFAs, including acetic acid, propionic acid, and butyric acid, were all strongly negatively correlated with the proinflammatory factors TNF-$\alpha$, IL-1$\beta$, IFN-$\gamma$, IL-6, and MPO but strongly positively correlated with the inflammatory suppressor IL-10 (Fig. 4d). As important products of gut microbiota metabolism, SCFAs have certain anti-inflammatory effects and play an important role in maintaining normal intestinal morphology and function. Combined with the results of Fig. 3d and Fig. 4a to d as well as the improvement of physiological indicators (Fig. 1b to d), pathological indicators (Fig. 2a to g), and inflammatory factors (Fig. 1e) after ingestion of Lp082, we speculated that Lp082 may alleviate DSS-induced UC by regulating SCFAs through the following mechanisms (Fig. S4). That is, after the ingestion of Lp082, the abundance of SCFA-producing intestinal microbes increased, which promoted the content of SCFAs. SCFAs have the function of promoting the secretion of inflammatory cytokines and suppressing the secretion of inflammatory factors. The changes in inflammatory cytokines affect the physiological indicators of mice, which increases weight, colon length, drinking water intake, and eating volume and reduces the DAI score and immune organ index. The changes in inflammatory cytokines also affected the pathological indexes of mice, resulting in a decrease in histopathological score and an increase in immunofluorescence protein content of ZO-1 and MUC-2.

**Intake of Lp082 regulates the transcriptome of intestinal epithelial cells in DSS-induced colitis mice.** Gene distribution was analyzed using colonic transcriptome data. Results presented on a volcano map show that Lp082 significantly affected gene expression distribution (Fig. S5a to f). To further explore the impact of these differentially expressed genes (DEGs), we analyzed the pathways involved in DEGs.

Figure 5a shows the results of the Gene Ontology (GO) enrichment analysis. GO results can be divided into three categories, namely, biological processes, cellular component, and molecular function. The results of the GO analysis ($n = 6$) showed that the DEGs of the DSS group and the control group were mainly involved in biological processes such as the humoral immune response, activation of an immune response, and negative regulation of hemostasis; cellular components such as blood microparticle and membrane attack complex; and molecular functions such as lipid binding, lipopolysaccharide binding, and thrombospondin receptor activity (Fig. 5a). However, the DEGs of the Lp082 and DSS groups were mainly involved in biological processes such as blood coagulation, fibrin clot formation, regulation of humoral immune markers, and regulation of inflammatory cytokines; cellular components such as Golgi lumen and endoplasmic reticulum; and molecular functions such as endopeptidase activity and peptidase activity (Fig. 5b).

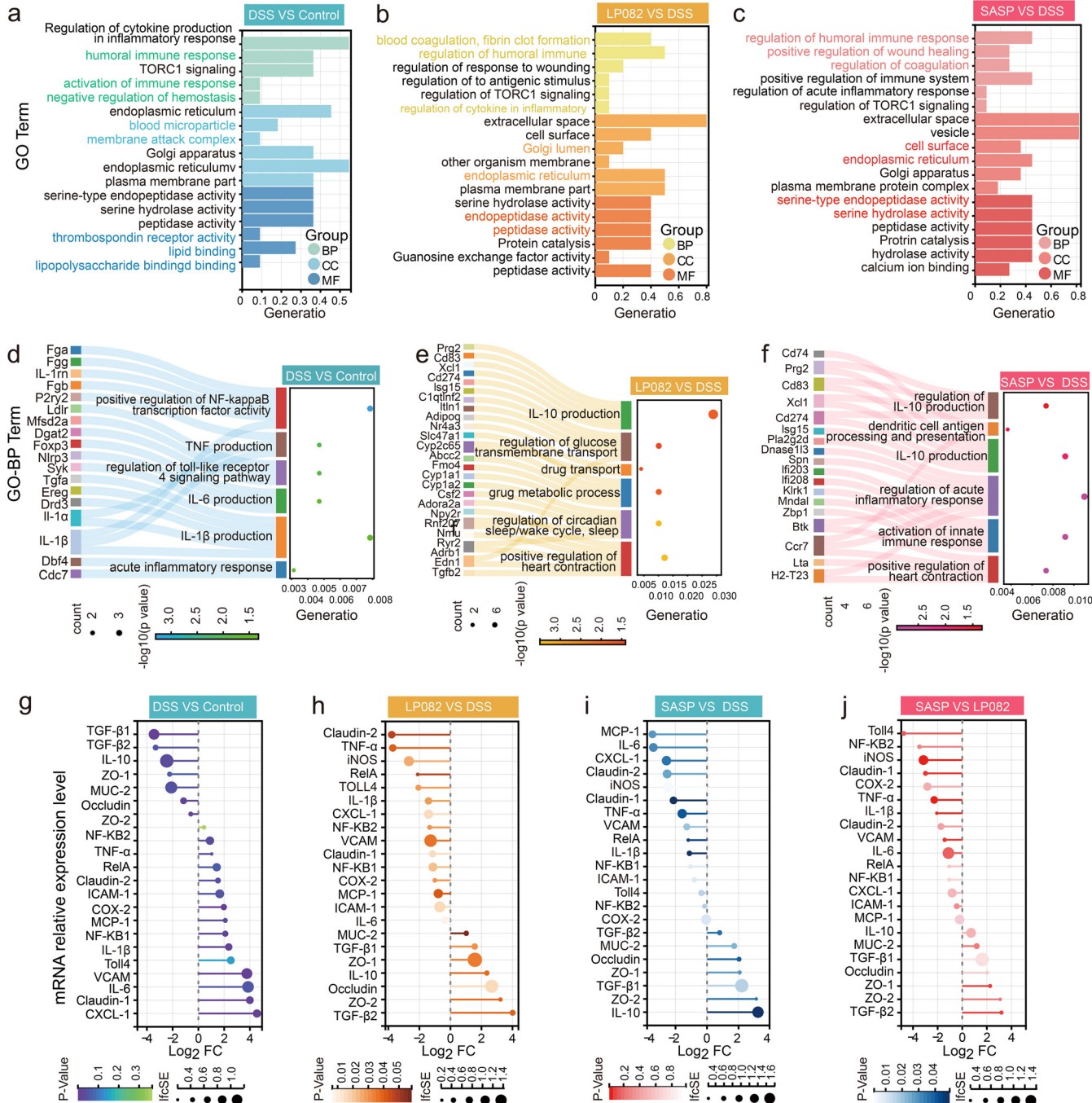

**FIG 5** Comparative study of the transcriptome of intestinal epithelial cells. (a) Gene Ontology (GO) pathway analysis of differentially expressed genes in DSS and control groups. (b) GO pathway analysis of differentially expressed genes in Lp082 and DSS groups. (c) GO pathway analysis of differentially expressed genes in SASP and DSS groups. GeneRatio indicates the ratio of the number of genes related to this term to the total number of genes. (d) GO-BP analysis of significantly upregulated differentially expressed genes in DSS and control groups. (e) GO-BP analysis of significantly upregulated differentially expressed genes in Lp082 and DSS groups. (f) GO-BP analysis of significantly upregulated differentially expressed genes in SASP and DSS groups. (g) mRNA expression of specific genes in DSS and control groups. (h) mRNA expression of specific genes in Lp082 and DSS groups. (i) mRNA expression of specific genes in SASP and DSS groups. (j) mRNA expression of specific genes in SASP and Lp082 groups. The lfcSE is the standard error, which is the value obtained from the standard deviation (SD) of the sample divided by the square root of the previous sample size. The smaller the standard error is, the smaller the difference between sample mean and population mean. A Wilcoxon signed-rank test was used. The significant difference was considered at *, $P < 0.05$; **, $P < 0.01$; and ***, $P < 0.001$. Each group had at least 6 biological replicates.

Considering that in the Lp082 group, the upregulated DEGs were far more than down-regulated DEGs (Fig. S5a to f), and the DEGs have the largest proportion of participation in biological processes (Fig. 5a to c), we further conducted a GO biological process (GO-BP) analysis ($n = 6$) on significantly upregulated DEGs. The results of the GO-BP analysis

showed that compared to the control group, upregulated DEGs in the DSS group were mainly enriched in the 6 inflammation-related GO-BPs. Among those, the genes *IL-1β* and *IL-1α* were both involved in IL-1β production and TNF production, the oncogene *Ereg* was involved in IL-1β production, the genes *IL-1β* and *IL-1rn* and oncogene *Fga* were all involved in positive regulation of nuclear factor-kappa B (NF-κB) transcription factor activity, the oncogenes *Ldlr*, *Dgat2*, and *Mfsd2a* were all involved in the regulation of the Toll-like receptor 4 signaling pathway, the prooncogenes *Cdc7* and *Dbf4* were all involved in the acute inflammatory response, and the antitumor gene *Syk* and the inflammatory genes *Nlrp3* and *Syk* were all involved in the proinflammatory factor IL-6 production (Fig. 5d). Compared to the DSS group, the upregulated genes in the Lp082 group were enriched mainly in the 6 anti-inflammatory-related GO-BPs. Among them, the gene *Isg15*, which exerts both its antiviral and anti-inflammatory effects in innate immunity, and the gene *Prg2*, which plays an important role in wound healing, were involved in the production of the anti-inflammatory factor IL-10 (Fig. 5e).

To further observe whether Lp082 treatment would suppress these inflammatory and cancer genes enriched in inflammatory pathways in the DSS group, we supplemented a picture (Fig. S6). As can be seen from Fig. S6, among the 13 inflammatory genes or oncogenes that were upregulated and enriched in the inflammatory pathway in the DSS group, the following 10 genes were significantly downregulated in the Lp082 group: *IL-1β, IL-1α, Ereg, IL-1rn, Fga, Ldlr, Dgat2, Mfsd2a, Cdc7*, and *Dbf4* (Fig. S6).

The results of KEGG analysis ($n = 6$) showed that the DEGs in the DSS and control groups were mainly enriched in systemic lupus erythematosus, *Staphylococcus aureus* infection, viral carcinogenesis, pathways in cancer, TNF signaling pathway, cellular senescence, and mitogen-activated protein kinase (MAPK) signaling pathways (Fig. S2a). However, the DEGs in both Lp082 and DSS groups, SASP and DSS groups, and SASP and Lp082 groups were mainly enriched in the following five pathways: complement and coagulation cascades, platelet activation, autophagy-animal, phagosome, and *N*-glycan biosynthesis (Fig. S2b to d). Additionally, the DEGs in the Lp082 and DSS groups and SASP and DSS groups were involved in protein processing in the endoplasmic reticulum and metabolic pathways (Fig. S2b and c).

The results of the gut mucosal barrier analysis showed that gene expression of MUC-2, ZO-1, ZO-2, and occludin was significantly reduced in the DSS group but significantly increased in the Lp082 and SASP groups ($P < 0.05$), and the gene expression of intercellular cell adhesion molecule-1 (ICAM-1), vascular cell adhesion molecule (VCAM,) claudin-1, and claudin-2 increased significantly in the DSS group but decreased significantly in the Lp082 and SASP groups ($P < 0.05$) (Fig. 5g to j). It is worth mentioning that MUC-2 is an essential component of the gut mucosa, ICAM-1 and VCAM induce gut mucosal lesions, ZO-1, ZO-2, and occludin promote tight junctions of gut epithelial cells, and claudin-1 and claudin-2 increase intestinal permeability and aggravate inflammation.

Results of the gene analysis related to the NF-κB pathway showed that Lp082 also inhibits the mRNA expression of NF-κB1, NF-κB2, cyclooxygenase-2 (COX-2), inducible nitric oxide synthase (iNOS), Toll-4, and RelA. These genes are signaling molecules in the NF-κB signaling pathway (Fig. 5g to j).

**The potential mechanism of Lp082 alleviates DSS-induced colitis.** After confirming that probiotics can help relieve UC, we explored their potential mechanisms of action. For this, correlations and interactions between Lp082 and the mouse symbiotic gut microbiome were elucidated by conducting a Pearson correlation analysis.

Lp082 was positively correlated with six strains that are positively correlated with two metabolic pathways: pyruvate fermentation to propanoate I and pyruvate fermentation to acetate and lactate II (Fig. 6a, upper right); these two metabolic pathways are positively correlated with acetic acid and propionic acid, and these two acids are negatively correlated with proinflammatory factors such as TNF-α, IL-1β, IFN-γ, IL-6, and MPO but positively correlated with the anti-inflammatory cytokine IL-10. The proinflammatory factors were negatively correlated with MUC-2, ZO-1, body weight, and

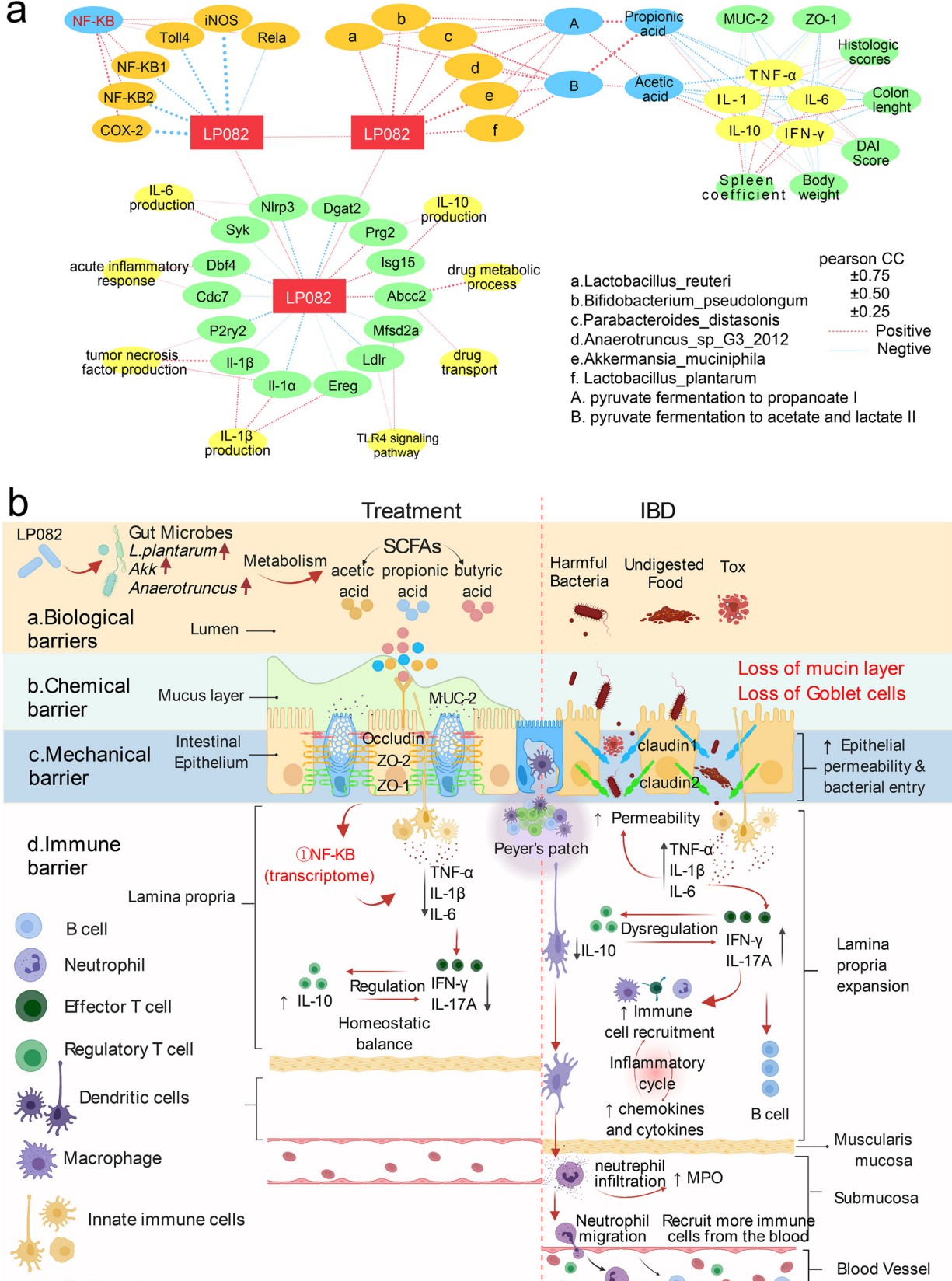

**FIG 6** The mechanism of Lp082 in relieving enteritis. (a) Correlation between Lp082 and various indicators. Red lines indicate positive correlation, blue lines indicate negative correlation, and thicker lines indicate stronger correlation. (b) Potential mechanism of action of Lp082 in relieving DSS-induced UC; Tox, toxin.

colon length while positively correlated with histologic scores, DAI score, and spleen coefficient; the opposite was true of the anti-inflammatory cytokine (Fig. 6a).

The circle in the bottom half of Fig. 6a shows that Lp082 reduces inflammation by regulating DEGs and the GO-BPs they participate in. Lp082 was negatively correlated with the genes Nlrp3, Syk, and Cdc7; Dbf4 and IL-1$\beta$; IL-1$\alpha$, P2ry2, and IL-1$\beta$; IL-1$\alpha$, Ereg, and Ldlr; and Mfsd2a, which are involved in IL-6 production, acute response, TNF production, IL-1$\beta$ production, and Toll-like receptor 4 (TLR4) signaling pathways, respectively. Lp082 was positively correlated with Abcc2, Prg2, and Isg15, which were positively correlated with drug transport, drug metabolic process, and IL-10 production, respectively (Fig. 6a).

The upper left of Fig. 6a shows that Lp082 was negatively correlated with NF-$\kappa$B2, NF-$\kappa$B1, COX-2, RelA, Toll4, and iNOS, all of which were positively correlated with NF-$\kappa$B, indicating that Lp082 inhibits the NF-$\kappa$B signaling pathway by downregulating the genes of NF-$\kappa$B signaling molecules, thus relieving inflammation (Fig. 6a).

In a further comprehensive analysis of the data, we found that Lp082 improves the intestinal mucosal barrier by optimizing the following 4 barriers. First, Lp082 improved the biological barrier by improving the gut microbiota diversity and optimizing species composition, increasing the bacteria that produce SCFAs, enhancing the metabolic pathway of SCFAs and the content of SCFAs. Second, Lp082 improved the chemical barrier by increasing the content of goblet cells and MUC-2 and reducing the ICAM-1 and VCAM content. Third, Lp082 improved the mechanical barrier by increasing the mRNA expression of ZO-1, ZO-2, and occludin and decreasing the mRNA expression of claudin-1 and claudin-2. Fourth, Lp082 improved the immune barrier by reducing the content of IL-1$\beta$, IL-6, TNF-$\alpha$, MPO, and IFN-$\gamma$ and increasing the content of IL-10, transforming growth factor-beta1 (TGF-$\beta$1), and transforming growth factor-beta2 (TGF-$\beta$2) (Fig. 6b).

## DISCUSSION

The normal intestinal mucosal barrier is composed of mechanical, chemical, immune, and biological barriers. Lp082 has good efficacy in treating UC, which motivates us to explore further its mechanism of action in the treatment of UC. The results of this study found that Lp082 can improve the intestinal mucosal barrier by synergistically optimizing the biological, chemical, mechanical, and immune barriers, thereby alleviating UC. In addition to optimizing the intestinal mucosal barrier, regulating inflammatory pathways and influencing neutrophil infiltration are potential mechanisms of Lp082 in treating UC.

**Intake of Lp082 improves the chemical barrier.** The chemical barrier refers to the glue-like mucin layer covering the surface of intestinal epithelial cells, which is mainly composed of MUC-2 secreted by goblet cells, digestive secretions, and bacteriostatic substances produced by normal parasitic bacteria in the intestinal lumen (31). The chemical barrier plays an important role in isolating the internal and external environment of the intestinal tract, lubricating the intestinal mucosa, and inhibiting the entry of harmful substances in the intestinal lumen (33). The intestinal mucosal wall thickness was significantly increased in the DSS group, whereas it was significantly decreased after Lp082 ingestion (Fig. 2c). In DSS-induced UC, the intestinal mucosal wall was thicker, indicating more severe inflammation. In addition, the H&E staining results showed that the number of goblet cells decreased in the DSS group (red arrow). In contrast, the number of goblet cells increased (yellow arrow) after Lp082 ingestion (Fig. 2a). The immunofluorescent protein content of MUC-2, which is mainly secreted by goblet cells, was significantly decreased in the DSS group (Fig. 2d), and the areal density of MUC-2 and the mRNA expression of MUC-2 were also significantly decreased in the DSS group (Fig. 2d and f), while the immunofluorescence protein content, areal density, and mRNA expression of MUC-2 all increased in the Lp082 group. Sun et al. (30) observed the same phenomenon that *Lactobacillus plantarum* 12 can repair the intestinal mucosal chemical barrier by increasing the content of MUC-2. Burgervan Paassen et al. (22) found that intake of SCFAS could increase the expression

abundance of MUC-2 mRNA in cells. Taniguchi et al. (34) found that ICAM-1 increases colonic mucosal damage. In our study, we found that Lp082 can not only decrease the mRNA expression of ICAM-1 and VCAM-1 but also relieve intestinal mucosal lesions (i.e., reduced ulceration and inflammatory cell infiltration caused by DSS). The adhesion molecules ICAM-1 and VCAM-1 are the key to the induction of intestinal mucosal lesions (35). This suggests that Lp082 may reduce intestinal mucosal lesions by reducing mRNA expression of ICAM-1 and VCAM, thereby alleviating neutrophil infiltration and ulceration. The above results showed that probiotic Lp082 increased the MUC-2 content in the mucus layer by restoring the number of goblet cells and relieved the intestinal mucosal damage caused by ICAM-1 and VCAM-1 so as to repair the chemical barrier.

**Intake of Lp082 improves the mechanical barrier.** The mechanical barrier is the most important part of the intestinal mucosal barrier. Its structural basis is the intestinal mucosal epithelial cells and the tight junctions (TJ) between the epithelial cells (36). The mechanical barrier can effectively prevent harmful substances, such as bacteria and endotoxins, from entering the blood through the intestinal mucosa. The aberrant structure of tight junction (TJ) proteins between intestinal epithelial cells, such as the reduction of ZO-1, ZO-2, and occludin, is one of the critical factors leading to the disruption of the gut mechanical barrier in UC patients (37). Several studies have identified TJ protein as a new target for the current treatment of UC (38). Cordeiro et al. (39) found that the content of ZO-1 and ZO-2 was significantly decreased in UC mice but was increased after probiotic minas frescal cheese intake. Because Lp082 excellently improved histopathology, we speculated that Lp082 also has a regulatory effect on TJ molecules. To this end, we analyzed major TJ proteins, including ZO-1, ZO-2, and occludin. As expected, the mRNA expression and immunofluorescence protein content of ZO-1 and the mRNA expression of ZO-2 and occludin were significantly decreased in DSS-induced UC mice but were significantly improved in the Lp082 group, indicating that the improvement of the mechanical barrier by regulating TJs may be one of the mechanisms by which probiotic Lp082 exerts anti-UC effects. In addition, ICAM-1 and VCAM-1, which are abnormally expressed in UC patients, were increased in the DSS group (40). Adhesion molecules ICAM-1 and VCAM-1 can not only induce intestinal mucosal injury (41) but also increase the permeability of intestinal mucosa (34), while anti-ICAM-1 treatment can alleviate colonic mucosal injury (42). Interestingly, the mRNA expression of ICAM-1 and VCAM-1 was found to decrease after Lp082 ingestion. Therefore, it can be thought that the alleviation of UC by Lp082 may be due to downregulation of ICAM-1 and VCAM-1 and increased protein quantity and mRNA expression of ZO-1 and ZO-2 to reduce intestinal mucosal permeability, thereby inhibiting the entry of harmful bacteria and undigested food and toxins into the body and reducing inflammation. These results suggest that Lp082 repairs the intestinal mechanical barrier by regulating TJ.

**Intake of Lp082 improves the immune barrier.** Although the exact etiology of UC is complex and uncertain, studies suggest that the NF-$\kappa$B pathway plays a vital role in the pathogenesis of UC (43). Our study has proven that Lp082 inhibits the NF-$\kappa$B pathway by downregulating the mRNA expression of NF-$\kappa$B2, NF-$\kappa$B1, COX-2, RelA, Toll4, and iNOS and that NF-$\kappa$B can also regulate inflammation by regulating cytokines (44). Therefore, it can be suggested that Lp082 also has a specific regulatory effect on cytokines. To confirm this, we analyzed the cytokines associated with NF-$\kappa$B. As expected, we observed that the mRNA expression level of proinflammatory cytokines (TNF-$\alpha$, IL-1$\beta$, and IL-6) was significantly increased in the DSS group but significantly decreased in the Lp082 group. It is interesting to note that the protein levels of TNF-$\alpha$, IL-1$\beta$, and IL-6 detected by enzyme-linked immunosorbent assay (ELISA) kit were also increased in the DSS group and decreased after Lp082 intake. Among them, TNF-$\alpha$ can promote the proliferation and differentiation of T cells and increase intestinal inflammation (45). The upregulation of IL-1$\beta$ is involved in the recruitment and retention of leukocytes in inflamed tissues and can activate innate immune lymphocytes (46). IL-6 activates NF-$\kappa$B to regulate dextran sulfate sodium-induced colitis in mice (47). The above results indicate that Lp082 alleviates UC by inhibiting the levels of proinflammatory factors

(TNF-$\alpha$, IL-1$\beta$, and IL-6). Interestingly, we also found that the mRNA expression of anti-inflammatory cytokines IL-10, TGF-1, and TGF-2 was significantly decreased in the DSS group but increased in the Lp082 group. IL-10 protein levels measured by ELISA also decreased in the DSS group and increased in the Lp082 group. Surprisingly, IL-10, TGF-1, and TGF-2 were shown to activate regulatory T cell (Treg) and anti-inflammatory macrophages to alleviate UC (48). Sato et al. (49) also found that the loss of IL-10 spontaneously gave rise to IBD, and Hume et al. (50) found that TGF-$\beta$1 and TGF-$\beta$2 could dramatically relieve intestinal inflammation in DSS-induced colitis mice. These results suggest that Lp082 alleviates UC by increasing the levels of anti-inflammatory factors IL-10, TGF-1, and TGF-2. We further analyzed the specific regulatory effects of Lp082 on intestinal mucosal immunity. In addition to inflammatory factors, we also noticed that a heme protein, MPO, was significantly reduced in the Lp082 group. Trevisin et al. (51) found that MPO caused UC by producing cytokines and hypochlorite and that MPO in the colon of UC patients is mainly produced by neutrophil infiltration (52). Interestingly, this is consistent with the fact that the DSS group had severe neutrophil infiltration in this study. However, neutrophil infiltration and MPO content were significantly decreased in the Lp082 group. This shows that Lp082 alleviates UC by reducing neutrophil infiltration and its secreted MPO content. Thus, our results suggest that Lp082 may have an anti-UC effect by inhibiting the NF-$\kappa$B pathway, downregulating proinflammatory cytokines, and upregulating anti-inflammatory cytokines, reducing MPO content, thereby maintaining immune balance and protecting the immune barrier.

The mucosal immune system of the intestine mainly consists of Peyer's patches and lamina propria under enterocytes (53). The Peyer's patch can deliver captured antigens to dendritic cells (54). Then, dendritic cells can not only trigger T cell-mediated cellular immunity and B cell-mediated humoral immunity by presenting antigens but also affect lamina propria immunity (55). Combining previous studies, we found that DSS causes inflammation in the following six ways. First, gut permeability increases, and harmful substances enter to activate innate immunity, such as stimulating innate immune cells to produce TNF-$\alpha$, IL-1$\beta$, and IL-6 (56). Second, regulatory T cells produce less IL-10 and have less of an inhibitory effect on effector T cells, resulting in the phenomenon of effector T and regulatory T cell dysregulation in UC patients (57). Third, effector T cells promote B cell-mediated humoral immunity by promoting the secretion of IFN-$\gamma$ and IL-17A (58). Fourth, effector T cells carry out immune cell recruitment and form a vicious immune cycle with chemokines and cytokines (59). Fifth, Peyer's patches recognize antigens and present them to other immune cells through dendritic cells (54). Sixth, antigen-activated neutrophils can both secrete MPO and recruit more immune cells from the bloodstream to the site of inflammation, further exacerbating inflammation (60) (Fig. 6b). Based on the above 6 reasons, we suggest that in addition to relieving inflammation by inhibiting the NF-$\kappa$B pathway, Lp082 can also regulate inflammatory factors to maintain the balance between regulatory T cells and effector T cells to regulate intestinal mucosal immunity, thus maintaining the intestinal mucosal barrier.

**Intake of Lp082 improves the biological barrier.** Numerous studies (51) have shown that probiotics improve the clinical outcome of IBD patients by influencing host gut microbiota (61). Herein, we performed a shotgun metagenomics analysis to investigate whether Lp082 can improve gut dysbiosis in the UC mouse model. As expected, we observed that the intake of DSS significantly reduced the Shannon value but increased PCoA distance, a finding that is consistent with Wang et al. (62). The Shannon index reflects gut microbiota richness and uniformity and is positively correlated with gut microbiota diversity, while the PCoA distance reflects the difference in the structure of the gut microbiota between different groups; the higher the PCoA value, the greater the difference in the gut microbiota structure (63). In particular, Lp082 treatment remarkably increased the gut microbiota diversity and reduced gut microbiota structural differences in gut microbiota, as shown by the cluster analysis and PCoA analysis. However, Lp082 also optimized species composition; that is, the abundance of proinflammatory microbiota decreased in

the Lp082 group, such as *Helicobacter hepaticus*, a potential pathogen of colitis. Likewise, we observed an increasing trend in the abundance of potential probiotics in the Lp082 group, such as *Bifidobacterium pseudolongum* and *Bacteroides ovatus*, which reduce colonic inflammation (64), *Parabacteroides distasonis*, which is negatively associated with obesity and diabetes (65), *Akkermansia muciniphila* and *Lactobacillus reuteri*, a widely studied probiotic, and *Anaerotruncus* sp. G3 2012 and *Lactobacillus plantarum*, potential SCFA-producing bacteria (66). The above results indicate that Lp082 is beneficial to optimizing the diversity, structure, and composition of gut microbiota. After demonstrating that Lp082 can increase the abundance of potential SCFA-producing bacteria, further analysis found that Lp082 can activate two SCFA-producing microbial metabolic pathways and SCFA content. Subsequently, correlation analysis proved that Lp082 might increase SCFAs by activating the SCFA-producing metabolic pathway of SCFA-producing bacteria to inhibit inflammation (24) and regulate host physiological activity through SCFAs (67). All of these data suggest that Lp082 repairs the microbial barrier by regulating the gut microbiome.

In conclusion, Lp082 has an exciting therapeutic effect on UC rather than SASP. Also, shotgun metagenome and transcriptome analysis confirmed that Lp082 could improve gut microbiota dysbiosis, protect the intestinal mucosal barrier, regulate inflammatory pathways, and affect neutrophil infiltration. These findings firmly support and advocate the clinical translation of Lp082 in treating UC. It can be suggested that the application of gut microbiota and probiotics in the treatment of UC should receive more attention. The findings of this study not only provide new clues for revealing the complex mechanism of gut microbiota in relieving UC but also provide evidence for Lp082 as a potential gut microbiota regulator to treat UC.

## MATERIALS AND METHODS

**Animals and their management.** A total of 32 C57BL/6J mice (male, 7 weeks old, 20 to 22 g) were purchased from Hunan Slac Jingda Laboratory Animal (Changsha, China). The mice were carefully placed in individually ventilated cages (Suzhou Fengshi Laboratory Animal Equipment Co., Ltd., Suzhou, China) under standard rearing conditions (temperature, $26 \pm 2°C$; humidity, $50\% \pm 5\%$) and maintained under a 12-h light/12-h dark cycle. Mice in all groups were fed standard normal commercial mouse chow (composed mainly of crude protein, crude fiber, crude fat, and trace elements). Mice in the control group were free to drink normal water within 15 days, and the other three groups were free to drink DSS water for the first 7 days and were changed to normal water from the 8th day. Mice were acclimatized to these conditions for 2 weeks before the experiment. After the experiment, the mice were intraperitoneally injected with 1% pentobarbital sodium solution and euthanized, and samples were collected (68). The animal experiments and all experimental protocols were approved by the Ethics Committee of Hainan University (number HNUAUCC-2021-00122) and performed according to the Guiding Principles of the Care and Use of Animals approved by the American Physiological Society.

**Animal experimental design.** C57BL/6J mice aged 7 weeks were randomly divided into 4 groups: control group ($n = 8$), dextran sulfate sodium (DSS) group ($n = 8$), *Lactobacillus plantarum* HNU082 (Lp082) group ($n = 8$), and salazosulfapyridine (SASP) group ($n = 8$). From day 1 to day 7, mice in the 3 treatment groups were given 3.5% (69) (wt/vol) DSS water (36 to 50 kDa; Coolaber Company, Beijing, China) to establish a UC model. From day 8 to day 15, mice in the DSS group, Lp082 group, and SASP group were given phosphate-buffered saline (PBS), Lp082 ($1 \times 10^9$ CFU/mL), and SASP (150 mg/kg), respectively (20) (Fig. 1a). After the UC model was established by DSS, mice were given Lp082 by gavage to observe the therapeutic effect of the bacteria on DSS-induced UC. Various tissue samples, including immune organs, serum, proximal colon, fecal, cecal contents, distal colon, and other tissues, were collected. Techniques, such as ELISA, immunohistochemistry, metagenomic sequencing, and RNA sequencing were used to assess inflammation, microbial community composition, and gene expression (Fig. 1a).

**Physiological indexes.** Physiological indexes were used to assess the health status of mice, including water intake, food intake, body weight, and disease activity index (DAI) scores modified from previous studies (70) (Table S1 in the supplemental material), and fecal occult blood measured by the *o*-toluidine method (X-Y Biotechnology, Shanghai, China). After the mice were euthanized, the colon length of 8 mice in each group was measured, the weight of the spleen, liver, and kidney of 8 mice in each group was measured, and the immune organ index was calculated according to the following formulae: spleen index = spleen weight (mg)/body weight (g); liver index = liver weight (mg)/body weight (g); and kidney index = kidney weight (mg)/body weight (g) (71).

**Inflammatory cytokines.** Blood was collected from the orbital venous plexus of mice with a capillary tube before euthanasia. Before euthanasia, 6 mice were randomly selected from each group, and blood was collected from the orbital venous plexus by a capillary tube. First, the blood sample was coagulated naturally for 30 min. Second, blood samples were centrifuged at 3,000 rpm at 4°C for 20 min to separate and collect the serum (29). Third, the levels of interleukin-1beta (IL-1$\beta$), interleukin-6 (IL-6), interleukin-10 (IL-10), interleukin-17A (IL-17A), interferon gamma (IFN-$\gamma$), tumor necrosis factor-alpha

(TNF-$\alpha$), and myeloperoxidase (MPO) in the sera of 6 randomly selected mice from each group were measured using the corresponding ELISA kits (X-Y Biotechnology, Shanghai, China), as previously described (72).

**Pathological indicators.** At the end of the experiment, we euthanized the mice, and the 1-cm portion of the distal colon of 6 mice in each group was randomly selected for H&E staining, and histopathological scores and intestinal wall thickness were further measured ($n = 6$). The following are the specific dyeing steps used: colon samples were first fixed in 4% (wt/vol) paraformaldehyde for 24 h (Servicebio Company, Wuhan, China) after washing with PBS, followed by dehydration, embedding in paraffin, and sectioning (3-$\mu$m sections). Next, some colon samples were stained with hematoxylin and eosin (H&E) as previously described (73) and were observed for histopathological changes by a light microscope (Eclipse Ci-L microscope, Nikon Corporation, Japan), as previously described (74) and modified to assess histological damage scores (Table S2). The thickness of the intestinal mucosal wall was measured by using Image-Pro Plus 6.0 analysis software to measure the thickness of the mucosal layer at 5 positions of each layer (first from the right) in a unified millimeter standard unit, and the average value was calculated.

On the other hand, 8 mice were selected from each group, and their colonic tissues were labeled with mucin 2 and ZO-1 antibodies, respectively (75), for further immunofluorescence staining (Servicebio, Wuhan, China). Fluorescein was linked to the antibodies ZO-1 and MUC-2 to form fluorescent antibodies. By specifically binding to the antigen to form a multicomponent complex, ZO-1 and MUC-2 can be characterized and localized in the intestinal tissue by means of fluorescence microscopy. The surface density of immunofluorescence ZO-1 and MUC-2 was measured and calculated by using an Eclipse CI-L fluorescence photography microscope to select the target area of tissues for 200-fold imaging. After the imaging was completed, Image-Pro Plus 6.0 analysis software was used to convert green/red fluorescent monochrome photos into black and white pictures, and then the same black was selected as the unified standard to judge the positivity of all photos. The pixel area was used as the standard unit. The positive cumulative optical density (IOD) and the corresponding tissue pixel area in each section were measured, respectively, and areal density (areal density = IOD/area) was calculated.

**Fecal DNA extraction, shotgun metagenomic sequencing, and data quality control.** Six mice were randomly selected at two time points for metagenomic sequencing of feces. At the end of modeling (day 7 of the experiment), feces of 6 mice in each group were randomly selected for metagenomic sequencing. At the end of treatment (day 15 of the experiment), feces of 6 mice in each group were randomly selected for metagenomic sequencing to observe the effects of DSS and Lp082 on the intestinal microecology of mice.

Metagenomic DNA was extracted from fecal samples following the previously reported procedure (76). Then, sequencing was performed by Beijing Novogene Co., Ltd. (Beijing, China). The raw data were processed using the sliding window method, which deletes the low-quality sequence from the original sequence, thus generating clean and high-quality data (77). Finally, for data quality control, sickle software was used for tailoring and modification of reads, and MetaPhlan3 software was used for metagenomic species annotation. Humann2 was used to analyze the diversity of the mouse gut microbiota and the composition of the microbiota at the species level (78). The Uniref90 database was used to annotate metagenomic functional characteristics, functional genes, and metabolic pathways (79).

**Determination of SCFAs in colonic contents of mice.** At the end of the experiment, the cecal contents of 6 mice from each group were randomly selected for SCFA determination.

First, 40 mg of colonic contents of the mice was lyophilized and weighed, and 600 $\mu$L of normal saline (85%) was added. The sample was then placed in a shaker for 5 min to mix thoroughly and centrifuged (8,000 rpm for 5 min). Then, 200 $\mu$L of supernatant was taken, and 100 $\mu$L of sulfuric acid (50%) was added for acidification. Four hundred microliters of $n$-hexane was added to extract SCFAs. The solution was finally passed through an organic membrane and transferred to a vial (80). Gas chromatography-mass spectrometry (GCMS; Agilent-7890, Santa Clara, CA, USA) was used to analyze the concentrations of SCFAs, including acetic acid, propionic acid, butyric acid, isobutyric acid, and valeric acid. The concentrations of SCFAs were calculated by the external standard method and expressed as $\mu$mol/g dry sample following the previously described method (81).

**RNA sequencing.** At the end of the experiment, 6 mice from each group were randomly selected for colon transcriptome RNA sequencing, and the volcanic map was drawn based on the preliminary gene distribution analysis results. Sequencing was performed by Beijing Novogene Co., Ltd. (Beijing, China). The RNA extraction minikit (Qiagen, Hilden, Germany) was used for total RNA extraction from the mouse colon samples, and a NanoDrop 2000 was used for quantification. Library construction and quality control were performed, and the raw RNA-sequencing data were filtered (82). After constructing the RNA library, an Illumina Novaseq 6000 was used for sequencing, and FeatureCounts was used to estimate gene expression (83).

**Bioinformatics and statistical analyses.** GraphPad (GraphPad Software, San Diego, CA, USA) and R software were used for statistical analysis of the data, and all data are expressed as the mean $\pm$ standard deviation (SD). Treatment groups were compared using the Wilcoxon signed-rank test (84). The values *, $P < 0.05$; **, $P < 0.01$; and ***, $P < 0.001$ were considered to indicate significant, strongly significant, and extremely significant differences, respectively.

The Ade4 package and ggplot2 package were used for principal coordinate analysis (PCoA) and to draw box charts and bubble charts, respectively. The DESeq2 package was used for $P$ value filtering and correction in the bubble diagram of the metabolic pathway (85). The correlation between the matrices was then analyzed according to the Corrplot package (86). Correlation between gut microbiota, metabolic pathways, SCFAs, inflammatory factors, ZO-1, MUC-2, pathological indicators, genes, and their

enriched pathways in mice were calculated by Pearson coefficient, and Cytoscape (Version 3.7.1) software was used for visualization (87).

DESeq2 was used for gene-centric differential expression analysis, and a $P$ of $<0.05$ and a $|\log_2$ (fold change)$|$ of $>1$ were used to screen for differentially expressed genes; volcano plots were used for differential gene visualization (88). Cluster profiler software was used to perform GO and KEGG analysis for differentially expressed genes (DEGs) (89).

**Data availability.** The sequence data reported in this paper have been deposited in the NCBI database (metagenomic sequencing data and transcriptome sequencing data, PRJNA812272).

## SUPPLEMENTAL MATERIAL

Supplemental material is available online only.
**SUPPLEMENTAL FILE 1**, PDF file, 1.8 MB.

## ACKNOWLEDGMENTS

This work was supported by the specific research fund of "The Innovation Platform for Academicians of Hainan Province (YSPTZX202121)" and The National Natural Science Foundation of China Program (number 31871773).

The study was designed by J.Z. and Q.Z. The experiments were carried out by Y.W., A.L., and H.L. Data collection was performed by Y.W. Data analysis was performed by Y.W., Z.Z., and C.Z. The manuscript was written by Y.W., R.J., and J.Z. All of the authors perused and approved the manuscript.

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
