## [Reviewer comments · Microbiology Spectrum]

Microbiology Spectrum

Probiotics (*Lactobacillus plantarum* HNU082) supplementation relieves ulcerative colitis by affecting intestinal barrier functions, immunity-related genes expression, gut microbiota, and metabolic pathways in mice

yuqing Wu, Rajesh Jha, Ao Li, Huanwei Liu, Zeng Zhang, Chengcheng Zhang, Qixiao Zhai, and Jiachao Zhang

Corresponding Author(s): Jiachao Zhang, Hainan University

Review Timeline:

Submission Date:	May 4, 2022
Editorial Decision:	June 9, 2022
Revision Received:	August 4, 2022
Editorial Decision:	October 7, 2022
Revision Received:	October 11, 2022
Accepted:	October 14, 2022

Editor: Xiaoyu Tang

Reviewer(s): The reviewers have opted to remain anonymous.

Transaction Report:

DOI: <https://doi.org/10.1128/spectrum.01651-22>

June 9, 2022

Prof. Jiachao Zhang
Hainan University
Food Science
58 renmin road
Haikou, Hainan 570228
China

Re: Spectrum01651-22 (**Probiotics (*Lactiplantibacillus plantarum* HNU082) supplementation relieves ulcerative colitis by affecting intestinal barrier functions, immunity-related genes expression, gut microbiota, and metabolic pathways in mice.**)

Dear Prof. Jiachao Zhang:

Link Not Available

Sincerely,

Xiaoyu Tang

Journals Department
Reviewer comments:

Reviewer #1 (Public repository details (Required)):

Metagenome and transcriptome raw data

Reviewer #1 (Comments for the Author):

Article summary and impression:

In the article Spectrum01651-22, the authors seek to describe the impact of supplementation of the food-derived bacterial strain *Lactiplantibacillus plantarum* HNU082 (Lp082) on the commonly used DSS-induced IBD model in C57BL/6J adult male mice with otherwise normal microbiota and diet. The authors induce inflammation with DSS supplementation in animal water, stop DSS supplementation, and then add either Lp082 or the compound SASP (although the rationale for using SASP is not provided, I assume this is a positive control for alleviation of DSS induced inflammation) to evaluate Lp082 impacts on DSS treated mice. The authors perform a number of analyses in an attempt to provide a comprehensive assessment of the impact of Lp082 treatment on DSS treated mice including the following: assessment of 1) animal behavior, 2) immune organ weight, 3) serum inflammatory cytokines, 4) colon structure and histopathology and stool formation, 5) colonic mucin and tight junction integrity, 6) microbial taxa changes and abundance, 7) SCFA acid content, 8) host epithelial transcriptional responses, as well as an attempt to connect microbiome changes to host physiology through correlation modeling. If presented accurately and completely, such a compilation is a useful addition to the scientific community and would provide a greater understanding of the impact of *Lactiplantibacillus* on colitis in healthy mouse models. However, the current version of the manuscript has a number of shortcomings, many of which are summarized below. Overall, the text and figures are confusing to follow as key information required to accurately assess the data and author conclusions has been left out. Information omission begins at the beginning of the paper and builds to where it's difficult to assess the content and accuracy of subsequent data.

Preface to the following comments:

The manuscript does not use page numbers and line numbers. To review this document, I exported the pdf to word and refer to the title page as page 1, with the first line of the title being line 1.

Major points:

1. Conditions used in figure 3A-D are inadequately described, such that I cannot sufficiently assess sample timing, sample size, comparisons made, and biological meaning. A primary contributor to this is a lack of a clear description on what M_A-M_D and T_A-T_D are and how the figures relate to sample timing. This makes it hard to assess other data in the manuscript, including overall conclusions that assess microbiome impact on the host response, which is a primary conclusion that the authors try to address.
2. Although Lp082 probiotic introduction is the primary study intervention, the authors do not mention or discuss Lp082 presence in the stool and its own genomic and metabolic contributions to the host response and the SCFA content. There is a label on Figure 3D that says "*Lactobacillus plantarum*" but it is not discussed. I'd like to see specific Lp082 evaluation and discussion in their metagenome or via another sampling method (like stool qPCR if samples still exist) that indicates the abundance of Lp082 at the times that they sampled in Figure 3 and preferably discussed in light of the experiments and data discussed in Figures 4-6.
3. The Results section "The regulatory roles of SCFAs" and Figure 4 appear to be among the weaker sections in the paper. The figures are not well described, making it difficult to understand the graphs and interpret the data (specific points made below in "minor points"). Lines 172-175 claim "the contents of acetic acid, propionic acid, butyric acid were significantly decreased in the DSS group but significantly increased in the Lp082 group ($p < 0.05$) (Fig. 4b)," but this information does not match the data in Fig. 4b. Fig. 4b shows that the cecal levels of all five evaluated SCFAs are lower than the control in DSS, Lp082, and SASP. Additionally, none of the five SCFAs are higher in Lp082 cecal contents than DSS, and in most cases, the five SCFAs appear lower in Lp082 than DSS. Thus, Fig 4b contradicts their claim that SCFAs improve host outcomes in response to Lp082 treatment after DSS. This is further reiterated by the rather small fold-change increase in the two pathways they indicate promote SCFA production in Lp082 "the fermentation of pyruvate to propionate I and the fermentation of pyruvate to acetate and lactate II" in figure 4A. The authors' conclusion that Lp082 promotes SCFA production is heavily leveraged in the discussion section, but is not well supported in their data.
4. The authors attempt to model microbial impact on the host using the bacterial metagenome and a host transcriptional analysis. This comparison would be better made if there was a microbial metatranscriptome/proteome included in this paper to support the microbial genomic data. In the absence of this, an evaluation of Lp082 itself in the host, and a weak finding on SCFA changes in response to Lp082, I find the correlations reported in figure 7 to be more speculative rather than well supported by the manuscript.
5. I'd like to see an analysis or discussion of the genes in Figure 6D for Lp082. The authors indicate that these genes in 6D are upregulated in DSS and some are pro-inflammatory. I'd like to know if Lp082 treatment suppresses these genes when compared to DSS alone.
6. Other missing information that should be addressed in the manuscript:
 - a. Rationale:
 - I. Why was SASP used?
 - II. Why was Lp082 used specifically?
 - b. Experimental design, conditions, methods:
 - I. Fig S1B does not adequately describe mouse behavior as it's a single non-descriptive image of each mouse.
 - c. Timing of experiments: After line 101, the sampling times of most experiments are omitted or inadequately described.
 - d. Sample sizes: Sample sizes and number of repeats are omitted. In most cases, the specific datapoints in figures are not well described as to what they are measuring.
 - e. Statistics:
 - a. Only one statistical test is indicated in the paper, Wilcoxin signed rank test, line 546 in the methods. Adding the test run to each figure legend would be appropriate and helpful.
 - b. Conditions statistical tests being used on are not obvious, in part due to the lack of descriptions on sample sizes and

replicates.

Minor points:

1. Missing information that should be addressed:

a. Rationale:

I. The introduction provides weak descriptions and evidence for use of a probiotic in general to treat UC and Lp082 specifically. The introduction would benefit from further elaboration on what is known about probiotic treatment of UC and indicate what is or isn't known about Lp082 usage in UC specifically rather than using general "probiotics" references. Along with this, lines 55-59 are confusing as written, but this may be addressed when more information is added about those two points.

II. The intro (starting at line 62) provides weak background on data for SCFA alleviation of IBD. Citing work and adding text of SCFA impact of IBD (preferably UC and action through immune cells) would be helpful.

b. Impact:

I. Referencing lines 94-115: No text is provided to indicate what the alterations in water intake, food intake, body weight, DAI, neurological responses, immune organ index, spleen and colon color and structure, hyperemia, and feces structure mean in the context of disease in DSS or in the Lp082 treated animals. This is not addressed elsewhere in the paper and would help the reader understand the impact of your results.

II. Lines 125-145: Text here would benefit from at least a little description on what this data means at this point in the writing. E.g. what does MUC2 loss and ZO1 abundance suggest about Lp082 effects?

c. Methods:

I. Scoring: Since understanding the scoring system used is important to understanding the data, further describing the numbering and what that means would help the reader understand the severity of the DSS model and the subsequent relief without looking up the methods reference (either in the figure legends (Figure 1B) or in the methods (see lines 480-481 where the modifications to DAI are not indicated). DAI and immune organ index should be described at some level in the results and figure legends as well so the reader knows what the data is describing without the methods.)

II. How was "surface density" quantified? Line 144, figure 2F-G.

III. Indicate the specific diet provided to the mice (line 459).

IV. Elaborate on what you mean by "mouse colon samples" on line 537 for RNA-seq.

d. Results structure:

I. The experiments, including the rationale, the samples, and the conditions, should be described at some level prior to discussing the results in the Results text so the readers know what the results are referencing.

II. Brief overall conclusions should be provided in the Results text to continue engaging the reader and leading them along your thought process. This can be partially addressed by moving text from the Discussion section to the Results. E.g. lines 302-306 can be moved to the results section where diversity is discussed.

e. Figures:

I. Figure 1:

Fig 1A - the arrows make it look like PBS only led to weight and colon assessment, probiotics to immune indices, SASP to sequencing. Collapsing the arrows would address this.

Fig 1B - what's being compared for the stats is not well described

Fig 1C - the bars for stats are shifted (also make sure the lines are the same point thickness for stats in each figure)

Fig 1D - "molding ending" is not described in the text. Rephrase or define. Also decrease the numbers in the X axis as they are too condensed. The title "duration of probiotic intervention (day)" is an incorrect title as this figure shows duration of the entire experiment, including pre-treatment with DSS before probiotics.

Fig 1E - there's no Y-axis label and the datapoints are not described

II. Figure 2:

Fig 2A - you might try to line up your red boxes better so they better represent the blow ups (and make straighter red lines).

Fig 2B - add microscopy information for the antibody stains in the legend and/or the methods section. Although the staining method cites another paper, it's best to include antibody information in the methods section. MUC2, ZO-1, and the blue marker are not labeled in the figure and in the figure legend.

Fig 2C - the y axis is missing a metric

Fig 2f-g - the y axes are missing metrics (as noted above, the method to define these numbers is not stated).

III. Figure 3:

Fig 3A-C groupings not labeled as indicated above

Fig 3D - The meaning of the red highlighting is not indicated in the figure legend. No information is provided about the tree, including what it represents and what the colors indicate. The heat map values are not described - what is being compared and what does a value of zero mean?

IV. Figure 4:

Fig. 4A - It is not entirely clear where this data comes from. My assumption was the metagenome, but the Acetic acid sub section has me unsure. Describe this figure more, taking care to describe what the acetic acid subsection is evaluating.

Fig. 4C-D - A description of the tree components is missing. Describe the correlation analysis more in the text and figure legend.

V. Figure 5: I think this entire figure would be best placed in the supplement as it's really just a sub-point of the contents of figure 6 (but it won't fit in figure 6). You might also remove "distribution" from the title and legend as this suggests tissue spatial information but is not needed.

VI. Figure 6: Overall, the less color you use, the clearer this figure will be.

Fig 6A-C: I recommend condensing as Fig 6A. Describe what gene ratio is in the figure legend.

Fig 6D-F: I recommend condensing as Fig 6B.

Fig 6G-J: I recommend condensing as Fig 6C. I and j legends are swapped. Describe ifcSE in the legend.

2. The authors confuse whether they are studying Lp082 prevention or treatment of colitis by using verbiage referring to "prevention" and "treatment" interchangeably. This makes it difficult to track what the authors are trying to accomplish (for example, line 60 says "relieving", lines 76 and 87-88 say "prevention"). Because the authors state that the colitis inducer (DSS) is administered at the time of treatment (Lp082) in the beginning of the Results to evaluate prevention (line 87), but Figure 1A shows that Lp082 is being added at day 8 (so not at the time of induction), I cannot assess which is being studied: Lp082 1) treatment or 2) prevention of UC. My best assumption is that the methods section is correct, and the methods says that DSS is used prior to addition of Lp082, and thus the authors are studying Lp082 relief of colitis. Thus, the language in the paper should be altered to indicate that Lp082 was administered after DSS induced colitis and observed effects are Lp082 alleviation of symptoms, not prevention of symptoms.
3. The abstract, discussion section, and figure 7B describe the effect of Lp082 on the animal model through the groups: biological barrier, chemical barrier, mechanical barrier, and immune barrier. I don't recommend subdividing "biological, chemical, and mechanical barrier", as everything you are referring to is biological, chemical, and mechanical in nature. Rather, use categories akin to "microbiota/microbiome alterations, barrier function improvements, and inflammation reduction."
4. In general, the abstract could be re-written to describe the results from a higher level, rather than just listing the altered genes. Close the abstract with a statement connecting the paper results to the broader scientific field.
5. As written, lines 72-73 suggest Yucha has resistance to acid and bile salts, but I assume that the authors mean Lp082 is resistant. Re-wording the sentence and adding a clarification on what point the authors are trying to make about acid and bile salt resistance would help alleviate the confusion here.
6. Referring to lines 77-92: The authors interchange physiological results with techniques as if they are the same things. Before describing the specific things you were evaluating, describe what you were looking for at a high level. Then separate physiological indicators from methods (e.g., rather than say, "evaluated physiological indexes and shotgun metagenomic sequencing," use language like "evaluated inflammation, microbial community composition and activity...using ELISA, immunohistochemistry, metagenomic sequencing, and RNA-seq."
7. Potentially incorrect information: Lines 97-98 days and scores do not line up with the data reported in figure 1B.
8. Abbreviations should be described in the text as they arise, not in an additional section at the end of the paper (page 20).
9. After revising the manuscript, a thorough and detailed assessment and correction of sentence structure would improve the readability of the paper dramatically.
10. Abbreviations, capitalization, italics, and spacing are inconsistent throughout and should be fixed for a final draft. E.g. Lp082(most commonly used in the draft)/LP082 (lines 78-79) or HNU082 (correct)/HNU082 (line 23).
11. Review your usage of "prove" in your manuscript (notably in the discussion section) as the experiments presented provide largely correlative data.

Reviewer #2 (Public repository details (Required)):

metagenomics sequencing and metabolome data are needed to deposit at a repository.

Reviewer #2 (Comments for the Author):

The manuscript aimed to demonstrate the beneficial roles and elucidate the mechanisms of Lp082 on treatment of UC. Study on specific probiotic strain is demanding, and this manuscript is timely and the knowledge obtained from this study would enrich and broaden our understanding on probiotics. However, this manuscript does need MAJOR revision before consideration for acceptance.

Major comments:

1. Authors claim that "we chose LP082 to study the mechanism of probiotics in preventing UC", however, the animal was treated with various reagents followed by DSS challenge. Please explain how this setting could serve well for assessing the effects of probiotics on prevention UC? Authors should discriminate the difference between "prevention" and "treatment", and pay more attention for accuracy of wording.
2. Basically only one biological repeat was conducted in this study. At least two biological repeats are acceptable for this purpose. Please repeat one more animal assay during next round of revision.
3. Please improve layouts of figures, and pay attention to size, location of symbols.
4. Please improve the language and grammar.
5. Please provide the H&E staining results for entire swiss roll in figure 2.
6. Authors claim that "that LP082 could improve UC by regulating gut microbiota, intestinal mucosal barrier, inflammatory pathways and neutrophil infiltration", please provide direct evidence to support Lp082 effects on "mucosal barrier". Manuscript shows the transcriptome data, however, transcriptome analysis on host genes are far away from real expression and function.

Minor comments:

1. Please provide line numbering.
2. Figure 1a depicted the study design and methodology, which might be better to merge into M&M part.

3. Information of study design and methodology are not appropriate present in Results section. The tables or figures should be displayed at a consecutive and sequential order. In current version figure S1b appeared ahead of S1a.

Staff Comments:

Preparing Revision Guidelines

Please return the manuscript within 60 days; if you cannot complete the modification within this time period, please contact me. If you do not wish to modify the manuscript and prefer to submit it to another journal, please notify me of your decision immediately so that the manuscript may be formally withdrawn from consideration by Microbiology Spectrum.

Article summary and impression:

In the article Spectrum01651-22, the authors seek to describe the impact of supplementation of the food-derived bacterial strain *Lactiplantibacillus plantarum* HNU082 (Lp082) on the commonly used DSS-induced IBD model in C57BL/6J adult male mice with otherwise normal microbiota and diet. The authors induce inflammation with DSS supplementation in animal water, stop DSS supplementation, and then add either Lp082 or the compound SASP (although the rationale for using SASP is not provided, I assume this is a positive control for alleviation of DSS induced inflammation) to evaluate Lp082 impacts on DSS treated mice. The authors perform a number of analyses in an attempt to provide a comprehensive assessment of the impact of Lp082 treatment on DSS treated mice including the following: assessment of 1) animal behavior, 2) immune organ weight, 3) serum inflammatory cytokines, 4) colon structure and histopathology and stool formation, 5) colonic mucin and tight junction integrity, 6) microbial taxa changes and abundance, 7) SCFA acid content, 8) host epithelial transcriptional responses, as well as an attempt to connect microbiome changes to host physiology through correlation modeling. If presented accurately and completely, such a compilation is a useful addition to the scientific community and would provide a greater understanding of the impact of *Lactiplantibacillus* on colitis in healthy mouse models. However, the current version of the manuscript has a number of shortcomings, many of which are summarized below. Overall, the text and figures are confusing to follow as key information required to accurately assess the data and author conclusions has been left out. Information omission begins at the beginning of the paper and builds to where it's difficult to assess the content and accuracy of subsequent data.

Preface to the following comments:

The manuscript does not use page numbers and line numbers. To review this document, I exported the pdf to word and refer to the title page as page 1, with the first line of the title being line 1.

Major points:

1. Conditions used in figure 3A-D are inadequately described, such that I cannot sufficiently assess sample timing, sample size, comparisons made, and biological meaning. A primary contributor to this is a lack of a clear description on what M_A-M_D and T_A-T_D are and how the figures relate to sample timing. This makes it hard to assess other data in the manuscript, including overall conclusions that assess microbiome impact on the host response, which is a primary conclusion that the authors try to address.
2. Although Lp082 probiotic introduction is the primary study intervention, the authors do not mention or discuss Lp082 presence in the stool and its own genomic and metabolic contributions to the host response and the SCFA content. There is a label on Figure 3D that says "Lactobacillus plantarum" but it is not discussed. I'd like to see specific Lp082 evaluation and discussion in their metagenome or via another sampling method (like stool qPCR if samples still exist) that indicates the abundance of Lp082 at the times that they sampled in Figure 3 and preferably discussed in light of the experiments and data discussed in Figures 4-6.

3. The Results section “The regulatory roles of SCFAs” and Figure 4 appear to be among the weaker sections in the paper. The figures are not well described, making it difficult to understand the graphs and interpret the data (specific points made below in “minor points”). Lines 172-175 claim “the contents of acetic acid, propionic acid, butyric acid were significantly decreased in the DSS group but significantly increased in the Lp082 group ($p < 0.05$) (Fig. 4b),” but this information does not match the data in Fig. 4b. Fig. 4b shows that the cecal levels of all five evaluated SCFAs are lower than the control in DSS, Lp082, and SASP. Additionally, none of the five SCFAs are higher in Lp082 cecal contents than DSS, and in most cases, the five SCFAs appear lower in Lp082 than DSS. Thus, Fig 4b contradicts their claim that SCFAs improve host outcomes in response to Lp082 treatment after DSS. This is further reiterated by the rather small fold-change increase in the two pathways they indicate promote SCFA production in Lp082 “*the fermentation of pyruvate to propionate I and the fermentation of pyruvate to acetate and lactate II*” in figure 4A. The authors’ conclusion that Lp082 promotes SCFA production is heavily leveraged in the discussion section, but is not well supported in their data.
4. The authors attempt to model microbial impact on the host using the bacterial metagenome and a host transcriptional analysis. This comparison would be better made if there was a microbial metatranscriptome/proteome included in this paper to support the microbial genomic data. In the absence of this, an evaluation of Lp082 itself in the host, and a weak finding on SCFA changes in response to Lp082, I find the correlations reported in figure 7 to be more speculative rather than well supported by the manuscript.
5. I’d like to see an analysis or discussion of the genes in Figure 6D for Lp082. The authors indicate that these genes in 6D are upregulated in DSS and some are pro-inflammatory. I’d like to know if Lp082 treatment suppresses these genes when compared to DSS alone.
6. Other missing information that should be addressed in the manuscript:
 - a. Rationale:
 - I. Why was SASP used?
 - II. Why was Lp082 used specifically?
 - b. Experimental design, conditions, methods:
 - I. Fig S1B does not adequately describe mouse behavior as it’s a single non-descriptive image of each mouse.
 - c. Timing of experiments: After line 101, the sampling times of most experiments are omitted or inadequately described.
 - d. Sample sizes: Sample sizes and number of repeats are omitted. In most cases, the specific datapoints in figures are not well described as to what they are measuring.
 - e. Statistics:
 - a. Only one statistical test is indicated in the paper, Wilcoxin signed rank test, line 546 in the methods. Adding the test run to each figure legend would be appropriate and helpful.
 - b. Conditions statistical tests being used on are not obvious, in part due to the lack of descriptions on sample sizes and replicates.

Minor points:

1. Missing information that should be addressed:

a. Rationale:

- I. The introduction provides weak descriptions and evidence for use of a probiotic in general to treat UC and Lp082 specifically. The introduction would benefit from further elaboration on what is known about probiotic treatment of UC and indicate what is or isn't known about Lp082 usage in UC specifically rather than using general "probiotics" references. Along with this, lines 55-59 are confusing as written, but this may be addressed when more information is added about those two points.
- II. The intro (starting at line 62) provides weak background on data for SCFA alleviation of IBD. Citing work and adding text of SCFA impact of IBD (preferably UC and action through immune cells) would be helpful.

b. Impact:

- I. Referencing lines 94-115: No text is provided to indicate what the alterations in water intake, food intake, body weight, DAI, neurological responses, immune organ index, spleen and colon color and structure, hyperemia, and feces structure mean in the context of disease in DSS or in the Lp082 treated animals. This is not addressed elsewhere in the paper and would help the reader understand the impact of your results.
- II. Lines 125-145: Text here would benefit from at least a little description on what this data means at this point in the writing. E.g. what does MUC2 loss and ZOI abundance suggest about Lp082 effects?

c. Methods:

- I. Scoring: Since understanding the scoring system used is important to understanding the data, further describing the numbering and what that means would help the reader understand the severity of the DSS model and the subsequent relief without looking up the methods reference (either in the figure legends (Figure 1B) or in the methods (see lines 480-481 where the modifications to DAI are not indicated). DAI and immune organ index should be described at some level in the results and figure legends as well so the reader knows what the data is describing without the methods.)
- II. How was "surface density" quantified? Line 144, figure 2F-G.
- III. Indicate the specific diet provided to the mice (line 459).
- IV. Elaborate on what you mean by "mouse colon samples" on line 537 for RNA-seq.

d. Results structure:

- I. The experiments, including the rationale, the samples, and the conditions, should be described at some level prior to discussing the results in the Results text so the readers know what the results are referencing.
- II. Brief overall conclusions should be provided in the Results text to continue engaging the reader and leading them along your thought process. This can be partially addressed by moving text from the

Discussion section to the Results. E.g. lines 302-306 can be moved to the results section where diversity is discussed.

e. Figures:

I. Figure 1:

Fig 1A – the arrows make it look like PBS only led to weight and colon assessment, probiotics to immune indices, SASP to sequencing.

Collapsing the arrows would address this.

Fig 1B – what's being compared for the stats is not well described

Fig 1C – the bars for stats are shifted (also make sure the lines are the same point thickness for stats in each figure)

Fig 1D – “molding ending” is not described in the text. Rephrase or define. Also decrease the numbers in the X axis as they are too condensed. The title “duration of probiotic intervention (day)” is an incorrect title as this figure shows duration of the entire experiment, including pre-treatment with DSS before probiotics.

Fig 1E – there's no Y-axis label and the datapoints are not described

II. Figure 2:

Fig 2A – you might try to line up your red boxes better so they better represent the blow ups (and make straighter red lines).

Fig 2B – add microscopy information for the antibody stains in the legend and/or the methods section. Although the staining method cites another paper, it's best to include antibody information in the methods section. MUC2, ZO-1, and the blue marker are not labeled in the figure and in the figure legend.

Fig 2C – the y axis is missing a metric

Fig 2f-g – the y axes are missing metrics (as noted above, the method to define these numbers is not stated).

III. Figure 3:

Fig 3A-C groupings not labeled as indicated above

Fig 3D – The meaning of the red highlighting is not indicated in the figure legend. No information is provided about the tree, including what it represents and what the colors indicate. The heat map values are not described – what is being compared and what does a value of zero mean?

IV. Figure 4:

Fig. 4A – It is not entirely clear where this data comes from. My assumption was the metagenome, but the Acetic acid sub section has me unsure. Describe this figure more, taking care to describe what the acetic acid subsection is evaluating.

Fig. 4C-D – A description of the tree components is missing. Describe the correlation analysis more in the text and figure legend.

V. Figure 5: I think this entire figure would be best placed in the supplement as it's really just a sub-point of the contents of figure 6 (but it won't fit in figure 6). You might also remove “distribution” from the title and legend as this suggests tissue spatial information but is not needed.

- VI. Figure 6: Overall, the less color you use, the clearer this figure will be.
Fig 6A-C: I recommend condensing as Fig 6A. Describe what gene ratio is in the figure legend.
Fig 6D-F: I recommend condensing as Fig 6B.
Fig 6G-J: I recommend condensing as Fig 6C. I and j legends are swapped. Describe ifcSE in the legend.
2. The authors confuse whether they are studying Lp082 prevention or treatment of colitis by using verbiage referring to “prevention” and “treatment” interchangeably. This makes it difficult to track what the authors are trying to accomplish (for example, line 60 says “relieving”, lines 76 and 87-88 say “prevention”). Because the authors state that the colitis inducer (DSS) is administered at the time of treatment (Lp082) in the beginning of the Results to evaluate prevention (line 87), but Figure 1A shows that Lp082 is being added at day 8 (so not at the time of induction), I cannot assess which is being studied: Lp082 1) treatment or 2) prevention of UC. My best assumption is that the methods section is correct, and the methods says that DSS is used prior to addition of Lp082, and thus the authors are studying Lp082 relief of colitis. Thus, the language in the paper should be altered to indicate that Lp082 was administered after DSS induced colitis and observed effects are Lp082 alleviation of symptoms, not prevention of symptoms.
 3. The abstract, discussion section, and figure 7B describe the effect of Lp082 on the animal model through the groups: biological barrier, chemical barrier, mechanical barrier, and immune barrier. I don’t recommend subdividing “biological, chemical, and mechanical barrier”, as everything you are referring to is biological, chemical, and mechanical in nature. Rather, use categories akin to “microbiota/microbiome alterations, barrier function improvements, and inflammation reduction.”
 4. In general, the abstract could be re-written to describe the results from a higher level, rather than just listing the altered genes. Close the abstract with a statement connecting the paper results to the broader scientific field.
 5. As written, lines 72-73 suggest Yucha has resistance to acid and bile salts, but I assume that the authors mean Lp082 is resistant. Re-wording the sentence and adding a clarification on what point the authors are trying to make about acid and bile salt resistance would help alleviate the confusion here.
 6. Referring to lines 77-92: The authors interchange physiological results with techniques as if they are the same things. Before describing the specific things you were evaluating, describe what you were looking for at a high level. Then separate physiological indicators from methods (e.g., rather than say, “evaluated physiological indexes and shotgun metagenomic sequencing,” use language like “evaluated inflammation, microbial community composition and activity...using ELISA, immunohistochemistry, metagenomic sequencing, and RNA-seq.”
 7. Potentially incorrect information: Lines 97-98 days and scores do not line up with the data reported in figure 1B.
 8. Abbreviations should be described in the text as they arise, not in an additional section at the end of the paper (page 20).

9. After revising the manuscript, a thorough and detailed assessment and correction of sentence structure would improve the readability of the paper dramatically.
10. Abbreviations, capitalization, italics, and spacing are inconsistent throughout and should be fixed for a final draft. E.g. Lp082 (most commonly used in the draft)/LP082 (lines 78-79) or HNU082 (correct)/*HNU082* (line 23).
11. Review your usage of “prove” in your manuscript (notably in the discussion section) as the experiments presented provide largely correlative data.

**Manuscript No.: Spectrum 01651-22**
**Title: Probiotics (*Lactobacillus plantarum* HNU082) supplementation relieves**
**ulcerative colitis by affecting intestinal barrier functions, immunity-related genes**
**expression, gut microbiota, and metabolic pathways in mice.**

**Dear Dr. Xiaoyu Tang,**

On behalf of my co-authors, I thank you very much for allowing us to revise our
manuscript. We appreciate the time and effort that you and the reviewers dedicated to
providing feedback on our manuscript and are grateful for the insightful comments on
and valuable improvements to our manuscript. We have discussed reviewer's
comments carefully and revised the manuscript taking all the comments positively.
All revisions in the manuscript have been highlighted in yellow. Please find the
point-to-point responses to reviewers' comments in the following text. We thoroughly
double-checked the manuscript. In addition, the revised manuscript with tracked
changes is also uploaded as "Marked Up Manuscript" files.

The sequence data reported in this paper have been deposited in the NCBI
database (metagenomic sequencing data and transcriptome sequencing
data:PRJNA812272). As is customary, our data will be made public after the article is
received.

We would like to have this revised manuscript considered for publication in
"*Microbiology Spectrum*." We deeply appreciate your consideration of our manuscript.
If you have any queries, please don't hesitate to contact us at the following e-mail
address.

We would like to express our great appreciation again to you and the reviewers for
their comments on our paper. We are looking forward to hearing from you.

Sincerely,
Jiachao Zhang

Yours sincerely,
E-mail: Jiachao Zhang^{1*}, zhjch321123@163.com
College of Food Science and Engineering, Hainan University, Haikou 570228, China

**Reviewer #1:**

Reviewer #1 (Public repository details (Required)):

Metagenome and transcriptome raw data

**Response:** We are very sorry for our negligence of metagenome and transcriptome
raw data. We have uploaded the metagenomic and transcriptome raw data, and the
modifications in the manuscript have been highlighted. (Page 27, Line: 790-792)

The sequence data reported in this paper have been deposited in the NCBI
database (metagenomic sequencing data and transcriptome sequencing
data:PRJNA812272).

As is customary, our data will be made public after the article is received.

Reviewer #1 (Comments for the Author):

Article summary and impression:

In the article Spectrum 01651-22, the authors seek to describe the impact of
supplementation of the food-derived bacterial strain *Lactobacillus plantarum* HNU082
(Lp082) on the commonly used DSS-induced IBD model in C57BL/6J adult male
mice with otherwise normal microbiota and diet. The authors induce inflammation
with DSS supplementation in animal water, stop DSS supplementation, and then add
either Lp082 or the compound SASP (although the rationale for using SASP is not
provided, I assume this is a positive control for alleviation of DSS induced
inflammation) to evaluate Lp082 impacts on DSS treated mice. The authors perform a
number of analyses in an attempt to provide a comprehensive assessment of the
impact of Lp082 treatment on DSS treated mice including the following: assessment
of 1) animal behavior, 2) immune organ weight, 3) serum inflammatory cytokines, 4)

colon structure and histopathology and stool formation, 5) colonic mucin and tight
junction integrity, 6) microbial taxa changes and abundance, 7) SCFAs acid content, 8)
host epithelial transcriptional responses, as well as an attempt to connect microbiome
changes to host physiology through correlation modeling. If presented accurately and
completely, such a compilation is a useful addition to the scientific community and
would provide a greater understanding of the impact of *Lactoplantibacillus* on colitis
in healthy mouse models. However, the current version of the manuscript has a
number of shortcomings, many of which are summarized below. Overall, the text and
figures are confusing to follow as key information required to accurately assess the
data and author conclusions has been left out. Information omission begins at the
beginning of the paper and builds to where it's difficult to assess the content and
accuracy of subsequent data.

**Response:** We appreciate the time and effort you dedicated to providing feedback on
our manuscript and are grateful for the insightful comments and valuable
improvements to our manuscript. We have discussed your comments carefully, and we
sincerely accept the suggestions. Your comments provided valuable insights to refine
its contents and analysis. In this document, we try to address the issues raised as best
as possible. All revisions in the manuscript have been highlighted in yellow. A list of
changes to the manuscript has been attached, and you can kindly find the
point-to-point responses to your comments in the following text.

Preface to the following comments:

The manuscript does not use page numbers and line numbers. To review this
document, I exported the pdf to word and refer to the title page as page 1, with the
first line of the title being line 1.

**Response:** We appreciate your helpful comments. It was a mistake. We have added
the page number and line number to the manuscript now. The title page is also called
page 1, and the first line of the title is line 1.

Major points:

1. Conditions used in figure 3A-D are inadequately described, such that I cannot
sufficiently assess sample timing, sample size, comparisons made, and biological
meaning. A primary contributor to this is a lack of a clear description on what
M_A-M_D and T_A-T_D are and how the figures relate to sample timing. This
makes it hard to assess other data in the manuscript, including overall conclusions that
assess microbiome impact on the host response, which is a primary conclusion that
the authors try to address.

**Response:** We are extremely grateful to the you for pointing out this problem. We are
very sorry for the inadequacy of the condition description. We have added the **Fig. S3**
to describe the sampling time and grouping of metagenomics sequencing. In addition,
we provide supplementary descriptions of all sample times, sample sizes, and
biological significance in the materials and methods and results sections, and
modifications in the manuscript are highlighted in yellow. A detailed description of
**Fig. S3** has been added to Supplemental materia. (Page 2, Line: 22-33)

**SUPPLEMENTARY FIGURE LEGENDS**

**Fig. S3**

(a) Timepoints and grouping of mouse metagenomic sequencing

M means the modeling period, T means the treatment period. Respectively, A, B, C

and D group mean 7 days normal water (ultrapure water), DSS, Lp082 and SASP
treatment after 7 days DSS gavage.

M-A means A group represents the control group on the 7th day of DSS
modeling, M-B represents the DSS group on the 7th day of DSS modeling, M-C
represents the Lp082 group on the 7th day of DSS modeling, M-D represents the
SASP on the 7th day of DSS treatment Group.

T-A means treating-A group represents the control group at the end of the
treatment, T-B represents the DSS group at the end of the treatment, T-C represents
the Lp082 group at the end of the treatment, and T-D represents the SASP group at the
end of the treatment.

As shown above, we collected mice fecal samples from group A (Control, $n=6$),
group B (DSS, $n=6$), group C (Lp082, $n=6$) and group D (SASP, $n=6$) on days 7 and
15 for metagenomic sequencing. On days 1-7, mice in the group B, group C and
group D drank DSS-containing water freely, the mice in the group A drank normal
water (ultrapure water). On days 8-15, group B, C and D mice stopped drinking DSS
water, Mice in groups A and B were gavaged with PBS water, mice in group C were
gavaged in PBS water and Lp082, and mice in group D were gavaged in PBS water
and SASP. The 7th day was the end of DSS modeling and the 15th day was the end of
Lp082 and SASP treatment, so we chose to take samples from the two key time points
for sequencing to observe the effect of DSS, Lp082 and SASP on the gut microbiome.
We are grateful for the suggestion.

2. Although Lp082 probiotic introduction is the primary study intervention, the
authors do not mention or discuss Lp082 presence in the stool and its own genomic
and metabolic contributions to the host response and the SCFAs content. There is a
label on Figure 3D that says "*Lactobacillus plantarum*" but it is not discussed. I'd like
to see specific Lp082 evaluation and discussion in their metagenome or via another
sampling method (like stool qPCR if samples still exist) that indicates the abundance
of Lp082 at the times that they sampled in Figure 3 and preferably discussed in light

of the experiments and data discussed in Figures 4-6.

**Response:** We appreciate your valuable and helpful comment. Previous studies [1],
have shown that the abundance of *lactobacillus plantarum* in mice was 0 [2], and it
was also found in our experiment (during modeling period, the abundance of
*lactobacillus plantarum* in control group (M-A), DSS group (M-B), Lp082 group
(M-C) and SASP group (M-D) was 0 , and during the treatment period, the abundance
of *lactobacillus plantarum* in the control group (T-A), DSS group (T-B) and SASP
group (T-D) was 0.), but we found that the abundance of *lactobacillus plantarum*
increased in the Lp082 group (T-C) only after *lactobacillus plantarum* HNU082
(Lp082) treatment. This is consistent with Wang et al [3] and Huang et al [4] that
probiotic Lp082 can colonize the mouse gut. Therefore, in our experiment, we can
infer that the change in *lactobacillus plantarum* was due to the probiotic Lp082
intake.

Added discussion (Page 10, Line: 287-295)

Next, we conducted a correlation analysis between Lp082 (*lactobacillus*
*plantarum*) and SCFAs, and found that Lp082 (*lactobacillus plantarum*) was strongly
positively correlated with SCFAs (acetic acid, propionic acid, butyric acid) (**Fig. 4c**),
the correlation results suggested that Lp082 can increase the content of SCFAs. The
above results inspired us to further explore the relationship between Lp082 and
SCFAs, and we further analyzed the bacterial species and metabolic pathways
associated with SCFAs. Further metagenomic data provided support for our above
speculation. Combined with metagenomic data, the species composition of mice gut
microbiota was further analyzed. The results showed that the relative abundance of
some special bacteria increased in the Lp082 group, such as, *lactobacillus plantarum*,
*Bifidobacterium pseudolongum*, *Akkermansia muciniphila*, *Bacteroides ovatus*,
*Parabacteroides distasonis*, *Lactobacillus reuteri*, *Anaerotruncus sp G3 2012* (these
bacteria are highlighted in red in **Fig. 3d**), all of which can metabolize produces the
SCFAs [5].

Subsequently, we further analyzed the metabolic pathways of gut microbiota in
mice. Results of differential metabolic pathways showed that the abundance of gut

microbiota metabolic pathways related SCFAs production decreased in DSS group but
increased in Lp082 group (**Fig. 4a**). We infer that Lp082 can promote the content of
SCFAs (acetate, propionate and butyrate) by adjust three metabolic pathways,
including Pyruvate fermentation to Propanoate I, Pyruvate fermentation to acetate and
lactate II, Acetyl CoA fermentation to Butanoate (**Fig. 4a**).

To prove the above findings, we further used gas chromatography-mass
spectrometry (GC-MS) to detect the content of SCFAs. Compared with control group,
the contents of butyric acid, valeric acid, acetic acid, propionic acid and isobutyric
acid were significantly decreased after ingestion of DSS ($P < 0.01$). Compared with
DSS group, the contents of butyric acid, acetic acid, propionic acid and isobutyric
acid were extremely significant increased after ingestion of Lp082 ($P < 0.01$). This
confirmed our previous hypothesis based on the correlation that Lp082 intake would
increase SCFAs levels (**Fig. 4b**). Based on the above results, we speculate that Lp082
increase the content of SCFAs by affecting the abundance of SCFAs-producing
microbes, as well as the metabolic pathways of SCFAs-producing microbes.

Reference

- 1. Pan Y, Ning Y, Hu J, Wang Z, Chen X, Zhao X. The Preventive Effect of
*lactobacillus plantarum* ZS62 on DSS-Induced IBD by Regulating Oxidative Stress
and the Immune Response. *Oxid Med Cell Longev*. 2021;2021:9416794; doi:
10.1155/2021/9416794.
- 2. Shao Y, Huo D, Peng Q, Pan Y, Jiang S, Liu B, et al. *lactobacillus plantarum*
HNU082-derived improvements in the intestinal microbiome prevent the development
of hyperlipidaemia. *Food & Function*. 2017;8(12):4508-16; doi: 10.1039/c7fo00902j.
- 3. Wang Y, li J, Ma C, Jiang S, Li C, Zhang L, et al. *lactobacillus plantarum*
HNU082 inhibited the growth of *Fusobacterium nucleatum* and alleviated the
inflammatory response introduced by *F. nucleatum* invasion. *Food & Function*.
2021;12(21):10728-40; doi: 10.1039/d1fo01388b.
- 4. Huang S, Jiang S, Huo D, Allaband C, Estaki M, Cantu V, et al. Candidate
probiotic *lactobacillus plantarum* HNU082 rapidly and convergently evolves within
human, mice, and zebrafish gut but differentially influences the resident microbiome.

Microbiome. 2021;9(1); doi: 10.1186/s40168-021-01102-0.

5. Y. W. Cheng, J. M. Liu and Z. X. Ling, Short-chain fatty acids-producing
probiotics: A novel source of psychobiotics, *Critical Reviews in Food Science and*
*Nutrition*, DOI: 10.1080/10408398.2021.1920884.

3. The Results section "The regulatory roles of SCFAs" and Figure 4 appear to be
among the weaker sections in the paper. The figures are not well described, making it
difficult to understand the graphs and interpret the data (specific points made below in
"minor points"). Lines 172-175 claim "the contents of acetic acid, propionic acid,
butyric acid were significantly decreased in the DSS group but significantly increased
in the Lp082 group ($p < 0.05$) (Fig. 4b)," but this information does not match the data
in Fig. 4b. Fig. 4b shows that the cecal levels of all five evaluated SCFAs are lower
than the control in DSS, Lp082, and SASP. Additionally, none of the five SCFAs are
higher in Lp082 cecal contents than DSS, and in most cases, the five SCFAs appear
lower in Lp082 than DSS. Thus, Fig 4b contradicts their claim that SCFAs improve
host outcomes in response to Lp082 treatment after DSS. This is further reiterated by
the rather small fold-change increase in the two pathways they indicate promote
SCFAs production in Lp082 "the fermentation of pyruvate to propionate I and the
fermentation of pyruvate to acetate and lactate II" in figure 4A. The authors'
conclusion that Lp082 promotes SCFAs production is heavily leveraged in the
discussion section, but is not well supported in their data.

**Response:** We apologize for any confusion caused and appreciate the valuable
suggestions. We sincerely thank you for pointing out the inconsistency between the
figure information and the manuscript information. After carefully examining and
comparing of the original drawing data, we found that the grouping in **Fig. 4b** was
wrong. We sincerely apologize for this, and the correct grouping is as follows. In **Fig.**
**4b**, red represents the control group, yellow represents the Lp082 group, blue
represents the SASP group, and green represents the DSS group. The content of
SCFAs described in the original manuscript is based on the correct grouping
mentioned above. We have revised the grouping of **Fig. 4b** and carefully checked all

the figures and full text to ensure the consistency of the manuscript and figures. In
addition, we have rewritten the results section "The regulatory roles of SCFAs" and
we have redescribed all panels in **Figure 4** including **Fig. 4a-Fig. 4d**. (Page 10, Line:
286-346). All revisions in the manuscript have been highlighted.

**The regulatory role of SCFAs**

Next, we conducted a correlation analysis between Lp082 (*Lactobacillus*
*plantarum*) and SCFAs, and found that Lp082 (*Lactobacillus plantarum*) was strongly
positively correlated with SCFAs (acetic acid, propionic acid, butyric acid) (**Fig. 4c**),
the correlation results suggested that Lp082 can increase the content of SCFAs. The
above results inspired us to further explore the relationship between Lp082 and
SCFAs, and we further analyzed the bacterial species and metabolic pathways
associated with SCFAs. Further metagenomic data provided support for our above
speculation. Combined with metagenomic data, the species composition of mice gut
microbiota was further analyzed. The results showed that the relative abundance of
some special bacteria increased in the Lp082 group, such as, *Lactobacillus plantarum*,
*Bifidobacterium pseudolongum*, *Akkermansia muciniphila*, *Bacteroides ovatus*,
*Parabacteroides distasonis*, *Lactobacillus reuteri*, *Anaerotruncus sp G3 2012* (these
bacteria are highlighted in red in **Fig. 3d**), all of which can metabolize produces the
SCFAs [1].

Subsequently, we further analyzed the metabolic pathways of gut microbiota in
mice. Results of differential metabolic pathways showed that the abundance of gut
microbiota metabolic pathways related SCFAs production decreased in DSS group but
increased in Lp082 group (**Fig. 4a**). We infer that Lp082 can promote the content of
SCFAs (acetate, propionate and butyrate) by adjust three metabolic pathways,
including Pyruvate fermentation to Propanoate I, Pyruvate fermentation to acetate and
lactate II, Acetyl CoA fermentation to Butanoate (**Fig. 4a**).

To prove the above findings, we further used gas chromatography-mass
spectrometry (GC-MS) to detect the content of SCFAs. Compared with control group,
the contents of butyric acid, valeric acid, acetic acid, propionic acid and isobutyric
acid were significantly decreased after ingestion of DSS ($P < 0.01$). Compared with

DSS group, the contents of butyric acid, acetic acid, propionic acid and isobutyric
acid were extremely significant increased after ingestion of Lp082 ($P < 0.01$). This
confirmed our previous hypothesis based on the correlation that Lp082 intake would
increase SCFAs levels (**Fig. 4b**). Based on the above results, we speculate that Lp082
increase the content of SCFAs by affecting the abundance of SCFAs-producing
microbes, as well as the metabolic pathways of SCFAs-producing microbes.

To further understand the role of SCFAs, we performed a Pearson correlation
analysis. The results showed that *helicobacter hepatica*, which was significantly
increased in the DSS group, was strongly negatively correlated with acetic acid,
propionic acid, and butyric acid (**Fig. 4c**). *lactobacillus plantarum*, *Bifidobacterium*
*pseudolongum*, *Akkermansia muciniphila*, *Parabacteroides distasonis*, *Lactobacillus*
*reuteri*, which were significantly increased in Lp082 group showed strong positive
correlation with acetic acid, propionic acid, and butyric acid. *Anaerotruncus sp G3*
*2012* and *Bacteroides ovatus* showed a strong positive correlation with butyric acid
and acetic acid, and a weak positive correlation with propionic acid (**Fig. 4c**). These
SCFAs including acetic acid, propionic acid, and butyric acid were all strong
negatively correlation with the pro-inflammatory factors TNF- α , IL-1 β , IFN- γ , IL-6,
MPO but strongly positively correlated with the inflammatory suppressor IL-10 (**Fig.**
**4d**). As important products of gut microbiota metabolism, SCFAs have certain
anti-inflammatory effects and play an important role in maintaining normal intestinal
morphology and function. Combined with the results of **Fig. 3d**, **Fig. 4a-4d**, as well
as the improvement of physiological indicators (**Fig. 1b-1d**), pathological indicators
(**Fig. 2a-2g**) and inflammatory factors (**Fig. 1e**) after ingestion of Lp082, we
speculated that Lp082 may alleviate DSS-induced UC by regulating SCFAs through
the following mechanisms (**Fig. S4**). That is, after the ingestion of Lp082, the
abundance of the intestinal microbes of SCFAs-producing increased, which promoted
the content of SCFAs. The SCFAs has the function of promoting the secretion of
inflammatory cytokine and suppressing the secretion of inflammatory factors. The
changes in inflammatory cytokines affect the physiological indicators of mice, which
increases the weight, colon length, drinking water and eating volume of mice, and

reduces the DAI score and immune organs index. The changes in inflammatory
 cytokines also affected the pathological indexes of mice, resulting in a decrease in
 histopathological score and an increase in immunofluorescence protein content of
 ZO-1 and MUC-2.

**Reference**

1. Y. W. Cheng, J. M. Liu and Z. X. Ling, Short-chain fatty acids-producing
 probiotics: A novel source of psychobiotics, Critical Reviews in Food Science and
 Nutrition, DOI: 10.1080/10408398.2021.1920884.

**Fig. 4**

The important role of SCFAs in alleviation of DSS-induced UC.

**Fig. S4**

**The underlying mechanism by which Lp082 regulates SCFAs to alleviate UC**

4. The authors attempt to model microbial impact on the host using the bacterial
 metagenome and a host transcriptional analysis. This comparison would be better
 made if there was a microbial metatranscriptome/proteome included in this paper to
 support the microbial genomic data. In the absence of this, an evaluation of Lp082
 itself in the host, and a weak finding on SCFAs changes in response to Lp082, I find
 the correlations reported in figure 7 to be more speculative rather than well supported
 by the manuscript.

**Response:** We appreciate your valuable and helpful comment. Indeed, it is a pity that
 the microbiome lacks transcriptome, but the absence of a microbial transcriptome in
 the Cordeiro et al. [1] and Wang et al. [2] articles did not affect the demonstration of
 the impact of microorganisms on the host.

Previous studies [3], have shown that the abundance of *lactobacillus plantarum*
 in mice was 0 [4], and it was also found in our experiment (during modeling period,
 the abundance of *lactobacillus plantarum* in control group (M-A), DSS group (M-B),
 Lp082 group (M-C) and SASP group (M-D) was 0 , and during the treatment period,
 the abundance of *lactobacillus plantarum* in the control group (T-A), DSS group (T-B)
 and SASP group (T-D) was 0.), but we found that the abundance of *lactobacillus*
 *plantarum* increased in the Lp082 group (T-C) only after *lactobacillus plantarum*
 Lp082 treatment. This is consistent with Wang et al [5] and Huang et al [6] that
 probiotic Lp082 can colonize the mouse gut. Therefore, in our experiment, we can
 infer that the change in *lactobacillus plantarum* was due to the probiotic Lp082
 intake.

Added discussion (Page 10, Line: 287-318)

Next, we conducted a correlation analysis between Lp082 (*Lactobacillus*
*plantarum*) and SCFAs, and found that Lp082 (*Lactobacillus plantarum*) was strongly
positively correlated with SCFAs (acetic acid, propionic acid, butyric acid) (**Fig. 4c**),
the correlation results suggested that Lp082 can increase the content of SCFAs. The
above results inspired us to further explore the relationship between Lp082 and
SCFAs, and we further analyzed the bacterial species and metabolic pathways
associated with SCFAs. Further metagenomic data provided support for our above
speculation. Combined with metagenomic data, the species composition of mice gut
microbiota was further analyzed. The results showed that the relative abundance of
some special bacteria increased in the Lp082 group, such as, *Lactobacillus plantarum*,
*Bifidobacterium pseudolongum*, *Akkermansia muciniphila*, *Bacteroides ovatus*,
*Parabacteroides distasonis*, *Lactobacillus reuteri*, *Anaerotruncus sp G3 2012* (these
bacteria are highlighted in red in **Fig. 3d**), all of which can metabolize produces the
SCFAs [7].

Subsequently, we further analyzed the metabolic pathways of gut microbiota in
mice. Results of differential metabolic pathways showed that the abundance of gut
microbiota metabolic pathways related SCFAs production decreased in DSS group but
increased in Lp082 group (**Fig. 4a**). We infer that Lp082 can promote the content of
SCFAs (acetate, propionate and butyrate) by adjust three metabolic pathways,
including Pyruvate fermentation to Propanoate I, Pyruvate fermentation to acetate and
lactate II, Acetyl CoA fermentation to Butanoate (**Fig. 4a**).

To prove the above findings, we further used gas chromatography-mass
spectrometry (GC-MS) to detect the content of SCFAS. Compared with control group,
the contents of butyric acid, valeric acid, acetic acid, propionic acid and isobutyric
acid were significantly decreased after ingestion of DSS ($P < 0.01$). Compared with
DSS group, the contents of butyric acid, acetic acid, propionic acid and isobutyric
acid were extremely significant increased after ingestion of Lp082 ($P < 0.01$). This
confirmed our previous hypothesis based on the correlation that Lp082 intake would
increase SCFAs levels (**Fig. 4b**). Based on the above results, we speculate that Lp082

increase the content of SCFAs by affecting the abundance of SCFAs-producing
microbes, as well as the metabolic pathways of SCFAs-producing microbes.

The above evidence is obtained from actual measurements, and the data is
objective and true, which is enough to prove that the increase in SCFAs is indeed
caused by the introduction of Lp082.

**Fig. 6a** (original named **Fig. 7a**) is a comprehensive network diagram. We have
performed pearson correlation analysis based on the actual measured data and
simulated possible mechanisms. In **Fig. 6a**, red lines indicate positive correlation,
blue lines indicate negative correlation, and thicker lines indicate stronger correlation.
The purpose of this picture is to combine the possible mechanism diagrams to better
understand the theme of the article, which is the usual method of many [8] articles [9].
**Fig. 6a** does not only analyze the correlation, we have really done a lot of
experiments and verifications in it. First, we studied some basic indicators and found
that Lp082 could not only significantly inhibit the decrease of body weight, water
intake and food intake induced by DSSS in mice, but also significantly inhibit the
increase of DAI and immune organ index induced by DSSS, as well as the decrease of
colon length caused by DSS (**Fig. 1a-1d**). Second, we measured the protein content of
six inflammatory cytokines in mouse serum, and found that Lp082 could significantly
reduce the increase of IL-1 β , IL-6, TNF- α , MPO, IFN- γ induced by DSS, and increase
the protein content of IL-10 in mice (**Fig. 1e**). Third, we performed HE staining
section experiment and immunofluorescence protein experiment. The results showed
that Lp082 could not only improve the crypt infiltration, goblet cell loss and intestinal
mucosal ulcer induced by DSS, but also could reduce the increase of histopathology
score caused by DSS and reduce the loss of ZO-1 and MUC-2 proteins caused by
DSS (**Fig. 2a-2g**). Fourth, we collected fecal samples on day 7 for metagenomic
sequencing. The results of Shotgun metagenomic data analysis showed that Lp082
could increase α -diversity and β -diversity, reduce the differences in species
composition, increase the content of beneficial bacteria and inhibit the abundance of
harmful bacteria in mice (**Fig. 3a-3d**). Fifth, we used gas chromatography-mass
spectrometry to determine the content of SCFAs in the intestinal contents of mice, and

found that Lp082 could significantly inhibit the reduction of acetic acid, propionic
acid, butyric acid, isobutyric acid and valeric acid induced by DSS, and restore the
content of SCFAs in mice (**Fig. 4b**). Sixth, we sequenced the transcriptome of colon
tissue, and the results showed that Lp082 not only affected gene expression
distribution, but also affected inflammation and cancer-related and KEGG,GO-BP
pathways (**Fig. 5a-5g**). From the above, it can be seen that our correlations are not
unreasonable speculation, but are based on experimental data from a large number of
real measurements. Our data were not less than 6 replicates in each group, and our
data were absolutely reliable . Collectively, our current data are objective and accurate
enough to support our conclusions.

**Reference**

- 1. Cordeiro BF, Alves JL, Belo GA, Oliveira ER, Braga MP, da Silva SH, et al.
Therapeutic Effects of Probiotic Minas Frescal Cheese on the Attenuation of UC in a
Murine Model. *Frontiers in Microbiology*. 2021;12; doi: 10.3389/fmicb.2021.623920.
- 2. Wang J, Ji HF, Wang SX, Liu H, Zhang W, Zhang DY, et al. Probiotic
*lactobacillus plantarum* Promotes Intestinal Barrier Function by Strengthening the
Epithelium and Modulating Gut Microbiota. *Frontiers in Microbiology*. 2018;9; doi:
10.3389/fmicb.2018.01953.
- 3. Pan Y, Ning Y, Hu J, Wang Z, Chen X, Zhao X. The Preventive Effect of
*lactobacillus plantarum* ZS62 on DSS-Induced IBD by Regulating Oxidative Stress
and the Immune Response. *Oxid Med Cell Longev*. 2021;2021:9416794; doi:
10.1155/2021/9416794.
- 4. Shao Y, Huo D, Peng Q, Pan Y, Jiang S, Liu B, et al. *lactobacillus plantarum*
HNU082-derived improvements in the intestinal microbiome prevent the development
of hyperlipidaemia. *Food & Function*. 2017;8(12):4508-16; doi: 10.1039/c7fo00902j.
- 5. Wang Y, li J, Ma C, Jiang S, Li C, Zhang L, et al. *lactobacillus plantarum*
HNU082 inhibited the growth of *Fusobacterium nucleatum* and alleviated the
inflammatory response introduced by *F. nucleatum* invasion. *Food & Function*.
2021;12(21):10728-40; doi: 10.1039/d1fo01388b.
- 6. Huang S, Jiang S, Huo D, Allaband C, Estaki M, Cantu V, et al. Candidate

probiotic *Lactobacillus plantarum* HNU082 rapidly and convergently evolves within
human, mice, and zebrafish gut but differentially influences the resident microbiome.
*Microbiome*. 2021;9(1); doi: 10.1186/s40168-021-01102-0.

7. Y. W. Cheng, J. M. Liu and Z. X. Ling, Short-chain fatty acids-producing
probiotics: A novel source of psychobiotics, *Critical Reviews in Food Science and*
*Nutrition*, DOI: 10.1080/10408398.2021.1920884.

8. Ma C, Wasti S, Huang S, Zhang Z, Mishra R, Jiang S, et al. The gut microbiome
stability is altered by probiotic ingestion and improved by the continuous
supplementation of galactooligosaccharide. *Gut Microbes*. 2020;12(1); doi:
10.1080/19490976.2020.1785252.

9. Z. P. Gu, Y. J. Zhu, S. M. Jiang, G. H. Xia, C. Li, X. Y. Zhang, J. C. Zhang and X.
R. Shen, Tilapia head glycolipids reduce inflammation by regulating the gut
microbiota in dextran sulphate sodium-induced colitis mice, *Food & Function*, 2020,
11, 3245-3255.

5. I'd like to see an analysis or discussion of the genes in Figure 6D for Lp082. The
authors indicate that these genes in 6D are upregulated in DSS and some are
pro-inflammatory. I'd like to know if Lp082 treatment suppresses these genes when
compared to DSS alone.

**Response:** We are grateful for the suggestion. We have added a more detailed
interpretation regarding analysis and discussion of Lp082 gene. More detailed
statistical analysis was added in the paper. Supplementary Figure **Fig. S6** illustrates
the effect of Lp082 treatment on up-regulated inflammatory genes in the DSS group
in **Fig. 6d**.

Our previous analysis idea was as follows: Since the preliminary analysis of
transcriptome data showed that the intake of Lp082 affects the gene expression
distribution (**Fig. 5**), in order to explore whether Lp082 also affects gene enrichment
pathways, we analyzed the GO pathway and KEGG pathway.

Since the differentially expressed genes (DEGs) were more enriched in the
biological process (BP) pathway among the three major GO pathway categories (**Fig.**

**5a-5c**), and the number of significantly up-regulated genes in Lp082 group is more
than the down-regulated genes compared with the DSS group (**Fig. 5d**), so we
performed further GO-BP analysis on the significantly up-regulated differentially
expressed genes (**Fig. 6d-6f**). Therefore, in **Fig. 6d**, more attention was paid to
inflammatory pathways enriched by up-regulated genes in the DSS group. We added
**Fig. S6** to see the changes of genes enriched in inflammatory pathways in the DSS
group, and their changes in the Lp082 group. We have supplemented **Fig. S6** content
in the article and highlighted it, the supplementary content is as follows (Page 14, line:
385-391):

To further observe whether Lp082 treatment would suppress these inflammatory
and cancer genes enriched on inflammatory pathways in the DSS group, we
supplemented Fig. S6. As can be seen from Fig. S6, among the 13 inflammatory genes
or oncogenes that were up-regulated and enriched in the inflammatory pathway in the
DSS group, the following 10 genes were significantly down-regulated in the Lp082
group: IL-1 β , IL-1 α , Ereg, IL-1rn, Fga, Ldlr, Dgat2, Mfsd2a, Cdc7, Dbf4 (Fig. S6).

A supplementary legend to Figure S6 has been added to the supplementary material
(Page 6, line: 51-58)

**SUPPLEMENTARY FIGURE LEGENDS**

**Fig. S6.** The effect of Lp082 treatment on up-regulated inflammatory genes in the
DSS group in **Fig. 6d**.

The 13 inflammatory genes or oncogenes that were up-regulated and enriched in the
inflammatory pathway in the DSS group, the following 10 genes were significantly
down-regulated in the Lp082 group: IL-1 β , IL-1 α , Ereg, IL -1rn, Fga, Ldlr, Dgat2,
Mfsd2a, Cdc7, Dbf4.

Wilcoxon signed-rank test is used here. Each group had at least 6 biological
replicates.

2. Other missing information that should be addressed in the manuscript:

a. Rationale:

I. Why was SASP used?

**Response:** Thank you for pointing this out. We have supplemented the description of
SASP, and relevant content has been added to the manuscript now (Page 5, line:
126-132). The details are as follows:

Sulfasalazine (SASP) is a commonly used medicine to treat UC at present [1].
Sulfasalazine is hydrolyzed into 5'-aminosalicylic acid and sulfamerydine by
intestinal bacteria when it enters the human intestine. The decomposed 5'
-aminosalicylic acid not only has good anti-inflammatory and antibacterial effects but
also can effectively suppress the outbreak of UC through immunosuppression [2].
Zhipeng Gu [3] used SASP as the positive control group of tilapia head sugar lipids in
the treatment of colitis.

Therefore, SASP was selected as the positive control group for Lp082 in the
treatment of UC.

**Reference**

- 1. Steinhart AH, Hemphill D, Greenberg GR. Sulfasalazine and mesalazine for the
maintenance therapy of Crohn's disease: a meta-analysis. The American journal of
gastroenterology. 1994;89(12):2116-24.
- 2. Klotz U, Maier K, Fischer C, Heinkel K. Therapeutic efficacy of sulfasalazine

and its metabolites in patients with UC and Crohn's disease. The New England journal
of medicine. 1980;303(26):1499-502; doi: 10.1056/nejm198012253032602.

3. Gu ZP, Zhu YJ, Jiang SM, Xia GH, Li C, Zhang XY, et al. Tilapia head
glycolipids reduce inflammation by regulating the gut microbiota in dextran sulphate
sodium-induced colitis mice. Food & Function. 2020;11(4):3245-55; doi:
10.1039/d0fo00116c.

II. Why was Lp082 used specifically?

**Response:** We are grateful for the suggestion. We have added a more detailed
interpretation regarding Lp082. Relevant content has been added to the text (Page 4,
line: 98-111). The revised content is as follows:

The strain of *Lactobacillus plantarum* HNU082 (Lp082) was originally isolated
from a traditional fermented food-fish tea of the Li people in Hainan Province,
China ,which has a good safety profile and tolerance to acids and bile salts [1]. The
results of Lp082 whole genome sequencing showed showed that this bacterium has
great potential to develop as a probiotic in terms of physiology and function [2]. In
our previous study, Lp082 not only can enhance the ecological and genetic stability of
the intestinal microbiota [3]. But also can inhibit the growth of *Fusobacterium*
*nucleatum* and reduce the inflammatory response [4]. Previous studies have also
shown that Lp082 exerts a preventive effect on hyperlipidemia through the
modulation of metabolism [5]. In addition, ingestion of Lp082 and supplementation
with prebiotics improved the stability of the intestinal microbiota and reduced the
occurrence of disorders associated with disease. These results invariably demonstrate
the probiotic potential of Lp082. However, the treatment effect of Lp082 on UC has
not been studied.

Therefore, we chose Lp082 to study the mechanism of probiotics in treating UC.

**Reference**

1. Zhang J, Wang X, Huo D, Li W, Hu Q, Xu C, et al. Metagenomic approach
reveals microbial diversity and predictive microbial metabolic pathways in Yucha, a
traditional Li fermented food. Scientific Reports. 2016;6; doi: 10.1038/srep32524.

2. Ma C, Wasti S, Huang S, Zhang Z, Mishra R, Jiang S, et al. The gut microbiome
stability is altered by probiotic ingestion and improved by the continuous
supplementation of galactooligosaccharide. *Gut Microbes*. 2020;12(1); doi:
10.1080/19490976.2020.1785252.

3. Huang S, Jiang S, Huo D, Allaband C, Estaki M, Cantu V, et al. Candidate
probiotic *Lactiplantibacillus plantarum* HNU082 rapidly and convergently evolves
within human, mice, and zebrafish gut but differentially influences the resident
microbiome. *Microbiome*. 2021;9(1); doi: 10.1186/s40168-021-01102-0.

4. Wang Y, li J, Ma C, Jiang S, Li C, Zhang L, et al. *Lactiplantibacillus plantarum*
HNU082 inhibited the growth of *Fusobacterium nucleatum* and alleviated the
inflammatory response introduced by *F. nucleatum* invasion. *Food & Function*.
2021;12(21):10728-40; doi: 10.1039/d1fo01388b.

5. Shao Y, Huo D, Peng Q, Pan Y, Jiang S, Liu B, et al. *Lactobacillus plantarum*
HNU082-derived improvements in the intestinal microbiome prevent the development
of hyperlipidaemia. *Food & Function*. 2017;8(12):4508-16; doi: 10.1039/c7fo00902j.

b. Experimental design, conditions, methods:

I. Fig S1B does not adequately describe mouse behavior as it's a single
non-descriptive image of each mouse.

**Response:** We are grateful for the suggestion. As suggested by the reviewer, we have
added more details of mouse behavior. Relevant content has been added to the text
(Page 7, line: 185-197). The details are as follows:

The mental state of the mice was observed daily, and the results are shown in **Fig.**
**S1 b**. On the 7th day of modeling, mice in the control group were in a normal state,
with normal urine and feces, shiny hair, active spirit, sensitive reaction and increased
body size. However, mice in the B,C and D group had yellow and smelly urine,
difficult defecation, bloody stool, dark and fried hair, slow reaction and easy panic,
arched back, and reduced body size (**Fig. S1 b**). On the last day of treatment (Day 15),
compared with the arched back, retarded response, hematochezia and lethargic in the
DSS group, the mental state of mice in the Lp082 and SASP groups gradually

returned to normal, with active spirit, no arched back, no hematochezia and shiny hair
(Fig. S1 b). These results indicated that Lp082 intake could alleviate the symptoms of
depression, crouching and untidy hair of mice in the DSS group in the middle and late
stage of the experiment (Fig. S1 b).

Fig. S1

(b) Mental state of experimental mice.

c. Timing of experiments: After line 101, the sampling times of most experiments are
omitted or inadequately described.

**Response:** We appreciate your valuable and helpful comment. It is true that the
sampling times of most experiments are inadequately described. We have rewritten
this section. The rewritten content is more detailed, and the details are as follows:

After the experiment, the spleen, liver, kidney and colon of 8 mice were selected from
each group for observation and measurement. (Page 6, line: 170-172)

To further evaluate the effects of Lp082 on inflammatory cytokines in mice with
colitis, serum of 6 mice in each group was randomly collected after the experiment,
and the levels of pro-inflammatory cytokines TNF- α , IL-1 β , IFN- α , IL-6, MPO and
anti-inflammatory cytokines IL-10 were detected by ELISA kit. (Page 8, line:
208-213)

At the end of the experiment, the 1cm portion of the distal colon of 6 mice in each
group was selected randomly for HE staining, and histopathological score and
intestinal wall thickness were further measured ($n=6$). (Page 8, line: 220-224)

At the end of modeling (day 7 of the experiment), feces of 6 mice in each group were
randomly selected for metagenomic sequencing, and at the end of treatment (day 15
of the experiment), feces of 6 mice in each group were selected for metagenomic
sequencing, to observe the effects of DSS and Lp082 on the intestinal microecology
of mice. (Page 9, line: 258-262)

To prove the above findings, we further used gas chromatography-mass spectrometry
(GC-MS) to detect the content of SCFAs in cecal contents of 6 mice in each group.
(Page 11, line: 308-309)

At the end of the experiment, 6 mice from each group were randomly selected for
colon transcriptome sequencing, and the volcanic map was drawn based on the
preliminary gene distribution analysis results. (Page 13, line: 350-352)

C57BL/6J mice aged 7 weeks were randomly divided into 4 groups: control group
($n=8$), dextran sulfate sodium (DSS) group ($n=8$), lactobacillus plantarum HNU082
(Lp082) group ($n=8$), and salazosulfapyridine (SASP) group ($n=8$). (Page 23, line:
659-661)

After the mice were euthanized, the colon length of 8 mice in each group was
measured, the weight of spleen, liver, and kidney of 8 mice in each group was
measured. (Page 23, line: 677-679)

Before euthanasia, 6 mice were randomly selected from each group, and blood was
collected from the orbital venous plexus by a capillary tube. (Page 24, line: 696-697)

Finally, the levels of interleukin-1beta (IL-1 β), interleukin-6 (IL-6), interleukin-10
(IL-10), interleukin-17A (IL-17A), interferon-gamma (IFN- γ), Tumor necrosis
factor-alpha (TNF- α), and Myeloperoxidase (MPO) in the serum of 6 randomly
selected mice from each group were measured using the corresponding ELISA kits
(X-Y Biotechnology, Shanghai, China), as previously described. (Page 24, line:
686-687)

After euthanasia, the distal 1cm colons of 6 mice in each group were randomly
selected for HE staining section, histopathological score, and intestinal wall thickness
measurement. (Page 24, line: 697-688)

On the other hand, 8 mice were selected from each group, and their colonic tissues
were labeled with mucin 2 and ZO-1 antibodies, respectively [75], for further
immunofluorescence staining (Servicebio, Wuhan, China). (Page 25, line: 710-712)

Six mice were randomly selected at two time points (day 7 and day 15 of the
experiment) for metagenomic sequencing of feces. (Page 25, line: 728-7429)

At the end of the experiment, the cecal contents of 6 mice from each group were
randomly selected for SCFAs determination, and the specific steps were as follows:
(Page 26, line: 742-743)

At the end of the experiment, colon tissues of 6 mice from each group were randomly
selected for RNA sequencing. (Page 26, line: 757-758)

634 d. Sample sizes: Sample sizes and number of repeats are omitted. In most cases, the
635 specific datapoints in figures are not well described as to what they are measuring.

**Response:** We appreciate your valuable and helpful comment and we deeply agree
with the opinions of reviewer. According to your helpful suggestions, we have

carefully checked the whole paper, and added descriptions of sample size and number
of repeats in material and methods, legends and corresponding places in the article.
The changes have been highlighted in the text in yellow. The rewritten content is
more detailed, and the details are as follows:

After the experiment, the spleen, liver, kidney and colon of 8 mice were selected from
each group for observation and measurement. (Page 6, line: 170-172)

To further evaluate the effects of Lp082 on inflammatory cytokines in mice with
colitis, serum of 6 mice in each group was randomly collected after the experiment,
and the levels of pro-inflammatory cytokines TNF- α , IL-1 β , IFN- α , IL-6, MPO and
anti-inflammatory cytokines IL-10 were detected by ELISA kit. (Page 8, line:
208-213)

At the end of the experiment, the 1cm portion of the distal colon of 6 mice in each
group was selected randomly for HE staining, and histopathological score and
intestinal wall thickness were further measured ($n=6$). (Page 8, line: 220-224)

At the end of modeling (day 7 of the experiment), feces of 6 mice in each group were
randomly selected for metagenomic sequencing, and at the end of treatment (day 15
of the experiment), feces of 6 mice in each group were selected for metagenomic
sequencing, to observe the effects of DSS and Lp082 on the intestinal microecology
of mice. (Page 9, line: 258-262)

To prove the above findings, we further used gas chromatography-mass spectrometry
(GC-MS) to detect the content of SCFAs in cecal contents of 6 mice in each group.
(Page 11, line: 308-309)

At the end of the experiment, 6 mice from each group were randomly selected for
colon transcriptome sequencing, and the volcanic map was drawn based on the
preliminary gene distribution analysis results. (Page 13, line: 350-352)

C57BL/6J mice aged 7 weeks were randomly divided into 4 groups: control group
($n=8$), dextran sulfate sodium (DSS) group ($n=8$), lactobacillus plantarum HNU082
(Lp082) group ($n=8$), and salazosulfapyridine (SASP) group ($n=8$). (Page 23, line:
659-661)

After the mice were euthanized, the colon length of 8 mice in each group was
measured, the weight of spleen, liver, and kidney of 8 mice in each group was
measured. (Page 23, line: 677-679)

Before euthanasia, 6 mice were randomly selected from each group, and blood was
collected from the orbital venous plexus by a capillary tube. (Page 24, line: 696-697)

Finally, the levels of interleukin-1beta (IL-1 β), interleukin-6 (IL-6), interleukin-10
(IL-10), interleukin-17A (IL-17A), interferon-gamma (IFN- γ), Tumor necrosis
factor-alpha (TNF- α), and Myeloperoxidase (MPO) in the serum of 6 randomly
selected mice from each group were measured using the corresponding ELISA kits
(X-Y Biotechnology, Shanghai, China), as previously described. (Page 24, line:
686-687)

After euthanasia, the distal 1cm colons of 6 mice in each group were randomly
selected for HE staining section, histopathological score, and intestinal wall thickness
measurement. (Page 24, line: 697-688)

On the other hand, 8 mice were selected from each group, and their colonic tissues
were labeled with mucin 2 and ZO-1 antibodies, respectively [75], for further
immunofluorescence staining (Servicebio, Wuhan, China). (Page 25, line: 710-712)

Six mice were randomly selected at two time points (day 7 and day 15 of the
experiment) for metagenomic sequencing of feces. (Page 25, line: 728-7429)

At the end of the experiment, the cecal contents of 6 mice from each group were
randomly selected for SCFAs determination, and the specific steps were as follows:
(Page 26, line: 742-743)

At the end of the experiment, colon tissues of 6 mice from each group were randomly
selected for RNA sequencing. (Page 26, line: 757-758)

e. Statistics:

a. Only one statistical test is indicated in the paper, Wilcoxin signed rank test, line 546
in the methods. Adding the test run to each figure legend would be appropriate and
helpful.

**Response:** We appreciate your valuable and helpful comment. We have added
statistical test methods to each of the graphical legends. The revised content is as
follows:

Wilcoxon signed-rank test is used here. The significant difference was
considered at * $p < 0.05$, ** $p < 0.01$ and *** $p < 0.001$. Each group had at least 6
biological replicates.

b. Conditions statistical tests being used on are not obvious, in part due to the lack of
descriptions on sample sizes and replicates.

**Response:** Thank you for your comments. We deeply agree with the opinions of
reviewer and we have carefully checked the whole paper, and added descriptions of
sample size and replicates in material and methods, legends and corresponding places
in the article. The changes have been highlighted in the text in yellow. The details are
as follows:

After the experiment, the spleen, liver, kidney and colon of 8 mice were selected from
each group for observation and measurement. (Page 6, line: 170-172)

To further evaluate the effects of Lp082 on inflammatory cytokines in mice with
colitis, serum of 6 mice in each group was randomly collected after the experiment,
and the levels of pro-inflammatory cytokines TNF- α , IL-1 β , IFN- α , IL-6, MPO and
anti-inflammatory cytokines IL-10 were detected by ELISA kit. (Page 8, line:
208-213)

At the end of the experiment, the 1cm portion of the distal colon of 6 mice in each
group was selected randomly for HE staining, and histopathological score and
intestinal wall thickness were further measured ($n=6$). (Page 8, line: 220-224)

At the end of modeling (day 7 of the experiment), feces of 6 mice in each group were
randomly selected for metagenomic sequencing, and at the end of treatment (day 15
of the experiment), feces of 6 mice in each group were selected for metagenomic
sequencing, to observe the effects of DSS and Lp082 on the intestinal microecology
of mice. (Page 9, line: 258-262)

To prove the above findings, we further used gas chromatography-mass spectrometry
(GC-MS) to detect the content of SCFAs in cecal contents of 6 mice in each group.
(Page 11, line: 308-309)

At the end of the experiment, 6 mice from each group were randomly selected for
colon transcriptome sequencing, and the volcanic map was drawn based on the
preliminary gene distribution analysis results. (Page 13, line: 350-352)

C57BL/6J mice aged 7 weeks were randomly divided into 4 groups: control group
($n=8$), dextran sulfate sodium (DSS) group ($n=8$), lactobacillus plantarum HNU082
(Lp082) group ($n=8$), and salazosulfapyridine (SASP) group ($n=8$). (Page 23, line:
659-661)

After the mice were euthanized, the colon length of 8 mice in each group was

measured, the weight of spleen, liver, and kidney of 8 mice in each group was
measured. (Page 23, line: 677-679)

Before euthanasia, 6 mice were randomly selected from each group, and blood was
collected from the orbital venous plexus by a capillary tube. (Page 24, line: 696-697)

Finally, the levels of interleukin-1beta (IL-1 β), interleukin-6 (IL-6), interleukin-10
(IL-10), interleukin-17A (IL-17A), interferon-gamma (IFN- γ), Tumor necrosis
factor-alpha (TNF- α), and Myeloperoxidase (MPO) in the serum of 6 randomly
selected mice from each group were measured using the corresponding ELISA kits
(X-Y Biotechnology, Shanghai, China), as previously described. (Page 24, line:
686-687)

After euthanasia, the distal 1cm colons of 6 mice in each group were randomly
selected for HE staining section, histopathological score, and intestinal wall thickness
measurement. (Page 24, line: 697-688)

On the other hand, 8 mice were selected from each group, and their colonic tissues
were labeled with mucin 2 and ZO-1 antibodies, respectively [75], for further
immunofluorescence staining (Servicebio, Wuhan, China). (Page 25, line: 710-712)

Six mice were randomly selected at two time points (day 7 and day 15 of the
experiment) for metagenomic sequencing of feces. (Page 25, line: 728-7429)

At the end of the experiment, the cecal contents of 6 mice from each group were
randomly selected for SCFAs determination, and the specific steps were as follows:
(Page 26, line: 742-743)

At the end of the experiment, colon tissues of 6 mice from each group were randomly
selected for RNA sequencing. (Page 26, line: 757-758)

Minor points:

1. Missing information that should be addressed:

a. Rationale:

I. The introduction provides weak descriptions and evidence for use of a probiotic in
general to treat UC and Lp082 specifically. The introduction would benefit from
further elaboration on what is known about probiotic treatment of UC and indicate
what is or isn't known about Lp082 usage in UC specifically rather than using general
"probiotics" references. Along with this, lines 55-59 are confusing as written, but this
may be addressed when more information is added about those two points.

**Response:** We appreciate your valuable comment. According to your helpful
suggestions, we have rewritten this section to include more references detailing the
etiology of UC, current status of *Lactobacillus plantarum* in the treatment of UC and
the reasons for using Lp082. The revised content is as follows: (Page 3, line: 62-111)

Inflammatory bowel disease (IBD) is a chronic non-specific inflammatory
disease occurring in the gastrointestinal tract, mainly including ulcerative colitis (UC)
and crohn's disease (CD) [1]. The clinical manifestations of UC patients are diarrhea,
blood in the stool, weight loss, and diffuse inflammation of the colonic mucosa [2].
UC has become a major health problem worldwide due to its chronicity, recurrence,
and high morbidity [3], high risk of developing into Colorectal cancer (CRC) [4]. Due
to the disadvantages of traditional surgery and drug therapy of UC, such as
postoperative complications, side effects, and high cost [5], there is an urgent need to
develop a new UC treatment method.

There is no consensus on the specific pathogenesis of UC, and many evidences
suggest that the pathogenesis of UC is multifactorial, involving genetic susceptibility,
epithelial barrier defects, immune response disorders and environmental factors [6].

Differences in gut microbiota (type and amount) between colitis patient and healthy
people are thought to be one of the key factors in disease progression [7]. In UC
patients, the immune response is activated, the intestinal permeability is increased, the
intestinal mucosal barrier structure is destroyed, the homeostasis of gut microbiota is

disturbed, and the intestinal symbiotic bacteria are destroyed, thus activating a more
serious immune response, leading to the recurrence of the disease [8].

Due to the shortcomings of traditional treatments, it is urgent to develop new
treatments for UC, among which probiotics, as a substitute for antibiotics, have
attracted much attention for regulating gut microbiota to effectively alleviate UC
[9].As one of the main probiotics, *lactobacillus plantarum* has the characteristics of
regulating the balance of gut microbiota, increasing the adhesion of beneficial bacteria
to intestinal mucosa, inhibiting the adhesion of pathogenic bacteria and inhibiting the
inflammatory reaction [10]. Both animal [11] and clinical trials [12] have reported
that *lactobacillus plantarum* can reduce chronic mucosal inflammation in patients
with UC and prevent the occurrence of experimental colitis induced by DSS. In
addition, Bibiloni et al. evaluated the efficacy of *lactobacillus VSL#3* in 20 patients
with IBD and VSL#3 in newly diagnosed children with IBD and found that the
*lactobacillus strain* was effective in mild to moderate adult patients with IBD [13].
Yin et al. [14] believe that *lactobacillus plantarum* can restore the damaged mucosal
barrier function, regulate the imbalance of intestinal microbiota, inhibit pathogenic
bacteria, enhance intestinal system immunity, and have a good effect on relieving IBD
symptoms and maintaining remission. However, there are few studies on the specific
mechanism of action of *lactobacillus plantarum* in UC treatment, and there is no
unified argument [15].

The strain of *lactobacillus plantarum* HNU082 (Lp082) was originally isolated
from a traditional fermented food-fish tea of the Li people in Hainan Province, China
[16],which has a good safety profile and tolerance to acids and bile salts [17]. The
results of Lp082 whole genome sequencing showed showed that this bacterium has
great potential to develop as a probiotic in terms of physiology and function [5]In our
previous study, Lp082 not only can enhance the ecological and genetic stability of the
intestinal microbiota [18]. But also can inhibit the growth of *Fusobacterium*
*nucleatum* and reduce the inflammatory response [19]. Previous studies have also
shown that Lp082 exerts a preventive effect on hyperlipidemia through the
modulation of metabolism [20]. In addition, ingestion of Lp082 and supplementation

with prebiotics improved the stability of the intestinal microbiota and reduced the
occurrence of disorders associated with disease. These results invariably demonstrate
the probiotic potential of Lp082. However, the treatment effect of Lp082 on UC has
not been studied.

Therefore, we chose Lp082 to study the mechanism of probiotics in treating UC.

**Reference**

- 1. Ng SC, Shi HY, Hamidi N, Underwood FE, Tang W, Benchimol EI, et al. THE
WORLDWIDE INCIDENCE AND PREVALENCE OF INFLAMMATORY BOWEL
DISEASE IN THE 21ST CENTURY: A SYSTEMATIC REVIEW OF
POPULATION-BASED STUDIES. *Gastroenterology*. 2017;152(5):S970-S1; doi:
10.1016/s0016-5085(17)33292-4.
- 2. Bryant RV, Burger DC, Delo J, Walsh AJ, Thomas S, von Herbay A, et al. Beyond
endoscopic mucosal healing in UC: histological remission better predicts
corticosteroid use and hospitalisation over 6 years of follow-up. *Gut*.
2016;65(3):408-14; doi: 10.1136/gutjnl-2015-309598.
- 3. Shamoony M, Martin NM, O'Brien CL. Recent advances in gut Microbiota
mediated therapeutic targets in inflammatory bowel diseases: Emerging modalities for
future pharmacological implications. *Pharmacological Research*. 2019;148; doi:
10.1016/j.phrs.2019.104344.
- 4. Jess T, Rungoe C, Peyrin-Biroulet L. Risk of Colorectal Cancer in Patients With
UC: A Meta-analysis of Population-Based Cohort Studies. *Clinical Gastroenterology
and Hepatology*. 2012;10(6):639-45; doi: 10.1016/j.cgh.2012.01.010.
- 5. Marchesi JR, Adams DH, Fava F, Hermes GDA, Hirschfield GM, Hold G, et al.
The gut microbiota and host health: a new clinical frontier. *Gut*. 2016;65(2):330-9; doi:
10.1136/gutjnl-2015-309990.
- 6. Ungaro R, Mehandru S, Allen PB, Peyrin-Biroulet L, Colombel JF. UC. *Lancet*.
2017;389(10080):1756-70; doi: 10.1016/s0140-6736(16)32126-2.
- 7. Moayyedi P, Surette MG, Kim PT, Libertucci J, Wolfe M, Onischi C, et al. Fecal
Microbiota Transplantation Induces Remission in Patients With Active UC in a
Randomized Controlled Trial. *Gastroenterology*. 2015;149(1):102-+; doi:

10.1053/j.gastro.2015.04.001.

8. Costello SP, Hughes PA, Waters O, Bryant RV, Vincent AD, Blatchford P, et al.
Effect of Fecal Microbiota Transplantation on 8-Week Remission in Patients With UC
A Randomized Clinical Trial. *Jama-Journal of the American Medical Association*.
2019;321(2):156-64; doi: 10.1001/jama.2018.20046.

9. De Greef E, Vandenplas Y, Hauser B, Devreker T, Veereman-Wauters G.
Probiotics and IBD. *Acta Gastroenterol Belg*. 2013;76(1):15-9.

10. Pan Y, Ning Y, Hu J, Wang Z, Chen X, Zhao X. The Preventive Effect of
*lactobacillus plantarum* ZS62 on DSS-Induced IBD by Regulating Oxidative Stress
and the Immune Response. *Oxid Med Cell Longev*. 2021;2021:9416794; doi:
10.1155/2021/9416794.

11. Hasannejad-Bibalan M, Mojtahedi A, Eshaghi M, Rohani M, Pourshafie MR,
Talebi M. The effect of selected *Lactobacillus* strains on dextran sulfate
sodium-induced mouse colitis model. *Acta Microbiologica Et Immunologica*
*Hungarica*. 2020;67(2):138-42; doi: 10.1556/030.2020.00834.

12. Prantera C, Scribano ML, Falasco G, Andreoli A, Luzzi C. Ineffectiveness of
probiotics in preventing recurrence after curative resection for Crohn's disease: a
randomised controlled trial with *Lactobacillus* GG. *Gut*. 2002;51(3):405-9; doi:
10.1136/gut.51.3.405.

13. Bibiloni R, Fedorak RN, Tannock GW, Madsen KL, Gionchetti P, Campieri M, et
al. VSL#3 probiotic-mixture induces remission in patients with active UC. *The*
*American journal of gastroenterology*. 2005;100(7):1539-46; doi:
10.1111/j.1572-0241.2005.41794.x.

14. Yin MM, Yan XB, Weng WH, Yang YZ, Gao RY, Liu MF, et al. Micro Integral
Membrane Protein (MIMP), a Newly Discovered Anti-Inflammatory Protein of
*lactobacillus plantarum*, Enhances the Gut Barrier and Modulates Microbiota and
Inflammatory Cytokines. *Cellular Physiology and Biochemistry*. 2018;45(2):474-90;
doi: 10.1159/000487027.

15. Kostic AD, Xavier RJ, Gevers D. The Microbiome in Inflammatory Bowel
Disease: Current Status and the Future Ahead. *Gastroenterology*.

- 2014;146(6):1489-99; doi: 10.1053/j.gastro.2014.02.009.
- 16. Zhang J, Wang X, Huo D, Li W, Hu Q, Xu C, et al. Metagenomic approach
reveals microbial diversity and predictive microbial metabolic pathways in Yucha, a
traditional Li fermented food. *Scientific Reports*. 2016;6; doi: 10.1038/srep32524.
- 17. Huang S, Jiang S, Huo D, Allaband C, Estaki M, Cantu V, et al. Candidate
probiotic *Lactobacillus plantarum* HNU082 rapidly and convergently evolves within
human, mice, and zebrafish gut but differentially influences the resident microbiome.
*Microbiome*. 2021;9(1); doi: 10.1186/s40168-021-01102-0.
- 18. Ma C, Wasti S, Huang S, Zhang Z, Mishra R, Jiang S, et al. The gut microbiome
stability is altered by probiotic ingestion and improved by the continuous
supplementation of galactooligosaccharide. *Gut Microbes*. 2020;12(1); doi:
10.1080/19490976.2020.1785252.
- 19. Wang Y, Li J, Ma C, Jiang S, Li C, Zhang L, et al. *Lactobacillus plantarum*
HNU082 inhibited the growth of *Fusobacterium nucleatum* and alleviated the
inflammatory response introduced by *F. nucleatum* invasion. *Food & Function*.
2021;12(21):10728-40; doi: 10.1039/d1fo01388b.
- 20. Shao Y, Huo D, Peng Q, Pan Y, Jiang S, Liu B, et al. *Lactobacillus plantarum*
HNU082-derived improvements in the intestinal microbiome prevent the development
of hyperlipidaemia. *Food & Function*. 2017;8(12):4508-16; doi: 10.1039/c7fo00902j.

II. The intro (starting at line 62) provides weak background on data for SCFAs
alleviation of IBD. Citing work and adding text of SCFAs impact of IBD (preferably
UC and action through immune cells) would be helpful.

**Response:** We appreciate your valuable comment. According to your helpful
suggestions, we have rewritten this part and added more literature describing the
effects of SCFAs on UC and its effects on immune cells. The revised content is as
follows: (Page 4, line: 112-125)

*Lactobacillus* has been reported to have potential benefits for inflammatory
Bowel Disease (IBD) and colorectal cancer (CRC) symptoms due to its ability to
promote the formation of short-chain fatty acids (SCFAs) [1]. SCFAs are one of the

important metabolites of gut microbiota, and the main components in intestinal tract
are butyrate, acetate and propionate. Many studies have shown that SCFAs has
immunomodulatory effects [2], can reduce the expression of pro-inflammatory factors,
reduce inflammatory response, and play an important role in the treatment of UC [3].
Studies have shown that SCFAs can act on immune cells such as monocyte
macrophages and lymphocytes, change their gene expression, affect differentiation,
chemotaxis, proliferation and apoptosis, and thus participate in immune regulation [4].
In inflammatory response, SCFAs can reduce the expression of C5aR, thus regulating
the aggregation of macrophages and neutrophils [5], In addition, SCFAs can maintain
the integrity and permeability of intestinal epithelial cells, promote the secretion of
mucin in goblet cells, and protect the intestinal epithelial barrier so as to alleviate UC
[6].

**Reference**

- 1. Venegas DP, De la Fuente MK, Landskron G, Gonzalez MJ, Quera R, Dijkstra G,
et al. Short Chain Fatty Acids (SCFAs)-Mediated Gut Epithelial and Immune
Regulation and Its Relevance for Inflammatory Bowel Diseases. *Frontiers in*
*Immunology*. 2019;10; doi: 10.3389/fimmu.2019.00277.
- 2. Burger-van Paassen N, Vincent A, Puiman PJ, van der Sluis M, Bouma J, Boehm
G, et al. Regulation of the intestinal mucin MUC2 expression by short chain fatty
acids: implications for epithelial protection. *Faseb Journal*. 2009;23.
- 3. Rodriguez-Nogales A, Algieri F, Garrido-Mesa J, Vezza T, Pilar Utrilla M,
Chueca N, et al. Differential intestinal anti-inflammatory effects of *Lactobacillus*
*fermentum* and *Lactobacillus salivarius* in DSS mouse colitis: impact on microRNAs
expression and microbiota composition. *Molecular Nutrition & Food Research*.
2017;61(11); doi: 10.1002/mnfr.201700144.
- 4. Wang SL, Zhang SY, Huang SM, Wu ZH, Pang JM, Wu YJ, et al. Resistant
Maltodextrin Alleviates Dextran Sulfate Sodium-Induced Intestinal Inflammatory
Injury by Increasing Butyric Acid to Inhibit Proinflammatory Cytokine Levels.
*Biomed Research International*. 2020;2020; doi: 10.1155/2020/7694734.
- 5. Zhou YL, Xu HM, Xu J, Guo X, Zhao HL, Chen Y, et al. *F. prausnitzii* and its

supernatant increase SCFAs-producing bacteria to restore gut dysbiosis in
TNBS-induced colitis. *Amb Express*. 2021;11(1); doi: 10.1186/s13568-021-01197-6.
6. Hosseinkhani F, Heinken A, Thiele I, Lindenburg PW, Harms AC, Hankemeier T.
The contribution of gut bacterial metabolites in the human immune signaling pathway
of non-communicable diseases. *Gut Microbes*. 2021;13(1):1-22; doi:
10.1080/19490976.2021.1882927.

**b. Impact:**

I. Referencing lines 94-115: No text is provided to indicate what the alterations in
water intake, food intake, body weight, DAI, neurological responses, immune organ
index, spleen and colon color and structure, hyperemia, and feces structure mean in
the context of disease in DSS or in the Lp082 treated animals. This is not addressed
elsewhere in the paper and would help the reader understand the impact of your
results.

**Response:** Thank you for pointing this out. We have added the description according
to your suggestion. The revised content is as follows. (Page 6, line: 146-203)

People with UC have a disorder of colon function, poor absorption, loss of
appetite, weight loss, diarrhea, and bloody stools [1]. Therefore, the lower the body
weight, the lower the amount of water and food intake, and the higher the DAI score
(The scoring criteria is shown in **TABLE S1**), indicating the more severe enteritis.
Therefore, water intake, food intake, body weight, and DAI were monitored daily to
assess the severity of ulcerative enteritis modeling. "Molding ending" in **Fig. 1b** refers
to the end date of modeling UC with DSS on days 1-7, and no DSS water was
administered to mice beginning with day 8. The results showed that from 1 to 7 days,
the water intake, food intake, and body weight of the DSS group, the Lp082 group,
and the SASP group all showed a similar degree of gradual decrease, and these three
groups were all significantly different from the Control group on day 7 ($p < 0.05$),
which may be because these three groups were all under the same DSS modeling
conditions on days 0-7. Then on the 8th to 15th day, the water intake, food intake, and
body weight of the DSS group were still decreasing, but the water intake, food intake,

and body weight of Lp082 and SASP group gradually increased. Specifically, the
water and food intake of the Lp082 combined SASP group increased significantly
from day 9 ($p < 0.05$), and body weight increased significantly from day 12 ($p < 0.05$).
The DAI index of the DSS group, Lp082 group, and SASP group increased
significantly ($p < 0.05$) from the third day compared with the Control group. After
stopping DSS gavage on the 8th day, the DAI index of the DSS self-healing group
still increased, while that of the Lp082 group and SASP group gradually decreased
from the 10th day, and the degree of decrease in the Lp082 group was greater than
that in the SASP group (**Fig. 1b**).

In DSS-induced UC mice, the immune organ index gradually increased and the
colon length gradually shortened with increasing disease severity [2]. Therefore, we
measured the spleen, liver, kidney, and colon of the mice. The results showed that the
immune organ index of the DSS group was significantly increased ($p < 0.05$), and the
immune organ index was significantly decreased after Lp082 intake ($p < 0.05$) (**Fig.**
**1c**). The colon length of the mice in the DSS group was significantly decreased ($p <$
0.05), and the colon length in Lp082 group was significantly increased ($p < 0.05$) (**Fig.**
**1d**). In addition, we also observed that the intestinal contents of the colitis mice in the
DSS group were loose, unformed and there was blood in the intestinal lumen, while
the intestinal contents in the Lp082 and Control groups were clear particles, hard stool,
and no blood (**Fig. 1d**). The fecal morphology of the intestinal contents was similar to
the results observed in mouse feces on the buttocks of mice. The feces of the mice in
the DSS group were blood-red, and the feces were loose and unformed, while there
was no blood in the feces after Lp082 ingestion (**Fig. S1 a**).

With the increase of disease degree, DSS-induced UC mice will have a worse
mental state, even abdominal pain, arch back, panic and other symptoms [3]. The
mental state of the mice was observed daily, and the results are shown in **Fig. S1 b**.
On the 7th day of modeling, mice in the control group were in a normal state, with
normal urine and feces, shiny hair, active spirit, sensitive reaction, and increased body
size. However, mice in the BCD group had yellow and smelly urine, difficult
defecation, bloody stool, dark and fried hair, slow reaction and easy panic, arched

back, and reduced body size (**Fig. S1 b**). On the last day of treatment (Day 15),
compared with the arched back, retarded response, hematochezia, and lethargic in the
DSS group, the mental state of mice in the Lp082 and SASP groups gradually
returned to normal, with an active spirit, no arched back, no hematochezia and shiny
hair (**Fig. S1 b**). These results indicated that Lp082 intake could alleviate the
symptoms of depression, crouching, and untidy hair of mice in the DSS group in the
middle and late stage of the experiment (**Fig. S1 b**).

Studies have shown that under the condition of inflammation, the spleen of mice
induced by DSS will increase hyperemia and even appear infection blackening.
Therefore, we looked at the spleens of mice and found that the spleens of mice in the
DSS group were significantly larger and darker than those of mice in the normal
group. The spleens of mice in the Lp082 and SASP groups were smaller and redder
rather than black than those in the DSS group (**Fig. S1 c**).

**Reference**

- 1. Costello SP, Hughes PA, Waters O, Bryant RV, Vincent AD, Blatchford P, et al.
Effect of Fecal Microbiota Transplantation on 8-Week Remission in Patients With UC
A Randomized Clinical Trial. *Jama-Journal of the American Medical Association*.
2019;321(2):156-64; doi: 10.1001/jama.2018.20046.
- 2. Rodriguez-Nogales A, Algieri F, Garrido-Mesa J, Vezza T, Pilar Utrilla M,
Chueca N, et al. Differential intestinal anti-inflammatory effects of *Lactobacillus*
*fermentum* and *Lactobacillus salivarius* in DSS mouse colitis: impact on microRNAs
expression and microbiota composition. *Molecular Nutrition & Food Research*.
2017;61(11); doi: 10.1002/mnfr.201700144.
- 3. Sun MY, Liu YJ, Song YL, Gao Y, Zhao FJZ, Luo YH, et al. The ameliorative
effect of *lactobacillus plantarum*-12 on DSS-induced murine colitis. *Food & Function*.
2020;11(6):5205-22; doi: 10.1039/d0fo00007h.

II. Lines 125-145: Text here would benefit from at least a little description on what
this data means at this point in the writing. E.g. what does MUC2 loss and ZOI

abundance suggest about Lp082 effects?

**Response:** Thank you for pointing this out; we have added the description according
to your suggestion, and the revised content is as follows. (Page 9, line: 239-254)

MUC-2 is the mucin secreted by goblet cells, which can form the protective layer
of intestinal mucosa epithelium [1]. Tight junction protein ZO-1 is an important
physical barrier located in the gap between intestinal epithelial cells [2]. Studies have
shown that the content of ZO-1 and MUC-2 is reduced in UC, and its structure and
function are destroyed, resulting in increased intestinal permeability and harmful
substances entering the body, aggravating inflammation. Therefore, the levels of
MUC-2 and ZO-1 in colon were determined by immunofluorescence protein assay.
The results showed that the MUC-2 protein (green fluorescence) and ZO-1 protein
(red fluorescence) contents were higher in the control group, almost disappeared in
the DSS group, and significantly recovered in the Lp082 and SASP groups ($p < 0.05$),
and even increased more than SASP in Lp082 group (**Fig. 2d-e**). These results were
consistent with the surface density results of the two proteins (**Fig. 2f-g**). This
suggests that Lp082 can reduce the decrease in the number of ZO-1 and MUC-2
caused by DSS, and maintain the normal structure and function of the intestinal
mucus protein layer and intestinal epithelial cells.

**Reference**

- 1. Li XX, Wei B, Goodglick L, Wen T, Xia LJ, Braun J. Investigating Therapeutic
Approach of IBD Using Recombinant Glycoprotein Mucin2. *Faseb Journal*. 2009;23.
- 2. Pan Y, Ning Y, Hu J, Wang Z, Chen X, Zhao X. The Preventive Effect of
*lactobacillus plantarum* ZS62 on DSS-Induced IBD by Regulating Oxidative Stress
and the Immune Response. *Oxid Med Cell Longev*. 2021;2021:9416794; doi:
10.1155/2021/9416794.

c. Methods:

I. Scoring: Since understanding the scoring system used is important to understanding
the data, further describing the numbering and what that means would help the reader
understand the severity of the DSS model and the subsequent relief without looking

up the methods reference (either in the figure legends (Figure 1B) or in the methods
(see lines 480-481 where the modifications to DAI are not indicated). DAI and
immune organ index should be described at some level in the results and figure
legends as well so the reader knows what the data is describing without the methods.)

**Response:** Thank you for pointing this out. We deeply agree with the opinions of
reviewer. According to your helpful suggestions, we have added the description, and
the revised content is as follows.

the higher the DAI score (The scoring criteria is shown in **TABLE S1**), indicating
the more severe enteritis. (Page 6, line: 148-149)

The immune organ index (mg/g) of mouse spleen, liver, and kidney. The immune
organ index = immune organ weight (mg)/body weight (g). Increased coefficient of
immune organs indicates congestion and edema of organs and increased inflammation.
(Page 40, line: 1147-1150)

In DSS-induced UC, the higher the histopathological scores, the thicker the
intestinal mucosal wall, indicating more severe disease and more severe inflammation.
(Page 8, line: 222-224)

The following has been added to the supplementary materials:

**FIGURE LEGENDS**

**Fig. 1.** Effects of Lp082 on DSS-induced UC mice.

(b) Water intake, food intake, body weight, and disease activity index (DAI scoring
system modified from previous studies (Table. S1)) in mice.

(c) The immune organ index (mg/g) of mouse spleen, liver, and kidney. The immune
organ index = immune organ weight (mg)/body weight (g). Increased coefficient of
immune organs indicates congestion and edema of organs and increased inflammation.
(Page 40, line: 1143-1150)

**SUPPLEMENTARY TABLE LEGENDS**

**Table S1.**

Disease activity index (DAI) scoring system of dextran sodium sulfate-induced

colitis.

The DAI scoring system consists of three parts: weight loss, stool consistency and
visible blood in feces. Each part has 5 grades from 0 to 4. A score of 0 means that the
three indicators are normal, and the closer the score is to 4, the more serious
inflammation it is. (Page 7, line: 65-70)

**Table S2.**

Histopathology scoring system of dextran sodium sulfate-induced colitis. The
Histopathology scoring system scoring system was modified from previous studies [2].
The modified scoring system consists of six parts, namely, depth of Inflammation,
range of inflammation (%), crypt damage, goblet cell loss and the degree of
neutrophil Infiltration. Each component was rated on a scale of 0 to 4, a score of 0
means that the three indicators are normal, and the closer the score is to 4, the more
serious inflammation it is. (Page 9, line: 75-81)

II. How was "surface density" quantified? Line 144, figure 2F-G.

**Response:** Thank you for pointing this out; we have added the surface density
description according to your suggestion, and the revised content is as follows. (Page
25, line: 716-724)

The surface density of immunofluorescence ZO-1 and MUC-2 was measured and
calculated as follows: Eclipse CI-L fluorescence photography microscope was used to
select the target area of tissues for 200-fold imaging. After the imaging was completed,
image-Pro Plus 6.0 analysis software was used to convert green/red fluorescent
monochrome photos into black and white pictures, and then the same black was
selected as the unified standard to judge the positivity of all photos. The pixel area
was used as the standard unit. The positive cumulative optical DENSITY (IOD) and
the corresponding tissue pixel area in each section were measured, respectively, and
areal density =IOD/area was calculated.

III. Indicate the specific diet provided to the mice (line 459).

**Response:** Thank you for your comment. We added the description of the specific
diet of mice according to your suggestion, and the revised content is as follows. (Page
22, line: 645-650)

Mice in all groups were fed standard normal commercial mouse chow (It is
mainly composed of crude protein, crude fiber, crude fat and trace elements). Mice in
the Control group were free to drink normal water within 15 days, and the other three
groups were free to drink DSS water for the first 7 days, and were changed to normal
water from the 8th day.

IV. Elaborate on what you mean by "mouse colon samples" on line 537 for RNA-seq.

**Response:** Thank you so much for pointing this out, and so sorry we didn't make it
clear here. The mouse colon sample here refers to the middle 1 cm of the mouse colon
for transcriptome sequencing. Requires RNA extraction mini-kit (Qiagen, Hilden,
Germany) to extract total RNA from mouse colon samples for transcriptome
sequencing.

1164 d. Results structure:

I. The experiments, including the rationale, the samples, and the conditions, should be
described at some level prior to discussing the results in the Results text so the readers
know what the results are referencing.

**Response:** Thank you for your comment. We deeply agree with the opinions of
reviewer. At your wise suggestion, We have carefully reviewed the entire article and
added explanations of experimental principles, samples, and conditions at the
beginning of all Discussion and Results sections.

People with UC have a disorder of colon function, poor absorption, loss of
appetite, weight loss, diarrhea, and bloody stools [8]. Therefore, the lower the body
weight, the lower the amount of water and food intake, and the higher the DAI score
(The scoring criteria is shown in **TABLE S1**), indicating the more severe enteritis.
Therefore, water intake, food intake, body weight, and DAI were monitored daily to
assess the severity of ulcerative enteritis modeling. (Page 6, line: 146-151)

With the increase of disease degree, DSS-induced UC mice will have a worse
mental state, even abdominal pain, arch back, panic and other symptoms [30]. The
mental state of the mice was observed daily, and the results are shown in **Fig. S1 b**.
(Page 7, line: 184-186)

In DSS-induced UC mice, the immune organ index gradually increased and the
colon length gradually shortened with increasing disease severity [23]. Therefore,
after the experiment, the spleen, liver, kidney and colon of 8 mice were selected from
each group for observation and measurement. (Page 6, line: 169-172)

To further evaluate the effects of Lp082 on inflammatory cytokines in mice with
colitis, serum of 6 mice in each group was randomly collected after the experiment,
and the levels of pro-inflammatory cytokines TNF-, IL-1 β , IFN- α , IL-6, MPO, and
anti-inflammatory cytokines IL-10 were detected by ELISA kit. (Page 8, line:
208-213)

At the end of the experiment, the 1cm portion of the distal colon of 6 mice in
each group was randomly selected for HE staining, and histopathological score and
intestinal wall thickness were further measured ($n=6$). In DSS-induced UC, the higher
the histopathological scores, the thicker the intestinal mucosal wall, indicating more
severe disease and more severe inflammation. (Page 8, line: 220-224)

MUC-2 is the mucin secreted by goblet cells, which can form the protective layer
of intestinal mucosa epithelium [30]. Tight junction protein ZO-1 is an important
physical barrier located in the gap between intestinal epithelial cells [10]. Studies
have shown that the content of ZO-1 and MUC-2 is reduced in UC, and its structure
and function are destroyed, resulting in increased intestinal permeability and harmful
substances entering the body, aggravating inflammation. Therefore, the levels of
MUC-2 and ZO-1 in the colon were determined by immunofluorescence protein assay.
(Page 9, line: 239-246)

To further observe the effects of Lp082 on the gut microbiota of mice, we
sequenced the metagenome of feces of mice. At the end of modeling (day 7 of the
experiment), feces of 6 mice in each group were randomly selected for metagenomic
sequencing. At the end of treatment (day 15 of the experiment), feces of 6 mice in

each group were randomly selected for metagenomic sequencing, to observe the
effects of DSS and Lp082 on the intestinal microecology of mice. (Page 9, line:
257-262)

To prove the above findings, we further used gas chromatography-mass
spectrometry (GC-MS) to detect the content of SCFAs in cecal contents of 6 mice in
each group. (Page 11, line: 308-318)

At the end of the experiment, 6 mice from each group were randomly selected
for colon transcriptome sequencing, and the volcanic map was drawn based on the
preliminary gene distribution analysis results. (Page 13, line: 350-352)

II. Brief overall conclusions should be provided in the Results text to continue
engaging the reader and leading them along your thought process. This can be
partially addressed by moving text from the Discussion section to the Results. E.g.
lines 302-306 can be moved to the results section where diversity is discussed.

**Response:** We agree with the comment. According to your excellent suggestion, we
moved the Discussion lines 302-306 to the Results section, where we discuss diversity,
with a slight modification. The revised content is as follows: (Page 10, line: 282-284)

The above results show that Lp082 treatment remarkably increased the gut
microbiota diversity and reduced gut microbiota structural differences in gut
microbiota, as shown by the cluster analysis and PCoA analysis, also optimized
species composition.

e. Figures:

I. Figure 1:

Fig 1A - the arrows make it look like PBS only led to weight and colon assessment,
probiotics to immune indices, SASP to sequencing. Collapsing the arrows would
address this.

**Response:** Thanks for your nice comments. In the revised manuscript, we have
corrected the figure. The folded arrow has been added to **Fig. 1a**. Here, PBS refers to
phosphate buffered solution, which can provide a relatively stable ionic environment

and pH buffering capacity, and is a buffer salt solution commonly used in biology. **Fig.**
**1a** shows that on days 8-15, mice in Control group and DSS group were intragastric
with PBS solution, mice in the Lp082 group were intragastric with probiotics solution,
and mice in the SASP group were intragastric with SASP solution. The purpose of
such different gavage is to observe the effect of Lp082 on UC by comparing with DSS
self-healing and SASP positive drugs.

Fig 1B - what's being compared for the stats is not well described

**Response:** We really appreciate your efforts and comments on our manuscript. We
have revised our manuscript according to your comments and suggestions. The
statistical data in **Fig. 1b** are re-described, and the revised content is as follows: (Page
6, line: 153-168)

The results showed that from 1 to 7 days, the water intake, food intake, and body
weight of the DSS group, the Lp082 group and the SASP group all showed a similar
degree of gradual decrease, and these three groups were all significantly different
from the Control group on day 7 ($p < 0.05$), which because these three groups were all
under the same DSS modeling conditions on days 0-7. Then on the 8th to 15th day,
the water intake, food intake, and body weight of the DSS group were still decreasing,
but the water intake, food intake, and body weight of Lp082 and SASP group
gradually increased. Specifically, the water and food intake of the Lp082 in SASP
group increased significantly from day 8 ($p < 0.05$), and body weight increased
significantly from day 11 ($p < 0.05$). The DAI index of the DSS group, Lp082 group,
and SASP group increased significantly ($p < 0.05$) from the second day compared
with the Control group. After stopping DSS gavage on the seventh day, the DAI index
of the DSS self-healing group still increased, while that of the Lp082 group and SASP
group gradually decreased from the 9th day, and the degree of decrease in the Lp082
group was greater than that in the SASP group. (**Fig. 1b**)

Fig 1C - the bars for stats are shifted (also make sure the lines are the same point
thickness for stats in each figure)

**Response:** Thanks for your helpful comments. We are very sorry for our negligence
and we have corrected **Fig. 1c** according to your helpful suggestion. We have checked
all the pictures carefully to make sure we don't have the same problem again.

Fig 1B - "molding ending" is not described in the text. Rephrase or define. Also
decrease the numbers in the X axis as they are too condensed. The title "duration of
probiotic intervention (day)" is an incorrect title as this figure shows duration of the
entire experiment, including pre-treatment with DSS before probiotics.

**Response:** Thank you for your helpful comment. We deeply agree with your
suggestion and we have made correction according to your nice suggestions.
"Molding ending" in **Fig. 1b** refers to the end date of modeling UC with DSS on days
1-7, no DSS water was administered to mice beginning with day 8. We have added the
description of "molding ending" in both the figure legend and the results section,
reduced the number on the X axis, and changed the "duration of probiotic intervention
(day)" to the duration of the entire experiment "Days" based on your good idea.

Fig 1E - there's no Y-axis label and the datapoints are not described

**Response:** Thank you for your helpful comment. We are very sorry for our
negligence and we have modified the figure according to your suggestion. The
changes have been highlighted in yellow in the text.

II. Figure 2:

Fig 2A - you might try to line up your red boxes better so they better represent the
blow ups (and make straighter red lines).

**Response:** Thank you for your helpful comment. We deeply agree with your
suggestion and we have made correction according to your nice suggestions.

Fig 2B - add microscopy information for the antibody stains in the legend and/or the
methods section. Although the staining method cites another paper, it's best to include
antibody information in the methods section. MUC2, ZO-1, and the blue marker are

not labeled in the figure and in the figure legend.

**Response:** Thank you for your helpful comment. We agree with your suggestion, and
we have added the description in the legend and method section according to your
suggestion. The details of the modification are as follows:

On the other hand, 8 mice were selected from each group, and their colonic tissues
were labeled with mucin 2 and ZO-1 antibodies, respectively [75], for further
immunofluorescence staining (Servicebio, Wuhan, China). Fluorescein is linked to the
antibodies ZO-1 and MUC-2 to form fluorescent antibodies. By specifically binding
to the antigen to form a multi-component complex, ZO-1 and MUC-2 can be
characterized and localized in the intestinal tissue by means of a fluorescence
microscope research. (Page 25, line: 710-716)

**FIGURE LEGENDS**

Fig. 2. Effects of Lp082 on histological parameters and immunofluorescent proteins.
(d) Immunofluorescence staining of MUC-2 (green fluorescence). Scale bar = 100 μ m.
Blue marker is the color of the negative of the photograph (colon tissue without
antigenic markers)
(e) Immunofluorescence staining of ZO-1 (red fluorescence). Scale bar = 100 μ m.
Blue marker is the color of the negative of the photograph (colon tissue without
antigenic markers) (Page 40, line: 1164-1169)

Fig 2C - the y axis is missing a metric

**Response:** Thank you very much for your reminder. We are very sorry for our
negligence of metric. **Fig. 2c**-Y axis refers to the thickness of the intestinal mucosal
wall, and its measurement method has been added to the material method section. We
have carefully checked the full text and have highlighted the changes in yellow. The
details are as follows. (Page 24, line: 706-709)

The thickness of the intestinal mucosal wall was measured in the following ways:
Image-Pro Plus 6.0 analysis software was used to measure the thickness of the
mucosal layer at 5 positions of each layer (first from the right) in a unified mm
standard unit, and the average value was calculated.

Fig 2f-g - the y axes are missing metrics (as noted above, the method to define these
numbers is not stated).

**Response:** Thank you for your helpful comment. We are very sorry for our
negligence of metric. **Fig. 2c**-Y axis refers to the areal density of MUC-2 and ZO-1,
and its measurement method has been added to the material method section. We have
carefully checked the full text and have highlighted the changes in yellow. The details
are as follows. (Page 25, line: 716-724)

The surface density of immunofluorescence ZO-1 and MUC-2 was measured and
calculated as follows: Eclipse CI-L fluorescence photography microscope was used to
select the target area of tissues for 200-fold imaging. After the imaging was completed,
image-Pro Plus 6.0 analysis software was used to convert green/red fluorescent
monochrome photos into black and white pictures, and then the same black was
selected as the unified standard to judge the positivity of all photos. The pixel area
was used as the standard unit. The positive cumulative optical DENSITY (IOD) and
the corresponding tissue pixel area in each section were measured, respectively, and
areal density =IOD/area was calculated.

III. Figure 3:

Fig 3A-C groupings not labeled as indicated above

**Response:** Thank you for your comment. We are grateful for your reminder. To be
more clear and in accordance with the reviewer's concerns,, we have added **Fig. S3**
to explain the groupings in **Fig 3a-3c**. We also supplemented the description of this part
in the supplementary material. The revised content is highlighted in yellow. The
specific content is as follows. (Page 3, line: 22-33)

**SUPPLEMENTARY FIGURE LEGENDS**

**Fig.S3**

(a) Timing and grouping of mouse metagenomic sequencing

M means the modeling period, T means the treatment period. Respectively, A, B, C

and D group mean 7 days normal water (ultrapure water), DSS, Lp082 and SASP
treatment after 7 days DSS gavage.

M-A means A group represents the control group on the 7th day of DSS modeling,
M-B represents the DSS group on the 7th day of DSS modeling, M-C represents the
Lp082 group on the 7th day of DSS modeling, M-D represents the SASP on the 7th
1362 day of DSS treatment Group.

T-A means treating-A group represents the control group at the end of the treatment,
T-B represents the DSS group at the end of the treatment, T-C represents the Lp082
group at the end of the treatment, and T-D represents the SASP group at the end of the
treatment.

Fig 3D - The meaning of the red highlighting is not indicated in the figure legend. No
information is provided about the tree, including what it represents and what the
colors indicate. The heat map values are not described - what is being compared and
what does a value of zero mean?

**Response:** Thank you for your helpful comment and your remind, we have
supplemented the description of the figure in the legend and all revisions have been
highlighted, and the revised content is as follows. (Page 41, line: 1181-1193)

FIGURE LEGENDS

**Fig. 3.** Effects of Lp082 strains on the gut microbiota in mice.

(d)The red highlight in the **Fig. 3d** refers to the significantly increased bacteria that
can produce SCFAs in the Lp082 group. The tree in the **Fig. 3d** represents the
phylogenetic tree, which is obtained by clustering the abundance of each color block
based on the unifracs distance after taking $\log_2(x*100)$ for the relative abundance at
the species level. The clustering does not reflect any evolutionary relationship. It
shows the abundance of bacterial species in the sample. 0 has no special meaning in it
(it is only used to facilitate the differentiation of overall abundance). The darker the
yellow in the color block in the Fig. 3d (the value closer to 2), the higher the relative
abundance. Darker blue (values closer to -2) indicate lower relative abundance.

IV. Figure 4:

Fig. 4A - It is not entirely clear where this data comes from. My assumption was the
metagenome, but the Acetic acid sub section has me unsure. Describe this figure more,
taking care to describe what the acetic acid subsection is evaluating.

**Response:** Thank you for your helpful comment. We are sorry to have failed to make
it clear and are very sorry about the inconvenience caused. According to your helpful
suggestions, we re-describe **Fig. 4** and the rewritten content is as follows: (Page 10,
line: 286-346)

**The regulatory role of SCFAs**

Next, we conducted a correlation analysis between Lp082 (*Lactobacillus*
*plantarum*) and SCFAs, and found that Lp082 (*Lactobacillus plantarum*) was strongly
positively correlated with SCFAs (acetic acid, propionic acid, butyric acid) (**Fig. 4c**),
the correlation results suggested that Lp082 can increase the content of SCFAs. The
above results inspired us to further explore the relationship between Lp082 and
SCFAs, and we further analyzed the bacterial species and metabolic pathways
associated with SCFAs. Further metagenomic data provided support for our above
speculation. Combined with metagenomic data, the species composition of mice gut
microbiota was further analyzed. The results showed that the relative abundance of
some special bacteria increased in the Lp082 group, such as, *Lactobacillus plantarum*,
*Bifidobacterium pseudolongum*, *Akkermansia muciniphila*, *Bacteroides ovatus*,
*Parabacteroides distasonis*, *Lactobacillus reuteri*, *Anaerotruncus sp G3 2012* (these
bacteria are highlighted in red in **Fig. 3d**), all of which can metabolize produces the
SCFAs [1].

Subsequently, we further analyzed the metabolic pathways of gut microbiota in
mice. Results of differential metabolic pathways showed that the abundance of gut
microbiota metabolic pathways related SCFAs production decreased in DSS group but
increased in Lp082 group (**Fig. 4a**). We infer that Lp082 can promote the content of
SCFAs (acetate, propionate and butyrate) by adjust three metabolic pathways,
including Pyruvate fermentation to Propanoate I, Pyruvate fermentation to acetate and

lactate II, Acetyl CoA fermentation to Butanoate (**Fig. 4a**).

To prove the above findings, we further used gas chromatography-mass
spectrometry (GC-MS) to detect the content of SCFAs. Compared with control group,
the contents of butyric acid, valeric acid, acetic acid, propionic acid and isobutyric
acid were significantly decreased after ingestion of DSS ($P < 0.01$). Compared with
DSS group, the contents of butyric acid, acetic acid, propionic acid and isobutyric
acid were extremely significant increased after ingestion of Lp082 ($P < 0.01$). This
confirmed our previous hypothesis based on the correlation that Lp082 intake would
increase SCFAs levels (**Fig. 4b**). Based on the above results, we speculate that Lp082
increase the content of SCFAs by affecting the abundance of SCFAs-producing
microbes, as well as the metabolic pathways of SCFAs-producing microbes.

To further understand the role of SCFAs, we performed a Pearson correlation
analysis. The results showed that *helicobacter hepatica*, which was significantly
increased in the DSS group, was strongly negatively correlated with acetic acid,
propionic acid, and butyric acid (**Fig. 4c**). *lactobacillus plantarum*, *Bifidobacterium*
*pseudolongum*, *Akkermansia muciniphila*, *Parabacteroides distasonis*, *Lactobacillus*
*reuteri*, which were significantly increased in Lp082 group showed strong positive
correlation with acetic acid, propionic acid, and butyric acid. *Anaerotruncus sp G3*
*2012* and *Bacteroides ovatus* showed a strong positive correlation with butyric acid
and acetic acid, and a weak positive correlation with propionic acid (**Fig. 4c**). These
SCFAs including acetic acid, propionic acid, and butyric acid were all strong
negatively correlation with the pro-inflammatory factors TNF- α , IL-1 β , IFN- γ , IL-6,
MPO but strongly positively correlated with the inflammatory suppressor IL-10 (**Fig.**
**4d**). As important products of gut microbiota metabolism, SCFAs have certain
anti-inflammatory effects and play an important role in maintaining normal intestinal
morphology and function. Combined with the results of **Fig. 3d**, **Fig. 4a-4d**, as well
as the improvement of physiological indicators (**Fig. 1b-1d**), pathological indicators
(**Fig. 2a-2g**) and inflammatory factors (**Fig. 1e**) after ingestion of Lp082, we
speculated that Lp082 may alleviate DSS-induced UC by regulating SCFAs through
the following mechanisms (**Fig. S4**). That is, after the ingestion of Lp082, the

abundance of the intestinal microbes of SCFAs-producing increased, which promoted
the content of SCFAs. The SCFAs has the function of promoting the secretion of
inflammatory cytokine and suppressing the secretion of inflammatory factors. The
changes in inflammatory cytokines affect the physiological indicators of mice, which
increases the weight, colon length, drinking water and eating volume of mice, and
reduces the DAI score and immune organs index. The changes in inflammatory
cytokines also affected the pathological indexes of mice, resulting in a decrease in
histopathological score and an increase in immunofluorescence protein content of
ZO-1 and MUC-2.

**Reference**

1. Y. W. Cheng, J. M. Liu and Z. X. Ling, Short-chain fatty acids-producing
probiotics: A novel source of psychobiotics, Critical Reviews in Food Science and
Nutrition, DOI: 10.1080/10408398.2021.1920884.

Fig. 4C-D - A description of the tree components is missing. Describe the correlation
analysis more in the text and figure legend.

**Response:** Thank you for your helpful comment and your reminder. We are sorry to
have failed to describe it clearly and are very sorry about the inconvenience caused.
According to your helpful suggestions, we have supplemented the description of the
figure in the legend, and all revisions have been highlighted, and the revised content is
as follows:

The following sections have been added to the legend: (Page 41, line: 1198-1241)

**FIGURE LEGENDS**

**Fig. 4.**

(c)Relationship between SCFAs and gut microbiota. The tree in the **Fig. 4c** represents
the phylogenetic tree, which is obtained by clustering the data. This clustering does
not reflect any evolutionary relationships but rather shows the abundance of the
samples. **Fig. 4c** is a correlation heat map drawn by Pearson correlation analysis
based on bacterial abundance and SCFAs abundance. The correlation range is from -1
to +1. The closer to -1 or +1, the stronger the correlation between bacterial species

and SCFAs. 0 means no correlation, a negative value means negative correlation, and
a positive value means positive correlation.

(d) Relationship between SCFAs and inflammatory cytokines. The tree in the **Fig. 4d**
represents the phylogenetic tree, which is obtained by clustering the data. This
clustering does not reflect any evolutionary relationships but rather shows the
abundance of the samples. **Fig. 4d** is a correlation heat map drawn by Pearson
correlation analysis based on the content of inflammatory cytokines and the
abundance of SCFAs. The horizontal axis in the **Fig. 4d** is the clustering based on the
abundance of SCFAs, and the vertical axis is based on the abundance of inflammatory
cytokines. 0 means no correlation, a negative value means negative correlation, and a
positive value means positive correlation.

The following sections have been added to the manuscript: (Page 12, line: 319-330)

To further understand the role of SCFAs, we performed a Pearson correlation
analysis. The results showed that helicobacter hepatica, which was significantly
increased in the DSS group, was strongly negatively correlated with acetic acid,
propionic acid, and butyric acid (**Fig. 4c**). lactobacillus plantarum, Bifidobacterium
pseudolongum, Akkermansia muciniphila, Parabacteroides distasonis, Lactobacillus
reuteri ,which were significantly increased in Lp082 group showed strong positive
correlation with acetic acid, propionic acid, and butyric acid. Anaerotruncus sp G3
2012 and Bacteroides ovatus showed a strong positive correlation with butyric acid
and acetic acid, and a weak positive correlation with propionic acid (**Fig. 4c**). These
SCFAs including acetic acid, propionic acid, and butyric acid were all strong
negatively correlation with the pro-inflammatory factors TNF- α , IL-1 β , IFN- γ , IL-6,
MPO but strongly positively correlated with the inflammatory suppressor IL-10 (**Fig.**
**4d**).

1502 V. Figure 5: I think this entire figure would be best placed in the supplement as it's
really just a sub-point of the contents of figure 6 (but it won't fit in figure 6). You
might also remove "distribution" from the title and legend as this suggests tissue
spatial information but is not needed.

**Response:** Thank you for your helpful comment. We agree with the suggestions of the
reviewer. To be more clear and in accordance with the reviewer's concerns, we
re-described **Fig. 4a** and **Fig. 4b** and have put the entire figure of **Fig. 5** in the
supplement according to your suggestion and named it **Fig. S5**. The revised content
has been highlighted in yellow.

VI. Figure 6: Overall, the less color you use, the clearer this figure will be.

**Response:** Thank you for your comment. We will take this into account in future
drawings. We are grateful for the suggestion. As suggested by the reviewer, we have
made some adjustments to the graphics. We have been deeply aware of this problem,
and we will also pay attention to reducing the use of colors in future drawings. Thank
you again for your help.

Fig 6A-C: I recommend condensing as Fig 6A. Describe what gene ratio is in the
figure legend.

**Response:** Thank you for your comment. We agree with your suggestion. According
to your helpful suggestions, we have renamed **Fig. 6a-6c** to **Fig. 6a** and have added a
description of the gene ratio in the legend. All revisions have been highlighted, and
the revised content is as follows: (Page 42, line: 1223-1224)

Gene Ratio: Ratio of the number of genes related to this Term to the total number of
genes

Fig 6D-F: I recommend condensing as Fig 6B.

**Response:** Thank you for your comment. We agree with your suggestion. According
to your helpful suggestions, we have renamed **Fig. 6d-6f** to **Fig. 6b**.

Fig 6G-J: I recommend condensing as Fig 6C. I and j legends are swapped. Describe
ifcSE in the legend.

**Response:** Thank you for your comment. We agree with your suggestion. According
to your helpful suggestions, we have renamed **Fig. 6g-6j** to **Fig. 6c**. and have

supplemented the description of the figure in the legend, all revisions have been
highlighted, and the revised content is as follows: (Page 43, line: 1233-1236)

The IfcSE is the standard error, which is the value obtained from the standard
deviation (SD) of the sample divided by the square root of the previous sample size.
The smaller the standard error is, the smaller the difference between sample mean and
population mean is.

2. The authors confuse whether they are studying Lp082 prevention or treatment of
colitis by using verbiage referring to "prevention" and "treatment" interchangeably.
This makes it difficult to track what the authors are trying to accomplish (for example,
line 60 says "relieving", lines 76 and 87-88 say "prevention"). Because the authors
state that the colitis inducer (DSS) is administered at the time of treatment (Lp082) in
the beginning of the Results to evaluate prevention (line 87), but Figure 1A shows that
Lp082 is being added at day 8 (so not at the time of induction), I cannot assess which
is being studied: Lp082 1) treatment or 2) prevention of UC. My best assumption is
that the methods section is correct, and the methods says that DSS is used prior to
addition of Lp082, and thus the authors are studying Lp082 relief of colitis. Thus, the
language in the paper should be altered to indicate that Lp082 was administered after
DSS induced colitis and observed effects are Lp082 alleviation of symptoms, not
prevention of symptoms.

**Response:** We appreciate your valuable and helpful comment. We apologize for the
language problems in the original manuscript. We sincerely apologize for the
confusion caused to you. We used DSS to establish a model of UC and then treated it
with Lp082. We have carefully checked the wording of the full text and corrected the
preventive effect to the therapeutic effect. Thank you very much for pointing this out.
It was very helpful. The changes have been highlighted in yellow in the article. And
the language presentation was improved with assistance from a native English speaker
with appropriate research background.

3. The abstract, discussion section, and figure 7B describe the effect of Lp082 on the
animal model through the groups: biological barrier, chemical barrier, mechanical
barrier, and immune barrier. I don't recommend subdividing "biological, chemical,
and mechanical barrier", as everything you are referring to is biological, chemical,
and mechanical in nature. Rather, use categories akin to "microbiota/microbiome
alterations, barrier function improvements, and inflammation reduction."

**Response:** We appreciate your valuable and helpful comment. You have provided an
excellent suggestion. Thank you for pointing out this problem. We agree with your
views on this issue. Following your suggestion, the discussion of these four intestinal
barriers has been rewritten in the discussion section, but we think it is reasonable to
describe it in terms of these four barriers. The pathogenesis of UC is the result of the
combined effect of genetically susceptible hosts and the environment, and its common
pathological outcome is the damage of the structure and function of the intestinal
mucosal barrier. The intestinal mucosal barrier is damaged, resulting in an increase in
the permeability of the intestinal epithelial barrier, and further stimulation of intestinal
contents, bacteria, and toxins promotes the immune response to intestinal
inflammation. The normal intestinal mucosal barrier consists of mechanical barrier,
chemical barrier, immune barrier, and biological barrier. The chemical barrier refers to
the glue-like mucin layer covering the surface of intestinal epithelial cells, which is
mainly composed of MUC-2 secreted by goblet cells, digestive juices, and
bacteriostatic substances produced by normal parasitic bacteria in the intestinal lumen
[1]. The mechanical barrier is the most important part of the intestinal mucosal barrier.
Its structural basis is the intestinal mucosal epithelial cells and the tight junctions (TJ)
between the epithelial cells [2]. The immune barrier is associated with immune cells,
and inflammatory factors [3]. The biological barrier is a normal intestinal colony of
bacteria that is resistant to colonization by foreign strains [4]. The results of the study
found that Lp082 can improve the intestinal mucosal barrier by synergistically
optimizing the biological barrier, chemical barrier, mechanical barrier and immune
barrier, thereby alleviating UC. Specifically, We found that Lp082 rebuilt the

biological barrier by regulating the intestinal microbiome and increasing the SCFAs.
Lp082 improved the chemical barrier by reducing ICAM-1, VCAM, and increasing
goblet cells and mucin2. Lp082 ameliorated the mechanical barrier by increasing the
ZO-1, ZO-2, and occludin and decreasing claudin-1 and claudin-2. Lp082 optimized
the immune barrier by reducing the content of IL-1 β , IL-6, TNF- α , MPO, IFN- γ and
increasing the IL-10, TGF- β 1, and TGF- β 2. In conclusion, we believe that it is
reasonable to use these four barriers to discuss the effect of Lp082 on DSS induced
UC. Maybe we didn't describe it very well, so we rewrote a discussion section that
explained the four barriers in more detail, with the following changes. (Page 17, line:
496-637)

Lp082 improved chemical barrier

The chemical barrier refers to the glue-like mucin layer covering the surface of
intestinal epithelial cells, which is mainly composed of MUC-2 secreted by goblet
cells, digestive juices and bacteriostatic substances produced by normal parasitic
bacteria in the intestinal lumen [1]. The chemical barrier plays an important role in
isolating the internal and external environment of the intestinal tract, lubricating the
intestinal mucosa, and inhibiting the entry of harmful substances in the intestinal
lumen [5]. The intestinal mucosal wall thickness was significantly increased in the
DSS group, whereas it was significantly decreased after Lp082 ingestion (**Fig. 2c**). In
DSS-induced UC, the thicker the intestinal mucosal wall, indicating more severe
inflammation. In addition, the H&E staining result showed that the number of goblet
cells decreased in the DSS group (red arrow), whereas the number of goblet cells
increased (yellow arrow) after Lp082 ingestion (**Fig. 2a**). The immunofluorescent
protein content of MUC-2, which is mainly secreted by goblet cells, was significantly
decreased in the DSS group (**Fig. 2d**), and the areal density of MUC-2 (**Fig. 2f**) and
the mRNA expression of MUC-2 were also significantly decreased in the DSS group
(**Fig. 5c**), while the immunofluorescence protein content, areal density and mRNA
expression of MUC-2 all increased in the Lp082 group,
Sun et al. [6] observed the same phenomenon that *lactobacillus plantarum 12* can
repair the intestinal mucosal chemical barrier by increasing the content of MUC-2.

Burger-van Paassen et al. [7] found that intake of SCFAS could increase the
expression abundance of MUC-2 mRNA in cells. The mRNA expressions of ICAM-1
and VCAM-1 were significantly decreased after Lp082 intake. Taniguchi et al. [8]
found that anti-ICAM-1 treatment significantly attenuated colonic mucosal damage,
while Philpott et al. [9] found that adhesion molecules ICAM-1 & VCAM-1 induced
intestinal mucosal lesions. Lp082 has been shown to be effective in relieving
intestinal mucosal lesions (i.e., reduced ulceration and inflammatory cell infiltration
caused by DSS). So, we speculate that Lp082 reduces mucosal lesions by reducing
ICAM-1 and VCAM. The above results showed that probiotic Lp082 increased the
MUC-2 content in the mucus layer by restoring the number of goblet cells, relieved
the intestinal mucosal damage caused by ICAM-1 and VCAM-1, so as to repaired the
chemical barrier.

Lp082 improved mechanical barrier

The mechanical barrier is the most important part of the intestinal mucosal
barrier. Its structural basis is the intestinal mucosal epithelial cells and the tight
junctions (TJ) between the epithelial cells [2]. The mechanical barrier can effectively
prevent harmful substances such as bacteria and endotoxins from entering the blood
through the intestinal mucosa.iers The aberrant structure of tight junction (TJ)
proteins between intestinal epithelial cells, such as the reduction of ZO-1, ZO-2, and
occludin, is one of the critical factors leading to the disruption of the gut mechanical
barrier in UC patients [10]. Several studies have identified TJ protein as a new target
for the current treatment of UC [11]. Because Lp082 excellently improved
histopathology, we speculated that Lp082 also has a regulatory effect on TJ molecules.
To this end, we analyzed major TJ proteins, including ZO-1, ZO-2, occludin. As
expected, the mRNA expression and immunofluorescence protein content of ZO-1
and the mRNA expression of ZO-2 and occludin were significantly decreased in
DSS-induced UC mice but improved in the Lp082 treatment group. These are
consistent with the findings of Cordeiro et al. [12] that ZO-1 and ZO-2 were
significantly decreased in UC but increased after probiotic Minas Frescal cheese
intake, indicating that the improvement of the mechanical barrier by regulating TJ

may be one of the mechanisms by which probiotic Lp082 exerts anti-UC. In addition,
the mRNA expression of another particular tight junction protein, ICAM-1 and
VCAM-1, was increased in the DSS group. It is consistent with the findings of
elevated ICAM-1 and VCAM-1 in IBD patients in clinical studies [13]. Mitselou et al.
[14] found that the adhesion molecules ICAM-1 and VCAM-1 induced intestinal
mucosal injury. Taniguchi et al. [8] found that anti-ICAM-1 treatment attenuated
colonic mucosal injury. It has been reported that ICAM-1 and VCAM-1 can increase
the permeability of intestinal mucosa [15]. Interestingly, the mRNA expression of
ICAM-1 and VCAM-1 was found to decrease after Lp082 ingestion. Therefore, it can
be thought that the alleviation of UC by Lp082 may be due to down-regulation of
ICAM-1, VCAM-1 and increase protein quantity and mRNA expression of
ZO-1, ZO-2, so as to reduce intestinal mucosal permeability, thereby inhibiting the
entry of harmful bacteria and undigested food and toxins into the body and reducing
inflammation. These results suggest that Lp082 repairs the intestinal mechanical
barrier by regulating TJ.

Lp082 improved the immune barrier

Although the exact etiology of UC is complex and uncertain, studies suggest that
the NF- κ B pathway plays a vital role in the pathogenesis of UC [3]. Our study has
proved that Lp082 inhibits the NF- κ B pathway by down-regulating the mRNA
expression of NF- κ B2, NF- κ B1, COX-2, RelA, Toll4, iNOS, and that NF- κ B can also
regulate inflammation by regulating cytokines [16]. Therefore, it can be suggested
that Lp082 also has a specific regulatory effect on cytokines. To confirm this, we
analyzed the cytokines associated with NF- κ B. As expected, we observed that the
mRNA expression level content of pro-inflammatory cytokines (TNF- α , IL-1 β , and
IL-6) were significantly increased in the DSS group but significantly decreased in the
Lp082 group, It is interesting to note that the protein levels of TNF- α , IL-1 β , and IL-6
detected by elisa kit were also increased in the DSS group and decreased after Lp082
intake. Among them, TNF- α can promote the proliferation and differentiation of T
cells and increase intestinal inflammation [17]. The upregulation of IL-1 β is involved
in the recruitment and retention of leukocytes in inflamed tissues and can activate

innate immune lymphocytes [18]. IL-6 activates NF- κ B to regulate the dextran sulfate
sodium-induced colitis in mice [19]. The above results indicate that Lp082 alleviates
UC by inhibiting the levels of pro-inflammatory factors (TNF- α , IL-1 β , and IL-6).
Interestingly, we also found that the mRNA expressions of anti-inflammatory
cytokines IL10, TGF-1, and TGF-2 were significantly decreased in the DSS group but
increased in the Lp082 group. Il-10 protein levels measured by elisa kit also decreased
in the DSS group and increased in the Lp082 group. Surprisingly, IL10, TGF-1, and
TGF-2 were shown to activate Treg and anti-inflammatory macrophages to alleviate
UC [20]. And Sato et al. [21] also found that the loss of IL-10 spontaneously gave rise
to IBD, and Hume et al. [22] found that TGF- β 1 and TGF- β 2 could dramatically
relieve intestinal inflammation in DSS-induced colitis mice. These results suggest that
Lp082 alleviates UC by increasing the levels of anti-inflammatory factors IL10,
TGF-1, and TGF-2. We further analyzed the specific regulatory effects of Lp082 on
intestinal mucosal immunity. In addition to inflammatory factors, we also noticed that
a heme protein, MPO, was significantly reduced in the Lp082 group. Trevisin et al.
[23] found that MPO caused UC by producing cytokines and hypochlorite and that
MPO in the colon of UC patients is mainly produced by neutrophil infiltration [24].
Interestingly, this is consistent with the fact that the DSS group had a severe
neutrophil infiltration in this study. However, neutrophil infiltration and MPO content
were significantly decreased in Lp082 group. This shows that Lp082 alleviates UC by
reducing neutrophil infiltration and its secreted MPO content. In a nutshell, our results
suggest that Lp082 may play an anti-UC effect by inhibiting the NF- κ B pathway,
down-regulating pro-inflammatory cytokines, and up-regulating anti-inflammatory
cytokines, reducing MPO content, thereby maintaining immune balance and
protecting the immune barrier.

The mucosal immune system of the intestine mainly consists of Peyer's patch
and lamina propria under enterocyte [25]. The Peyer's patch can deliver captured
antigens to dendritic cells [26]. Then dendritic cells can not only trigger T
cell-mediated cellular immunity and B cell-mediated humoral immunity by presenting
antigens but also affect lamina propria immunity [27]. Combining previous studies,

we found that DSS causes inflammation through the following six ways. First, gut
permeability increases, and harmful substances enter to activate innate immunity, such
as stimulating innate immune cells to produce TNF- α , IL-1 β , and IL-6 [28]. Second,
regulatory T cells produce less IL-10 and have a less inhibitory effect on effector T
cells, resulting in the phenomenon of effector T and regulatory T cell dysregulation in
UC patients [29]. Third, effector T cells promote B cell-mediated humoral immunity
by promoting the secretion of IFN- γ and L-17A [30]. Fourth, effector T cells carried
out immune cell recruitment and formed a vicious immune cycle with chemokines
and cytokines [31]. Fifth, Peyer's patch recognizes antigens and presents them to other
immune cells through dendritic cells [26]. Sixth, antigen-activated neutrophils can
both secrete MPO and recruit more immune cells from the bloodstream to the site of
inflammation, further exacerbating inflammation [32] (**Fig. 6b**). Based on the above 6
reasons, we suggest that in addition to relieving inflammation by inhibiting the NF- κ B
pathway, Lp082 can also regulate inflammatory factors to maintain the balance
between regulatory T cells and effector T cells to regulate intestinal mucosal
immunity, thus maintaining the intestinal mucosal barrier.

Lp082 improved the biological barrier

Numerous studies [23] have shown that probiotics improve the clinical outcome
of IBD patients by influencing host gut microbiota [4]. Herein, we performed a
shotgun metagenomic analysis to investigate whether Lp082 can improve gut
dysbiosis in the UC mice model. As expected, we observed that the intake of DSS
significantly reduced the shannon value but increased PCoA distance, a finding that is
consistent with Wang et al. [33]. The Shannon index reflects gut microbiota richness
and uniformity and is positively correlated with gut microbiota diversity, while the
PCoA distance reflects the difference in the structure of the gut microbiota between
different groups; the higher the PCoA value, the greater the difference in the gut
microbiota structure [34]. In particular, Lp082 treatment remarkably increased the gut
microbiota diversity and reduced gut microbiota structural differences in gut
microbiota, as shown by the cluster analysis and PCoA analysis. On the other hand,
Lp082 also optimized species composition; that is, the abundance of

pro-inflammatory microbiota decreased in the Lp082 group, such as *Helicobacter*
*hepaticus*, a potential pathogen of colitis. Likewise, we observed an increasing trend
in the abundance of potential probiotics in the Lp082 group, such as *Bifidobacterium*
*pseudolongum* and *Bacteroides ovatus*, which reduces colonic inflammation [35],
*Parabacteroides distasonis*, which is negatively associated with obesity and diabetes
[36], *Akkermansia muciniphila* and *Lactobacillus reuteri*, a widely studied probiotic,
*Anaerotruncus sp G3 2012* and *Lactobacillus plantarum*, potential SCFAs-producing
bacteria [37]. The above results indicate that Lp082 is beneficial to optimizing the
diversity, structure, and composition of gut microbiota. After demonstrating that
Lp082 can increase the abundance of potential SCFAs-producing bacteria, further
analysis found that Lp082 can activate two SCFAs-producing microbial metabolic
pathways and the content of SCFAs. Subsequently, correlation analysis proved that
Lp082 may increase SCFAs by activating the SCFAs-producing metabolic pathway of
SCFAs-producing bacteria, so as to inhibit inflammation [38] and regulate host
physiological activity through SCFAs [39]. All of these suggest that Lp082 repaired
the microbial barrier by regulating the gut microbiome.

In conclusions, the Lp082 has an exciting therapeutic effect on UC than SASP.
Also, shotgun metagenome and transcriptome analysis confirmed that Lp082 could
improve gut microbiota dysbiosis, protect intestinal mucosal barrier, regulate
inflammatory pathways, and affect neutrophil infiltration. These findings firmly
support and advocate the clinical translation of Lp082 in the treatment of UC. It can
be suggested that the application of gut microbiota and probiotics in the treatment of
UC should receive more attention. The findings of this study not only provide new
clues for revealing the complex mechanism of gut microbiota in relieving UC, but
also provide evidence for Lp082 as a potential gut microbiota regulator to treat UC.

**References**

- 1. Li XX, Wei B, Goodglick L, Wen T, Xia LJ, Braun J. Investigating Therapeutic
Approach of IBD Using Recombinant Glycoprotein Mucin2. *Faseb Journal*. 2009;23.
- 2. Shi JL, Xie QG, Yue YX, Chen QX, Zhao LN, Evivie SE, et al. Gut microbiota
modulation and anti-inflammatory properties of mixed lactobacilli in dextran sodium

sulfate-induced colitis in mice. Food & Function. 2021;12(11):5130-43; doi:
10.1039/d1fo00317h.

3. Hu LH, Liu JY, Yin JB. Eriodictyol attenuates TNBS-induced UC through
repressing TLR4/NF-kB signaling pathway in rats. Kaohsiung Journal of Medical
Sciences. 2021;37(9):812-8; doi: 10.1002/kjm2.12400.

4. Wang LA, Gao MX, Kang GB, Huang H. The Potential Role of Phytonutrients
Flavonoids Influencing Gut Microbiota in the Prophylaxis and Treatment of
Inflammatory Bowel Disease. Frontiers in Nutrition. 2021;8; doi:
10.3389/fnut.2021.798038.

5. Fedorak RN. Understanding why probiotic therapies can be effective in treating
IBD. Journal of clinical gastroenterology. 2008;42 Suppl 3 Pt 1:S111-5; doi:
10.1097/MCG.0b013e31816d922c.

6. Sun MY, Liu YJ, Song YL, Gao Y, Zhao FJZ, Luo YH, et al. The ameliorative
effect of *Lactobacillus plantarum*-12 on DSS-induced murine colitis. Food & Function.
2020;11(6):5205-22; doi: 10.1039/d0fo00007h.

7. Burger-van Paassen N, Vincent A, Puiman PJ, van der Sluis M, Bouma J, Boehm
G, et al. Regulation of the intestinal mucin MUC2 expression by short chain fatty
acids: implications for epithelial protection. FASEB Journal. 2009;23.

8. Taniguchi T, Tsukada H, Nakamura H, Kodama M, Fukuda K, Saito T, et al.
Effects of the anti-ICAM-1 monoclonal antibody on dextran sodium sulphate-induced
colitis in rats. Journal of gastroenterology and hepatology. 1998;13(9):945-9; doi:
10.1111/j.1440-1746.1998.tb00766.x.

9. Philpott JR, Miner PB, Jr. Antisense inhibition of ICAM-1 expression as therapy
provides insight into basic inflammatory pathways through early experiences in IBD.
Expert Opin Biol Ther. 2008;8(10):1627-32; doi: 10.1517/14712598.8.10.1627.

10. Edelblum KL, Turner JR. The tight junction in inflammatory disease:
communication breakdown. Current Opinion in Pharmacology. 2009;9(6):715-20; doi:
10.1016/j.coph.2009.06.022.

11. Stio M, Retico L, Annese V, Bonanomi AG. Vitamin D regulates the
tight-junction protein expression in active UC. Scandinavian Journal of

Gastroenterology. 2016;51(10):1193-9; doi: 10.1080/00365521.2016.1185463.

12. Cordeiro BF, Alves JL, Belo GA, Oliveira ER, Braga MP, da Silva SH, et al.

Therapeutic Effects of Probiotic Minas Frescal Cheese on the Attenuation of UC in a

Murine Model. *Frontiers in Microbiology*. 2021;12; doi: 10.3389/fmicb.2021.623920.

13. Nakamura S, Ohtani H, Watanabe Y, Fukushima K, Matsumoto T, Kitano A, et al.

In situ expression of the cell adhesion molecules in inflammatory bowel disease.

Evidence of immunologic activation of vascular endothelial cells. *Laboratory*

*investigation; a journal of technical methods and pathology*. 1993;69(1):77-85.

14. Mitselou A, Grammeniatis V, Varouktsi A, Papadatos SS, Klaroudas A, Katsanos

1815 K, et al. Immunohistochemical Study of Adhesion Molecules in Irritable Bowel

Syndrome: A Comparison to Inflammatory Bowel Diseases. *Advanced biomedical*

*research*. 2021;10:21; doi: 10.4103/abr.abr_2_20.

15. Ruco LP, de Laat PA, Matteucci C, Bernasconi S, Sciacca FM, van der Kwast TH,

et al. Expression of ICAM-1 and VCAM-1 in human malignant mesothelioma. *The*

*Journal of pathology*. 1996;179(3):266-71.

16. Bauer J, Namineni S, Reisinger F, Zoller J, Yuan DT, Heikenwalder M.

Lymphotoxin, NF-kappa B, and Cancer: The Dark Side of Cytokines. *Digestive*

*Diseases*. 2012;30(5):453-68; doi: 10.1159/000341690.

17. Xiao B, Laroui H, Ayyadurai S, Viennois E, Charania MA, Zhang YC, et al.

Mannosylated bioreducible nanoparticle-mediated macrophage-specific TNF-alpha

RNA interference for IBD therapy. *Biomaterials*. 2013;34(30):7471-82; doi:

10.1016/j.biomaterials.2013.06.008.

18. 김연하, 김유림, 김성중, 황호근, Choi S-C, 김경숙, et al. Rebamipide Protects

Colonic Damage Induced by Trinitrobenzene Sulfonic Acid (TNBS) via

Down-Regulation of TNF- α , IL-1 β , and ICAM-1. *Anatomy and Cell Biology*.

2004;37(2):149-56.

19. Zhou X, Liu H, Zhang J, Mu J, Zalan Z, Hegyi F, et al. Protective effect of

*Lactobacillus fermentum* CQPC04 on dextran sulfate sodium-induced colitis in mice

is associated with modulation of the nuclear factor-kappa B signaling pathway.

*Journal of Dairy Science*. 2019;102(11):9570-85; doi: 10.3168/jds.2019-16840.

- 20. Mohammadnia-Afrouzi M, Hosseini AZ, Khalili A, Abediankenari S, Amari A,
Aghili B, et al. Altered microRNA Expression and Immunosuppressive Cytokine
Production by Regulatory T Cells of UC Patients. *Immunological Investigations*.
2016;45(1):63-74; doi: 10.3109/08820139.2015.1103749.
- 21. Sato Y, Takahashi S, Kinouchi Y, Shiraki M, Endo K, Matsumura Y, et al. IL-10
deficiency leads to somatic mutations in a model of IBD. *Carcinogenesis*.
2006;27(5):1068-73.
- 22. Hume GE, Fowler EV, Lincoln D, Eri R, Templeton D, Florin TH, et al.
Angiotensinogen and transforming growth factor beta1: novel genes in the
pathogenesis of Crohn's disease. *Journal of medical genetics*. 2006;43(10):e51; doi:
10.1136/jmg.2005.040477.
- 23. Trevisin M, Pollock W, Dimech W, Savige J. Evaluation of a multiplex flow
cytometric immunoassay to detect PR3- and MPO-ANCA in active and treated
vasculitis, and in inflammatory bowel disease (IBD). *Journal of immunological*
*methods*. 2008;336(2):104-12; doi: 10.1016/j.jim.2008.03.012.
- 24. Chin AC, Parkos CA. Neutrophil transepithelial migration and epithelial barrier
function in IBD: potential targets for inhibiting neutrophil trafficking. *Annals of the*
*New York Academy of Sciences*. 2006;1072:276-87; doi: 10.1196/annals.1326.018.
- 25. Jonker MA, Hermsen JL, Sano Y, Heneghan AF, Lan JG, Kudsk KA. Small
intestine mucosal immune system response to injury and the impact of parenteral
nutrition. *Surgery*. 2012;151(2):278-86; doi: 10.1016/j.surg.2010.10.013.
- 26. Li HS, Gelbard A, Martinez GJ, Esashi E, Zhang HY, Nguyen-Jackson H, et al.
Cell-intrinsic role for IFN-alpha-STAT1 signals in regulating murine Peyer patch
plasmacytoid dendritic cells and conditioning an inflammatory response. *Blood*.
2011;118(14):3879-89; doi: 10.1182/blood-2011-04-349761.
- 27. Santucci L, Agostini M, Bruscoli S, Mencarelli A, Ronchetti S, Ayroldi E, et al.
GTR modulates innate and adaptive mucosal immunity during the development of
experimental colitis in mice. *Gut*. 2007;56(1):52-60; doi: 10.1136/gut.2006.091181.
- 28. Debnath T, Kim DH, Lim BO. Natural Products as a Source of
Anti-Inflammatory Agents Associated with Inflammatory Bowel Disease. *Molecules*.

2013;18(6):7253-70; doi: 10.3390/molecules18067253.

29. Goldberg R, Scotta C, Cooper D, Nissim-Eliraz E, Nir E, Tasker S, et al.
Correction of Defective T-Regulatory Cells From Patients With Crohn's Disease by
Ex Vivo Ligation of Retinoic Acid Receptor-alpha. *Gastroenterology*.
2019;156(6):1775-87; doi: 10.1053/j.gastro.2019.01.025.

30. Cook L, Stahl M, Han X, Nazli A, MacDonald KN, Wong MQ, et al. Suppressive
and Gut-Reparative Functions of Human Type 1 T Regulatory Cells. *Gastroenterology*.
2019;157(6):1584-98; doi: 10.1053/j.gastro.2019.09.002.

31. Singh UP, Singh NP, Murphy EA, Price RL, Fayad R, Nagarkatti M, et al.
Chemokine and cytokine levels in inflammatory bowel disease patients. *Cytokine*.
2016;77:44-9; doi: 10.1016/j.cyto.2015.10.008.

32. Zhou GX, Yu L, Fang LL, Yang WJ, Yu TM, Miao YL, et al. CD177(+)
neutrophils as functionally activated neutrophils negatively regulate IBD. *Gut*.
2018;67(6):1052-63; doi: 10.1136/gutjnl-2016-313535.

33. Wang R, Chen T, Wang Q, Yuan XM, Duan ZL, Feng ZY, et al. Total Flavone of
*Abelmoschus manihot* Ameliorates Stress-Induced Microbial Alterations Drive
Intestinal Barrier Injury in DSS Colitis. *Drug Design Development and Therapy*.
2021;15:2999-3016; doi: 10.2147/dddt.S313150.

34. Suchodolski JS, Markel ME, Garcia-Mazcorro JF, Unterer S, Heilmann RM,
Dowd SE, et al. The Fecal Microbiome in Dogs with Acute Diarrhea and Idiopathic
Inflammatory Bowel Disease. *Plos One*. 2012;7(12); doi:
10.1371/journal.pone.0051907.

35. Yang C, Du Y, Ren D, Yang X, Zhao Y. Gut microbiota-dependent catabolites of
tryptophan play a predominant role in the protective effects of turmeric
polysaccharides against DSS-induced UC. *Food Funct*. 2021;12(20):9793-807; doi:
10.1039/d1fo01468d.

36. Cai W, Xu JX, Li G, Liu T, Guo XL, Wang HJ, et al. Ethanol extract of propolis
prevents high-fat diet-induced insulin resistance and obesity in association with
modulation of gut microbiota in mice. *Food Research International*. 2020;130; doi:
10.1016/j.foodres.2019.108939.

- 37. Wang J, Ji HF, Wang SX, Liu H, Zhang W, Zhang DY, et al. Probiotic
*lactobacillus plantarum* Promotes Intestinal Barrier Function by Strengthening the
Epithelium and Modulating Gut Microbiota. *Frontiers in Microbiology*. 2018;9; doi:
10.3389/fmicb.2018.01953.
- 38. Wang SL, Zhang SY, Huang SM, Wu ZH, Pang JM, Wu YJ, et al. Resistant
Maltodextrin Alleviates Dextran Sulfate Sodium-Induced Intestinal Inflammatory
Injury by Increasing Butyric Acid to Inhibit Proinflammatory Cytokine Levels.
*Biomed Research International*. 2020;2020; doi: 10.1155/2020/7694734.
- 39. Holota Y, Dovbynychuk T, Kaji I, Vareniuk I, Dzyubenko N, Chervinska T, et al.
The long-term consequences of antibiotic therapy: Role of colonic short-chain fatty
acids (SCFAs) system and intestinal barrier integrity. *Plos One*. 2019;14(8); doi:
10.1371/journal.pone.0220642.

4. In general, the abstract could be re-written to describe the results from a higher
level, rather than just listing the altered genes. Close the abstract with a statement
connecting the paper results to the broader scientific field.

**Response:** We appreciate your valuable and helpful comment. According to your
suggestion, we have rewritten the abstract. The rewritten content links the results of
the paper with the broader scientific field. The revised content is as follows. (Page 2,
line: 25-59)

Probiotics can effectively improve ulcerative colitis (UC), but the mechanism is still
unclear. Here, shotgun metagenomic and transcriptome analyses were performed to
explore the therapeutic effect and the mechanism of the probiotic *lactobacillus*
*plantarum* HNU082 (Lp082) on UC. The results showed that Lp082 treatment
significantly ameliorated dextran sulfate sodium (DSS) -induced UC in mice, which
was manifested as increases in body weight, water intake, food intake, colon length,
and decreases in disease activity index (DAI), immune organ index, inflammatory
factors, and histopathological scores after Lp082 intake. An in-depth study discovered
that Lp082 could improve the intestinal mucosal barrier and relieve inflammation by
co-optimizing the biological barrier, chemical barrier, mechanical barrier and immune

barrier. Specifically, Lp082 rebuilt the biological barrier by regulating the intestinal
microbiome and increasing the production of short-chain fatty acids (SCFAs). Lp082
improved the chemical barrier by reducing intercellular cell adhesion molecule-1,
vascular cell adhesion molecule and increasing goblet cells and mucin2. Lp082
ameliorated the mechanical barrier by increasing the zonula occludens-1 (ZO-1),
zonula occludens-2 (ZO-2), and occludin while decreasing claudin-1 and claudin-2.
Lp082 optimized the immune barrier by reducing the content of IL-1 β , IL-6, TNF- α ,
MPO, IFN- γ and increasing the IL-10, TGF- β 1, and TGF- β 2, inhibiting the NF-kB
signalling pathway. Taken together, probiotic Lp082 can play a protective role in a
DSS-induced colitis mouse model by protecting the intestinal mucosal barrier,
attenuating the inflammatory response, and regulating microbial imbalance. This
study provides support for the development of probiotic-based microbial products as
an alternative treatment strategy for UC.

Importance

Many studies have focused on the therapeutic effect of probiotics on UC, but few
studies have paid attention to the mechanism of probiotics, especially the therapeutic
effect. This study suggests that Lp082 has a therapeutic effect on colitis in mice. Its
mechanisms of action include protect the mucosal barrier and actively modulate the
gut microbiome, modulate inflammatory pathways and reduce neutrophil infiltration.
Our study enriches the mechanism and provides a new prospect for probiotics in the
treatment of colitis, helps to deepen the understanding of the intestinal mucosal barrier,
and provides guidance for the future probiotic treatment of human colitis.

Keywords: Lactobacillus plantarum HNU082, ulcerative colitis, intestinal mucosal
barrier, short chain fatty acid, transcriptome, shotgun metagenome, cytokine

5. As written, lines 72-73 suggest Yucha has resistance to acid and bile salts, but I
assume that the authors mean Lp082 is resistant. Re-wording the sentence and adding
a clarification on what point the authors are trying to make about acid and bile salt
resistance would help alleviate the confusion here.

**Response:** Thank you for your comment. We deeply agree with your suggestion. It is
true that we did not express it clearly. We apologize for the confusion caused to you.
According to your helpful advice, we have revised this sentence and the revised
content is as follows. (Page 4, line: 98-100)

The strain of *Lactobacillus plantarum* HNU082 (Lp082) was originally isolated
from a traditional fermented food-fish tea of the Li people in Hainan Province,
China, which has a good safety profile and tolerance to acids and bile salts [1].

**Reference**

1. Zhang J, Wang X, Huo D, Li W, Hu Q, Xu C, et al. Metagenomic approach
reveals microbial diversity and predictive microbial metabolic pathways in Yucha, a
traditional Li fermented food. *Scientific Reports*. 2016;6; doi: 10.1038/srep32524.

6. Referring to lines 77-92: The authors interchange physiological results with
techniques as if they are the same things. Before describing the specific things you
were evaluating, describe what you were looking for at a high level. Then separate
physiological indicators from methods (e.g., rather than say, "evaluated physiological
indexes and shotgun metagenomic sequencing," use language like "evaluated
inflammation, microbial community composition and activity...using ELISA,
immunohistochemistry, metagenomic sequencing, and RNA-seq."

**Response:** We appreciate your valuable and helpful comment. We deeply agree with
your suggestion. We do indeed have a language problem on this issue which created
confusion. According to your helpful advice, we have changed this sentence and other
places in the article. The revised content is as follows. (Page 23, line: 666-671)

After the UC model was established by DSS, mice were given Lp082 by gavage to
observe the therapeutic effect of the bacteria on DSS-induced UC.. Various tissue
samples, including immune organs, serum, proximal colon, fecal, cecal contents,
distal colon, and other tissues, were collected. Techniques such as ELISA,
immunohistochemistry, metagenomic sequencing, and RNA-seq were used to assess
inflammation, microbial community composition, and gene expression. (Fig. 1a).

7. Potentially incorrect information: Lines 97-98 days and scores do not line up with
the data reported in figure 1B.

**Response:** Thank you for your comment. We are very sorry for our incorrect writing.
We apologize for the confusion caused to you. We have redescribed **Fig. 1b**, and the
modified contents are as follows. (Page 6, line: 153-168)

The results showed that from 1 to 7 days, the water intake, food intake, and body
weight of the DSS group, the Lp082 group, and the SASP group all showed a similar
degree of gradual decrease, and these three groups were all significantly different
from the Control group on day 7 ($p < 0.05$), which may be because these three groups
were all under the same DSS modeling conditions on days 0-7. Then on the 8th to
15th day, the water intake, food intake, and body weight of the DSS group were still
decreasing, but the water intake, food intake, and body weight of Lp082 and SASP
group gradually increased. Specifically, the water and food intake of the Lp082
combined SASP group increased significantly from day 9 ($p < 0.05$), and body weight
increased significantly from day 12 ($p < 0.05$). The DAI index of the DSS group,
Lp082 group, and SASP group increased significantly ($p < 0.05$) from the third day
compared with the Control group. After stopping DSS gavage on the 8th day, the DAI
index of the DSS self-healing group still increased, while that of the Lp082 group and
SASP group gradually decreased from the 10th day, and the degree of decrease in the
Lp082 group was greater than that in the SASP group (**Fig. 1b**).

8. Abbreviations should be described in the text as they arise, not in an additional
section at the end of the paper (page 20).

**Response:** We are grateful for the suggestion. Thank you very much for pointing out
our problem, we deeply agree with your suggestion. According to your helpful advice,
we have corrected this by adding a description of abbreviations to the article.

9. After revising the manuscript, a thorough and detailed assessment and correction of
sentence structure would improve the readability of the paper dramatically.

**Response:** We appreciate the reviewer's attention to the flaws of our text. After
revising the manuscript, we have made a comprehensive and careful assessment and
correction of the sentence structure and carefully checked the full text. The language
presentation was improved with assistance from a native English speaker with an
appropriate research background.

10. Abbreviations, capitalization, italics, and spacing are inconsistent throughout and
should be fixed for a final draft. E.g. Lp082(most commonly used in the draft)/Lp082
(lines 78-79) or HNU082 (correct)/HNU082 (line 23).

**Response:** Thank you for your comment. We have carefully checked abbreviations,
capitals, italics and spaces. We tried our best to improve the manuscript and made
some changes in the manuscript. These changes will not influence the content and
framework of the paper. And here we did not list the changes but marked in yellow in
revised paper.

11. Review your usage of "prove" in your manuscript (notably in the discussion
section) as the experiments presented provide largely correlative data.

**Response:** Thank you for your comment and we have corrected this error and used
the word "prove" more carefully. We also carefully checked the text to ensure the
accuracy of our other words.

Once again, we thank you for the time you put into reviewing our paper. We have
worked hard to answer your questions and look forward to meeting your expectations.
If you have any dissatisfaction, please communicate with us, and we will make
changes and improvements as quickly as possible. We are very grateful for your effort
in reviewing our paper and your positive feedback. Your evaluation of our work is
precise, and your dedication is commendable. Since your input is invaluable for future
publications, we would like to expressly thank you for your contribution.

Reviewer #2 (Public repository details (Required)):

metagenomics sequencing and metabolome data are needed to deposit at a repository.

**Response:** We really appreciate your reminder from the bottom of our hearts. We are
very sorry for our negligence of metagenome and transcriptome raw data. We have
uploaded the metagenomic and transcriptome raw data, and the modifications in the
manuscript have been highlighted. (Page 27, Line: 791-792)

The sequence data reported in this paper have been deposited in the NCBI
database (metagenomic sequencing data and transcriptome sequencing
data:PRJNA812272).

As is customary, our data will be made public after the article is received.

Reviewer #2 (Comments for the Author):

**Response:** We appreciate the time and effort you dedicated to providing feedback on
our manuscript and are grateful for the insightful comments and valuable
improvements to our manuscript. We have discussed your comments carefully and we
sincerely accept the suggestions. Your comments provided valuable insights to refine
its contents and analysis. In this document, we try to address the issues raised as best
as possible. All revisions in the manuscript have been highlighted in yellow. You can
kindly find the point-to-point responses to reviewers' comments in the following text.
We thoroughly double-checked the manuscript. For detail, please see the following
answers.

Major comments:

1. Authors claim that "we chose Lp082 to study the mechanism of probiotics in
preventing UC", however, the animal was treated with various reagents followed by

DSS challenge. Please explain how this setting could serve well for assessing the
effects of probiotics on prevention UC? Authors should discriminate the difference
between "prevention" and "treatment", and pay more attention for accuracy of
wording.

**Response:** We appreciate your valuable and helpful comment. We apologize for the
language problems in the original manuscript. The language presentation was
improved with assistance from a native English speaker with appropriate research
background. We apologize for the confusion and inconvenience caused to you. In fact,
we are studying the effect of Lp082 in the treatment of UC. We used DSS to establish
a model of UC and then treated it with Lp082. We have changed the sentence you
mentioned above to: So the Lp082 strain becomes a good choice for the study of
*lactobacillus plantarum* in the treatment of UC. The changes have been highlighted in
the article. We have carefully checked the wording of the full text and corrected the
preventive effect to the therapeutic effect. Thank you very much for pointing this out.
It was very helpful.

2. Basically only one biological repeat was conducted in this study. At least two
biological repeats are acceptable for this purpose. Please repeat one more animal
assay during next round of revision.

**Response:** We appreciate your valuable and helpful comment. Thank you very much
for pointing out this issue. It is true that we did not express clearly. In fact, we set up 6
biological replicates for each group. According to your helpful suggestions, we have
carefully checked the whole paper, and added descriptions of sample size and number
of repeats in material and methods, legends and corresponding places in the article.
The changes have been highlighted in the text in yellow. The rewritten content is
more detailed, and the details are as follows:

After the experiment, the spleen, liver, kidney and colon of 8 mice were selected from
each group for observation and measurement. (Page 6, line: 170-172)

To further evaluate the effects of Lp082 on inflammatory cytokines in mice with

colitis, serum of 6 mice in each group was randomly collected after the experiment,
and the levels of pro-inflammatory cytokines TNF- α , IL-1 β , IFN- α , IL-6, MPO and
anti-inflammatory cytokines IL-10 were detected by ELISA kit. (Page 8, line:
208-213)

At the end of the experiment, the 1cm portion of the distal colon of 6 mice in each
group was selected randomly for HE staining, and histopathological score and
intestinal wall thickness were further measured ($n=6$). (Page 8, line: 220-224)

At the end of modeling (day 7 of the experiment), feces of 6 mice in each group were
randomly selected for metagenomic sequencing, and at the end of treatment (day 15
of the experiment), feces of 6 mice in each group were selected for metagenomic
sequencing, to observe the effects of DSS and Lp082 on the intestinal microecology
of mice. (Page 9, line: 258-262)

To prove the above findings, we further used gas chromatography-mass spectrometry
(GC-MS) to detect the content of SCFAs in cecal contents of 6 mice in each group.
(Page 11, line: 308-309)

At the end of the experiment, 6 mice from each group were randomly selected for
colon transcriptome sequencing, and the volcanic map was drawn based on the
preliminary gene distribution analysis results. (Page 13, line: 350-352)

C57BL/6J mice aged 7 weeks were randomly divided into 4 groups: control group
($n=8$), dextran sulfate sodium (DSS) group ($n=8$), lactobacillus plantarum HNU082
(Lp082) group ($n=8$), and salazosulfapyridine (SASP) group ($n=8$). (Page 23, line:
659-661)

After the mice were euthanized, the colon length of 8 mice in each group was
measured, the weight of spleen, liver, and kidney of 8 mice in each group was

measured. (Page 23, line: 677-679)

Before euthanasia, 6 mice were randomly selected from each group, and blood was
collected from the orbital venous plexus by a capillary tube. (Page 24, line: 686-687)

Finally, the levels of interleukin-1beta (IL-1 β), interleukin-6 (IL-6), interleukin-10
(IL-10), interleukin-17A (IL-17A), interferon-gamma (IFN- γ), Tumor necrosis
factor-alpha (TNF- α), and Myeloperoxidase (MPO) in the serum of 6 randomly
selected mice from each group were measured using the corresponding ELISA kits
(X-Y Biotechnology, Shanghai, China), as previously described. (Page 24, line:
690-694)

After euthanasia, the distal 1cm colons of 6 mice in each group were randomly
selected for HE staining section, histopathological score, and intestinal wall thickness
measurement. (Page 24, line: 697-699)

On the other hand, 8 mice were selected from each group, and their colonic tissues
were labeled with mucin 2 and ZO-1 antibodies, respectively [75], for further
immunofluorescence staining (Servicebio, Wuhan, China). (Page 25, line: 710-712)

Six mice were randomly selected at two time points (day 7 and day 15 of the
experiment) for metagenomic sequencing of feces. (Page 25, line: 728-729)

At the end of the experiment, the cecal contents of 6 mice from each group were
randomly selected for SCFAs determination, and the specific steps were as follows:
(Page 26, line: 742-743)

At the end of the experiment, colon tissues of 6 mice from each group were randomly
selected for RNA sequencing. (Page 26, line: 757-758)

We consider our results to be credible on the premise of 6 biological replicates per
group. We have carefully reviewed the full text and supplemented descriptions of data
volumes and biological replicates where measurement data appeared. Modifications in
the article are highlighted in yellow.

3. Please improve layouts of figures, and pay attention to size, location of symbols.

**Response:** We appreciate your valuable and helpful suggestion. According to the your
comment, we have gone through all the images carefully and refined the layout, size
and placement of symbols.

4. Please improve the language and grammar.

**Response:** We apologize for the language problems in the original manuscript. The
language presentation was improved with assistance from a native English speaker
with an appropriate research background. We deeply appreciate your valuable and
helpful comments.

5. Please provide the H&E staining results for entire swiss roll in figure 2.

**Response::** We appreciate your valuable and helpful comment. Indeed, our slicing
pictures that are not in line with the rules. We supplement the full slicing results of
40X and use this to zoom in at 100X and 200X. Thank you very much for your
suggestion; we will pay more attention in the following writing.

6. Authors claim that "that Lp082 could improve UC by regulating gut microbiota,
intestinal mucosal barrier, inflammatory pathways and neutrophil infiltration", please
provide direct evidence to support Lp082 effects on "mucosal barrier". Manuscript
shows the transcriptome data, however, transcriptome analysis on host genes are far
away from real expression and function.

**Response:** We appreciate your valuable and helpful comment. The pathogenesis of
UC is the result of the combined effect of genetically susceptible hosts and the
environment, and its common pathological outcome is the damage of the structure and

function of the intestinal mucosal barrier. The intestinal mucosal barrier is damaged,
resulting in an increase in the permeability of the intestinal epithelial barrier, and
further stimulation of intestinal contents, bacteria, and toxins promotes the immune
response to intestinal inflammation. The normal intestinal mucosal barrier consists of
mechanical barrier, chemical barrier, immune barrier, and biological barrier. The
chemical barrier refers to the glue-like mucin layer covering the surface of intestinal
epithelial cells, which is mainly composed of MUC-2 secreted by goblet cells,
digestive juices, and bacteriostatic substances produced by normal parasitic bacteria
in the intestinal lumen [1]. The mechanical barrier is the most important part of the
intestinal mucosal barrier. Its structural basis is the intestinal mucosal epithelial cells
and the tight junctions (TJ) between the epithelial cells [2]. The immune barrier is
associated with immune cells, and inflammatory factors [3]. The biological barrier is a
normal intestinal colony of bacteria that is resistant to colonization by foreign strains
[4]. The results of the study found that Lp082 can improve the intestinal mucosal
barrier by synergistically optimizing the biological barrier, chemical barrier,
mechanical barrier and immune barrier, thereby alleviating UC. Specifically, We
found that Lp082 rebuilt the biological barrier by regulating the intestinal microbiome
and increasing the SCFAs. Lp082 improved the chemical barrier by reducing ICAM-1,
VCAM, and increasing goblet cells and mucin2. Lp082 ameliorated the mechanical
barrier by increasing the ZO-1, ZO-2, and occludin and decreasing claudin-1 and
claudin-2. Lp082 optimized the immune barrier by reducing the content of IL-1 β ,
IL-6, TNF- α , MPO, IFN- γ and increasing the IL-10, TGF- β 1, and TGF- β 2. From the
above four aspects, we demonstrated that Lp082 can indeed improve the "intestinal
mucosal barrier" to treat DSS-induced UC.

This result is not only supported by transcriptomic data, we have indeed done a
lot of experiments and validation. First, we studied some basic indicators and found
that Lp082 could not only significantly inhibit the decrease of body weight, water
intake and food intake induced by DSSS in mice, but also significantly inhibit the
increase of DAI and immune organ index induced by DSSS, as well as the decrease of

colon length caused by DSS (**Fig. 1a-1d**). Second, we measured the protein content of
six inflammatory cytokines in mouse serum, and found that Lp082 could significantly
reduce the increase of IL-1 β , IL-6, TNF- α , MPO, IFN- γ induced by DSS, and increase
the protein content of IL-10 in mice (**Fig. 1e**). Third, we performed HE staining
section experiment and immunofluorescence protein experiment. The results showed
that Lp082 could not only improve the crypt infiltration, goblet cell loss and intestinal
mucosal ulcer induced by DSS, but also could reduce the increase of histopathology
score caused by DSS and reduce the loss of ZO-1 and MUC-2 proteins caused by
DSS (**Fig. 2a-2g**). Fourth, we collected fecal samples on day 7 for metagenomic
sequencing. The results of Shotgun metagenomic data analysis showed that Lp082
could increase α -diversity and β -diversity, reduce the differences in species
composition, increase the content of beneficial bacteria and inhibit the abundance of
harmful bacteria in mice (**Fig. 3a-3d**). Fifth, we used gas chromatography-mass
spectrometry to determine the content of SCFAs in the intestinal contents of mice, and
found that Lp082 could significantly inhibit the reduction of acetic acid, propionic
acid, butyric acid, isobutyric acid and valeric acid induced by DSS, and restore the
content of SCFAs in mice (**Fig. 4b**). Sixth, we sequenced the transcriptome of colon
tissue, and the results showed that Lp082 not only affected gene expression
distribution, but also affected inflammation and cancer-related and KEGG,GO-BP
pathways (**Fig. 5a-5g**). These experiments provide data support for our derivation,
because the study did integrate metagenomics, transcriptomics, proteomics, HE
stained sections, immunofluorescent proteins and other experimental data, and found
that Lp082 can modulate the immune, chemical, mechanical and biological barriers,
which means that Lp082 can improve the intestinal mucosal barrier. Our data were
not less than 6 replicates in each group, and our data were absolutely reliable and
sufficient to support the results of our paper.

Maybe we didn't describe it very well, so based on your suggestion, we have
rewritten the discussion section to more clearly describe the improvement effect of
Lp082 on the intestinal mucosal barrier, and the rewritten content is as follows: (Page

16, line: 459-637)

DISCUSSION

The normal intestinal mucosal barrier is composed of mechanical, chemical immune
and biological barriers. The Lp082 has good efficacy in treating UC, which motivates
2258 us to explore further its mechanism of action in the treatment of UC. The results of
2259 the study found that Lp082 can improve the intestinal mucosal barrier by
2260 synergistically optimizing the biological, chemical, mechanical and immune barriers,
thereby alleviating UC. In addition to optimizing the intestinal mucosal barrier,
regulating inflammatory pathways and influencing neutrophil infiltration are potential
mechanisms of Lp082 in treating UC.

Lp082 improved chemical barrier

The chemical barrier refers to the glue-like mucin layer covering the surface of
intestinal epithelial cells, which is mainly composed of MUC-2 secreted by goblet
cells, digestive juices and bacteriostatic substances produced by normal parasitic
bacteria in the intestinal lumen [1]. The chemical barrier plays an important role in
isolating the internal and external environment of the intestinal tract, lubricating the
intestinal mucosa, and inhibiting the entry of harmful substances in the intestinal
lumen [5]. The intestinal mucosal wall thickness was significantly increased in the
DSS group, whereas it was significantly decreased after Lp082 ingestion (**Fig. 2c**). In
DSS-induced UC, the thicker the intestinal mucosal wall, indicating more severe
inflammation. In addition, the H&E staining result showed that the number of goblet
cells decreased in the DSS group (red arrow), whereas the number of goblet cells
increased (yellow arrow) after Lp082 ingestion (**Fig. 2a**). The immunofluorescent
protein content of MUC-2, which is mainly secreted by goblet cells, was significantly
decreased in the DSS group (**Fig. 2d**), and the areal density of MUC-2 (**Fig. 2f**) and
the mRNA expression of MUC-2 were also significantly decreased in the DSS group
(**Fig. 5c**), while the immunofluorescence protein content, areal density and mRNA
expression of MUC-2 all increased in the Lp082 group,
Sun et al. [6] observed the same phenomenon that *lactobacillus plantarum 12* can
repair the intestinal mucosal chemical barrier by increasing the content of MUC-2.

Burger-van Paassen et al. [7] found that intake of SCFAS could increase the
expression abundance of MUC-2 mRNA in cells. The mRNA expressions of ICAM-1
and VCAM-1 were significantly decreased after Lp082 intake. Taniguchi et al. [8]
found that anti-ICAM-1 treatment significantly attenuated colonic mucosal damage,
while Philpott et al. [9] found that adhesion molecules ICAM-1 & VCAM-1 induced
intestinal mucosal lesions. Lp082 has been shown to be effective in relieving
intestinal mucosal lesions (i.e., reduced ulceration and inflammatory cell infiltration
caused by DSS). So, we speculate that Lp082 reduces mucosal lesions by reducing
ICAM-1 and VCAM. The above results showed that probiotic Lp082 increased the
MUC-2 content in the mucus layer by restoring the number of goblet cells, relieved
the intestinal mucosal damage caused by ICAM-1 and VCAM-1, so as to repaired the
chemical barrier.

Lp082 improved mechanical barrier

The mechanical barrier is the most important part of the intestinal mucosal
barrier. Its structural basis is the intestinal mucosal epithelial cells and the tight
junctions (TJ) between the epithelial cells [2]. The mechanical barrier can effectively
prevent harmful substances such as bacteria and endotoxins from entering the blood
through the intestinal mucosa.iers The aberrant structure of tight junction (TJ)
proteins between intestinal epithelial cells, such as the reduction of ZO-1, ZO-2, and
occludin, is one of the critical factors leading to the disruption of the gut mechanical
barrier in UC patients [10]. Several studies have identified TJ protein as a new target
for the current treatment of UC [11]. Because Lp082 excellently improved
histopathology, we speculated that Lp082 also has a regulatory effect on TJ molecules.
To this end, we analyzed major TJ proteins, including ZO-1, ZO-2, occludin. As
expected, the mRNA expression and immunofluorescence protein content of ZO-1
and the mRNA expression of ZO-2 and occludin were significantly decreased in
DSS-induced UC mice but improved in the Lp082 treatment group. These are
consistent with the findings of Cordeiro et al. [12] that ZO-1 and ZO-2 were
significantly decreased in UC but increased after probiotic Minas Frescal cheese
intake, indicating that the improvement of the mechanical barrier by regulating TJ

may be one of the mechanisms by which probiotic Lp082 exerts anti-UC. In addition,
the mRNA expression of another particular tight junction protein, ICAM-1 and
VCAM-1, was increased in the DSS group. It is consistent with the findings of
elevated ICAM-1 and VCAM-1 in IBD patients in clinical studies [13]. Mitselou et al.
[14] found that the adhesion molecules ICAM-1 and VCAM-1 induced intestinal
mucosal injury. Taniguchi et al. [8] found that anti-ICAM-1 treatment attenuated
colonic mucosal injury. It has been reported that ICAM-1 and VCAM-1 can increase
the permeability of intestinal mucosa [15]. Interestingly, the mRNA expression of
ICAM-1 and VCAM-1 was found to decrease after Lp082 ingestion. Therefore, it can
be thought that the alleviation of UC by Lp082 may be due to down-regulation of
ICAM-1, VCAM-1 and increase protein quantity and mRNA expression of
ZO-1, ZO-2, so as to reduce intestinal mucosal permeability, thereby inhibiting the
entry of harmful bacteria and undigested food and toxins into the body and reducing
inflammation. These results suggest that Lp082 repairs the intestinal mechanical
barrier by regulating TJ.

Lp082 improved the immune barrier

Although the exact etiology of UC is complex and uncertain, studies suggest that
the NF- κ B pathway plays a vital role in the pathogenesis of UC [3]. Our study has
proved that Lp082 inhibits the NF- κ B pathway by down-regulating the mRNA
expression of NF- κ B2, NF- κ B1, COX-2, Rel α , Toll4, iNOS, and that NF- κ B can also
regulate inflammation by regulating cytokines [16]. Therefore, it can be suggested
that Lp082 also has a specific regulatory effect on cytokines. To confirm this, we
analyzed the cytokines associated with NF- κ B. As expected, we observed that the
mRNA expression level content of pro-inflammatory cytokines (TNF- α , IL-1 β , and
IL-6) were significantly increased in the DSS group but significantly decreased in the
Lp082 group, It is interesting to note that the protein levels of TNF- α , IL-1 β , and IL-6
detected by elisa kit were also increased in the DSS group and decreased after Lp082
intake. Among them, TNF- α can promote the proliferation and differentiation of T
cells and increase intestinal inflammation [17]. The upregulation of IL-1 β is involved
in the recruitment and retention of leukocytes in inflamed tissues and can activate

innate immune lymphocytes [18]. IL-6 activates NF- κ B to regulate the dextran sulfate
sodium-induced colitis in mice [19]. The above results indicate that Lp082 alleviates
UC by inhibiting the levels of pro-inflammatory factors (TNF- α , IL-1 β , and IL-6).
Interestingly, we also found that the mRNA expressions of anti-inflammatory
cytokines IL10, TGF-1, and TGF-2 were significantly decreased in the DSS group but
increased in the Lp082 group. Il-10 protein levels measured by elisa kit also decreased
in the DSS group and increased in the Lp082 group. Surprisingly, IL10, TGF-1, and
TGF-2 were shown to activate Treg and anti-inflammatory macrophages to alleviate
UC [20]. And Sato et al. [21] also found that the loss of IL-10 spontaneously gave rise
to IBD, and Hume et al. [22] found that TGF- β 1 and TGF- β 2 could dramatically
relieve intestinal inflammation in DSS-induced colitis mice. These results suggest that
Lp082 alleviates UC by increasing the levels of anti-inflammatory factors IL10,
TGF-1, and TGF-2. We further analyzed the specific regulatory effects of Lp082 on
intestinal mucosal immunity. In addition to inflammatory factors, we also noticed that
a heme protein, MPO, was significantly reduced in the Lp082 group. Trevisin et al.
[23] found that MPO caused UC by producing cytokines and hypochlorite and that
MPO in the colon of UC patients is mainly produced by neutrophil infiltration [24].
Interestingly, this is consistent with the fact that the DSS group had a severe
neutrophil infiltration in this study. However, neutrophil infiltration and MPO content
were significantly decreased in Lp082 group. This shows that Lp082 alleviates UC by
reducing neutrophil infiltration and its secreted MPO content. In a nutshell, our results
suggest that Lp082 may play an anti-UC effect by inhibiting the NF- κ B pathway,
down-regulating pro-inflammatory cytokines, and up-regulating anti-inflammatory
cytokines, reducing MPO content, thereby maintaining immune balance and
protecting the immune barrier.

The mucosal immune system of the intestine mainly consists of Peyer's patch
and lamina propria under enterocyte [25]. The Peyer's patch can deliver captured
antigens to dendritic cells [26]. Then dendritic cells can not only trigger T
cell-mediated cellular immunity and B cell-mediated humoral immunity by presenting
antigens but also affect lamina propria immunity [27]. Combining previous studies,

we found that DSS causes inflammation through the following six ways. First, gut
permeability increases, and harmful substances enter to activate innate immunity, such
as stimulating innate immune cells to produce TNF- α , IL-1 β , and IL-6 [28]. Second,
regulatory T cells produce less IL-10 and have a less inhibitory effect on effector T
cells, resulting in the phenomenon of effector T and regulatory T cell dysregulation in
UC patients [29]. Third, effector T cells promote B cell-mediated humoral immunity
by promoting the secretion of IFN- γ and L-17A [30]. Fourth, effector T cells carried
out immune cell recruitment and formed a vicious immune cycle with chemokines
and cytokines [31]. Fifth, Peyer's patch recognizes antigens and presents them to other
immune cells through dendritic cells [26]. Sixth, antigen-activated neutrophils can
both secrete MPO and recruit more immune cells from the bloodstream to the site of
inflammation, further exacerbating inflammation [32] (**Fig. 6b**). Based on the above 6
reasons, we suggest that in addition to relieving inflammation by inhibiting the NF- κ B
pathway, Lp082 can also regulate inflammatory factors to maintain the balance
between regulatory T cells and effector T cells to regulate intestinal mucosal
immunity, thus maintaining the intestinal mucosal barrier.

Lp082 improved the biological barrier

Numerous studies [23] have shown that probiotics improve the clinical outcome
of IBD patients by influencing host gut microbiota [4]. Herein, we performed a
shotgun metagenomic analysis to investigate whether Lp082 can improve gut
dysbiosis in the UC mice model. As expected, we observed that the intake of DSS
significantly reduced the shannon value but increased PCoA distance, a finding that is
consistent with Wang et al. [33]. The Shannon index reflects gut microbiota richness
and uniformity and is positively correlated with gut microbiota diversity, while the
PCoA distance reflects the difference in the structure of the gut microbiota between
different groups; the higher the PCoA value, the greater the difference in the gut
microbiota structure [34]. In particular, Lp082 treatment remarkably increased the gut
microbiota diversity and reduced gut microbiota structural differences in gut
microbiota, as shown by the cluster analysis and PCoA analysis. On the other hand,
Lp082 also optimized species composition; that is, the abundance of

pro-inflammatory microbiota decreased in the Lp082 group, such as *Helicobacter*
*hepaticus*, a potential pathogen of colitis. Likewise, we observed an increasing trend
in the abundance of potential probiotics in the Lp082 group, such as *Bifidobacterium*
*pseudolongum* and *Bacteroides ovatus*, which reduces colonic inflammation [35],
*Parabacteroides distasonis*, which is negatively associated with obesity and diabetes
[36], *Akkermansia muciniphila* and *Lactobacillus reuteri*, a widely studied probiotic,
*Anaerotruncus sp G3 2012* and *Lactobacillus plantarum*, potential SCFAs-producing
bacteria [37]. The above results indicate that Lp082 is beneficial to optimizing the
diversity, structure, and composition of gut microbiota. After demonstrating that
Lp082 can increase the abundance of potential SCFAs-producing bacteria, further
analysis found that Lp082 can activate two SCFAs-producing microbial metabolic
pathways and the content of SCFAs. Subsequently, correlation analysis proved that
Lp082 may increase SCFAs by activating the SCFAs-producing metabolic pathway of
SCFAs-producing bacteria, so as to inhibit inflammation [38] and regulate host
physiological activity through SCFAs [39]. All of these suggest that Lp082 repaired
the microbial barrier by regulating the gut microbiome.

In conclusions, the Lp082 has an exciting therapeutic effect on UC than SASP.
Also, shotgun metagenome and transcriptome analysis confirmed that Lp082 could
improve gut microbiota dysbiosis, protect intestinal mucosal barrier, regulate
inflammatory pathways, and affect neutrophil infiltration. These findings firmly
support and advocate the clinical translation of Lp082 in the treatment of UC. It can
be suggested that the application of gut microbiota and probiotics in the treatment of
UC should receive more attention. The findings of this study not only provide new
clues for revealing the complex mechanism of gut microbiota in relieving UC, but
also provide evidence for Lp082 as a potential gut microbiota regulator to treat UC.

**References**

- 1. Li XX, Wei B, Goodglick L, Wen T, Xia LJ, Braun J. Investigating Therapeutic
Approach of IBD Using Recombinant Glycoprotein Mucin2. *Faseb Journal*. 2009;23.
- 2. Shi JL, Xie QG, Yue YX, Chen QX, Zhao LN, Evivie SE, et al. Gut microbiota
modulation and anti-inflammatory properties of mixed lactobacilli in dextran sodium

sulfate-induced colitis in mice. *Food & Function*. 2021;12(11):5130-43; doi:
10.1039/d1fo00317h.

3. Hu LH, Liu JY, Yin JB. Eriodictyol attenuates TNBS-induced UC through
repressing TLR4/NF-kB signaling pathway in rats. *Kaohsiung Journal of Medical*
*Sciences*. 2021;37(9):812-8; doi: 10.1002/kjm2.12400.

4. Wang LA, Gao MX, Kang GB, Huang H. The Potential Role of Phytonutrients
Flavonoids Influencing Gut Microbiota in the Prophylaxis and Treatment of
Inflammatory Bowel Disease. *Frontiers in Nutrition*. 2021;8; doi:
10.3389/fnut.2021.798038.

5. Fedorak RN. Understanding why probiotic therapies can be effective in treating
IBD. *Journal of clinical gastroenterology*. 2008;42 Suppl 3 Pt 1:S111-5; doi:
10.1097/MCG.0b013e31816d922c.

6. Sun MY, Liu YJ, Song YL, Gao Y, Zhao FJZ, Luo YH, et al. The ameliorative
effect of *Lactobacillus plantarum*-12 on DSS-induced murine colitis. *Food & Function*.
2020;11(6):5205-22; doi: 10.1039/d0fo00007h.

7. Burger-van Paassen N, Vincent A, Puiman PJ, van der Sluis M, Bouma J, Boehm
G, et al. Regulation of the intestinal mucin MUC2 expression by short chain fatty
acids: implications for epithelial protection. *Faseb Journal*. 2009;23.

8. Taniguchi T, Tsukada H, Nakamura H, Kodama M, Fukuda K, Saito T, et al.
Effects of the anti-ICAM-1 monoclonal antibody on dextran sodium sulphate-induced
colitis in rats. *Journal of gastroenterology and hepatology*. 1998;13(9):945-9; doi:
10.1111/j.1440-1746.1998.tb00766.x.

9. Philpott JR, Miner PB, Jr. Antisense inhibition of ICAM-1 expression as therapy
provides insight into basic inflammatory pathways through early experiences in IBD.
*Expert Opin Biol Ther*. 2008;8(10):1627-32; doi: 10.1517/14712598.8.10.1627.

10. Edelblum KL, Turner JR. The tight junction in inflammatory disease:
communication breakdown. *Current Opinion in Pharmacology*. 2009;9(6):715-20; doi:
10.1016/j.coph.2009.06.022.

11. Stio M, Retico L, Annese V, Bonanomi AG. Vitamin D regulates the
tight-junction protein expression in active UC. *Scandinavian Journal of*

Gastroenterology. 2016;51(10):1193-9; doi: 10.1080/00365521.2016.1185463.

12. Cordeiro BF, Alves JL, Belo GA, Oliveira ER, Braga MP, da Silva SH, et al.
Therapeutic Effects of Probiotic Minas Frescal Cheese on the Attenuation of UC in a
Murine Model. *Frontiers in Microbiology*. 2021;12; doi: 10.3389/fmicb.2021.623920.

13. Nakamura S, Ohtani H, Watanabe Y, Fukushima K, Matsumoto T, Kitano A, et al.
In situ expression of the cell adhesion molecules in inflammatory bowel disease.
Evidence of immunologic activation of vascular endothelial cells. *Laboratory*
*investigation; a journal of technical methods and pathology*. 1993;69(1):77-85.

14. Mitselou A, Grammeniatis V, Varouksi A, Papadatos SS, Klaroudas A, Katsanos
2473 K, et al. Immunohistochemical Study of Adhesion Molecules in Irritable Bowel
Syndrome: A Comparison to Inflammatory Bowel Diseases. *Advanced biomedical*
*research*. 2021;10:21; doi: 10.4103/abr.abr_2_20.

15. Ruco LP, de Laat PA, Matteucci C, Bernasconi S, Sciacca FM, van der Kwast TH,
et al. Expression of ICAM-1 and VCAM-1 in human malignant mesothelioma. *The*
*Journal of pathology*. 1996;179(3):266-71.

16. Bauer J, Namineni S, Reisinger F, Zoller J, Yuan DT, Heikenwalder M.
Lymphotoxin, NF-kappa B, and Cancer: The Dark Side of Cytokines. *Digestive*
*Diseases*. 2012;30(5):453-68; doi: 10.1159/000341690.

17. Xiao B, Laroui H, Ayyadurai S, Viennois E, Charania MA, Zhang YC, et al.
Mannosylated bioreducible nanoparticle-mediated macrophage-specific TNF-alpha
RNA interference for IBD therapy. *Biomaterials*. 2013;34(30):7471-82; doi:
10.1016/j.biomaterials.2013.06.008.

18. 김연하, 김유림, 김성중, 황호근, Choi S-C, 김경숙, et al. Rebamipide Protects
Colonic Damage Induced by Trinitrobenzene Sulfonic Acid (TNBS) via
Down-Regulation of TNF- α , IL-1 β , and ICAM-1. *Anatomy and Cell Biology*.
2004;37(2):149-56.

19. Zhou X, Liu H, Zhang J, Mu J, Zalan Z, Hegyi F, et al. Protective effect of
*Lactobacillus fermentum* CQPC04 on dextran sulfate sodium-induced colitis in mice
is associated with modulation of the nuclear factor-kappa B signaling pathway.
*Journal of Dairy Science*. 2019;102(11):9570-85; doi: 10.3168/jds.2019-16840.

- 20. Mohammadnia-Afrouzi M, Hosseini AZ, Khalili A, Abediankenari S, Amari A,
Aghili B, et al. Altered microRNA Expression and Immunosuppressive Cytokine
Production by Regulatory T Cells of UC Patients. *Immunological Investigations*.
2016;45(1):63-74; doi: 10.3109/08820139.2015.1103749.
- 21. Sato Y, Takahashi S, Kinouchi Y, Shiraki M, Endo K, Matsumura Y, et al. IL-10
deficiency leads to somatic mutations in a model of IBD. *Carcinogenesis*.
2006;27(5):1068-73.
- 22. Hume GE, Fowler EV, Lincoln D, Eri R, Templeton D, Florin TH, et al.
Angiotensinogen and transforming growth factor beta1: novel genes in the
pathogenesis of Crohn's disease. *Journal of medical genetics*. 2006;43(10):e51; doi:
10.1136/jmg.2005.040477.
- 23. Trevisin M, Pollock W, Dimech W, Savige J. Evaluation of a multiplex flow
cytometric immunoassay to detect PR3- and MPO-ANCA in active and treated
vasculitis, and in inflammatory bowel disease (IBD). *Journal of immunological*
*methods*. 2008;336(2):104-12; doi: 10.1016/j.jim.2008.03.012.
- 24. Chin AC, Parkos CA. Neutrophil transepithelial migration and epithelial barrier
function in IBD: potential targets for inhibiting neutrophil trafficking. *Annals of the*
*New York Academy of Sciences*. 2006;1072:276-87; doi: 10.1196/annals.1326.018.
- 25. Jonker MA, Hermsen JL, Sano Y, Heneghan AF, Lan JG, Kudsk KA. Small
intestine mucosal immune system response to injury and the impact of parenteral
nutrition. *Surgery*. 2012;151(2):278-86; doi: 10.1016/j.surg.2010.10.013.
- 26. Li HS, Gelbard A, Martinez GJ, Esashi E, Zhang HY, Nguyen-Jackson H, et al.
Cell-intrinsic role for IFN-alpha-STAT1 signals in regulating murine Peyer patch
plasmacytoid dendritic cells and conditioning an inflammatory response. *Blood*.
2011;118(14):3879-89; doi: 10.1182/blood-2011-04-349761.
- 27. Santucci L, Agostini M, Bruscoli S, Mencarelli A, Ronchetti S, Ayroldi E, et al.
GITR modulates innate and adaptive mucosal immunity during the development of
experimental colitis in mice. *Gut*. 2007;56(1):52-60; doi: 10.1136/gut.2006.091181.
- 28. Debnath T, Kim DH, Lim BO. Natural Products as a Source of
Anti-Inflammatory Agents Associated with Inflammatory Bowel Disease. *Molecules*.

2013;18(6):7253-70; doi: 10.3390/molecules18067253.

29. Goldberg R, Scotta C, Cooper D, Nissim-Eliraz E, Nir E, Tasker S, et al.
Correction of Defective T-Regulatory Cells From Patients With Crohn's Disease by
Ex Vivo Ligation of Retinoic Acid Receptor-alpha. *Gastroenterology*.
2019;156(6):1775-87; doi: 10.1053/j.gastro.2019.01.025.

30. Cook L, Stahl M, Han X, Nazli A, MacDonald KN, Wong MQ, et al. Suppressive
and Gut-Reparative Functions of Human Type 1 T Regulatory Cells. *Gastroenterology*.
2019;157(6):1584-98; doi: 10.1053/j.gastro.2019.09.002.

31. Singh UP, Singh NP, Murphy EA, Price RL, Fayad R, Nagarkatti M, et al.
Chemokine and cytokine levels in inflammatory bowel disease patients. *Cytokine*.
2016;77:44-9; doi: 10.1016/j.cyto.2015.10.008.

32. Zhou GX, Yu L, Fang LL, Yang WJ, Yu TM, Miao YL, et al. CD177(+)
neutrophils as functionally activated neutrophils negatively regulate IBD. *Gut*.
2018;67(6):1052-63; doi: 10.1136/gutjnl-2016-313535.

33. Wang R, Chen T, Wang Q, Yuan XM, Duan ZL, Feng ZY, et al. Total Flavone of
*Abelmoschus manihot* Ameliorates Stress-Induced Microbial Alterations Drive
Intestinal Barrier Injury in DSS Colitis. *Drug Design Development and Therapy*.
2021;15:2999-3016; doi: 10.2147/dddt.S313150.

34. Suchodolski JS, Markel ME, Garcia-Mazcorro JF, Unterer S, Heilmann RM,
Dowd SE, et al. The Fecal Microbiome in Dogs with Acute Diarrhea and Idiopathic
Inflammatory Bowel Disease. *Plos One*. 2012;7(12); doi:
10.1371/journal.pone.0051907.

35. Yang C, Du Y, Ren D, Yang X, Zhao Y. Gut microbiota-dependent catabolites of
tryptophan play a predominant role in the protective effects of turmeric
polysaccharides against DSS-induced UC. *Food Funct*. 2021;12(20):9793-807; doi:
10.1039/d1fo01468d.

36. Cai W, Xu JX, Li G, Liu T, Guo XL, Wang HJ, et al. Ethanol extract of propolis
prevents high-fat diet-induced insulin resistance and obesity in association with
modulation of gut microbiota in mice. *Food Research International*. 2020;130; doi:
10.1016/j.foodres.2019.108939.

37. Wang J, Ji HF, Wang SX, Liu H, Zhang W, Zhang DY, et al. Probiotic
*lactobacillus plantarum* Promotes Intestinal Barrier Function by Strengthening the
Epithelium and Modulating Gut Microbiota. *Frontiers in Microbiology*. 2018;9; doi:
10.3389/fmicb.2018.01953.

38. Wang SL, Zhang SY, Huang SM, Wu ZH, Pang JM, Wu YJ, et al. Resistant
Maltodextrin Alleviates Dextran Sulfate Sodium-Induced Intestinal Inflammatory
Injury by Increasing Butyric Acid to Inhibit Proinflammatory Cytokine Levels.
*Biomed Research International*. 2020;2020; doi: 10.1155/2020/7694734.

39. Holota Y, Dovbynchuk T, Kaji I, Vareniuk I, Dzyubenko N, Chervinska T, et al.
The long-term consequences of antibiotic therapy: Role of colonic short-chain fatty
acids (SCFAs) system and intestinal barrier integrity. *Plos One*. 2019;14(8); doi:
10.1371/journal.pone.0220642.

**Minor comments:**

1. Please provide line numbering.

**Response:** We are grateful to the reviewer for pointing out this problem. We are very
sorry for our negligence with page numbers and line numbers. We have added the
page number and line number to the article. The title page is also called page 1, and
the first line of the title is line 1.

2. Figure 1a depicted the study design and methodology, which might be better to
merge into M&M part.

**Response:** We appreciate your valuable and helpful comment. Thank you for pointing
out this problem. We deeply agree with the reviewer's opinion on this problem, and
we have moved the content of this part to M&M. The changes in the text are
highlighted in yellow. (Page 24, line: 676-681)

3. Information of study design and methodology are not appropriate present in Results
section. The tables or figures should be displayed at a consecutive and sequential
order. In current version figure S1b appeared ahead of S1a.

**Response:** We appreciate your valuable and helpful comment. We have corrected this
problem and redescribed this part to make the article more coherent, and the rewritten
content is as follows: (Page 7, line: 166-200)

In DSS-induced UC mice, the immune organ index gradually increased and the colon
length gradually shortened with increasing disease severity [1]. Therefore, we
measured the spleen, liver, kidney, and colon of the mice. The results showed that the
immune organ index of the DSS group was significantly increased ($p < 0.05$), and the
immune organ index was significantly decreased after Lp082 intake ($p < 0.05$) (**Fig.**
**1c**). The colon length of the mice in the DSS group was significantly decreased ($p <$
0.05), and the colon length in Lp082 group was significantly increased ($p < 0.05$) (**Fig.**
**1d**). In addition, we also observed that the intestinal contents of the colitis mice in the
DSS group were loose, unformed and there was blood in the intestinal lumen, while
the intestinal contents in the Lp082 and Control groups were clear particles, hard stool,
and no blood (**Fig. 1d**). The fecal morphology of the intestinal contents was similar to
the results observed in mouse feces on the buttocks of mice. The feces of the mice in
the DSS group were blood-red, and the feces were loose and unformed, while there
was no blood in the feces after Lp082 ingestion (**Fig. S1 a**).

With the increase of disease degree, DSS-induced UC mice will have a worse mental
state, even abdominal pain, arch back, panic and other symptoms [2]. The mental state
of the mice was observed daily, and the results are shown in **Figure S1 b**. On the 7th
2604 day of modeling, mice in the control group were in a normal state, with normal urine
and feces, shiny hair, active spirit, sensitive reaction, and increased body size.
However, mice in the BCD group had yellow and smelly urine, difficult defecation,
bloody stool, dark and fried hair, slow reaction and easy panic, arched back, and
reduced body size (**Fig. S1 b**). On the last day of treatment(Day 15), compared with
the arched back, retarded response, hematochezia, and lethargic in the DSS group, the
mental state of mice in the Lp082 and SASP groups gradually returned to normal,
with an active spirit, no arched back, no hematochezia and shiny hair (**Fig. S1 b**).
These results indicated that Lp082 intake could alleviate the symptoms of depression,
crouching, and untidy hair of mice in the DSS group in the middle and late stage of

the experiment (**Fig. S1 b**).

Studies have shown that under the condition of inflammation, the spleen of mice
induced by DSS will increase hyperemia and even appear infection blackening.
Therefore, we looked at the spleens of mice and found that the spleens of mice in the
DSS group were significantly larger and darker than those of mice in the normal
group. The spleens of mice in the Lp082 and SASP groups were smaller and redder
rather than black than those in the DSS group (**Fig. S1 c**).

**Reference**

- 1. Rodriguez-Nogales A, Algieri F, Garrido-Mesa J, Vezza T, Pilar Utrilla M,
Chueca N, et al. Differential intestinal anti-inflammatory effects of *Lactobacillus*
*fermentum* and *Lactobacillus salivarius* in DSS mouse colitis: impact on microRNAs
expression and microbiota composition. *Molecular Nutrition & Food Research*.
2017;61(11); doi: 10.1002/mnfr.201700144.
- 2. Sun MY, Liu YJ, Song YL, Gao Y, Zhao FJZ, Luo YH, et al. The ameliorative
effect of *lactobacillus plantarum*-12 on DSS-induced murine colitis. *Food & Function*.
2020;11(6):5205-22; doi: 10.1039/d0fo00007h.

Once again, we thank you for the time you put into reviewing our paper, and we are
very grateful for your effort in reviewing our paper and your positive feedback. The
summary of our work as written by you is precise. Since your inputs have been
precious, we would like to acknowledge your contribution explicitly in the eventuality
of a publication.

October 7, 2022

Prof. Jiachao Zhang
Hainan University
Food Science
58 renmin road
Haikou, Hainan 570228
China

Re: Spectrum01651-22R1 (Probiotics (*Lactobacillus plantarum* HNU082) supplementation relieves ulcerative colitis by affecting intestinal barrier functions, immunity-related genes expression, gut microbiota, and metabolic pathways in mice)

Dear Prof. Jiachao Zhang:

Thank you for submitting your manuscript to Microbiology Spectrum. As you will see your paper is very close to acceptance. Please modify the manuscript along the lines the reviewer has recommended. As these revisions are quite minor, I expect that you should be able to turn in the revised paper in less than 30 days, if not sooner. If your manuscript was reviewed, you will find the reviewers' comments below.

When submitting the revised version of your paper, please provide (1) point-by-point responses to the issues raised by the reviewers as file type "Response to Reviewers," not in your cover letter, and (2) a PDF file that indicates the changes from the original submission (by highlighting or underlining the changes) as file type "Marked Up Manuscript - For Review Only". Please use this link to submit your revised manuscript. Detailed instructions on submitting your revised paper are below.

Link Not Available

Sincerely,

Xiaoyu Tang

Reviewer comments:

Reviewer #2 (Comments for the Author):

The manuscript has been improved a lot, please fix the following.

1. In results, the title of each section should be same as the line 145 that show a specific conclusion.
2. Experiment details should not be appeared in "Result sections".
3. In Results and Discussion, the author should be described the results more concisely, rather than a repetitive description. For example, Fig.S1a should be a part of the Disease Activity Index (DAI) score and so on. Please reorganize the description in both sections.
4. In Fig 5a, the data should be better presented regarding up-regulated genes and down-regulated genes involved in metabolic pathway, respectively.
5. In discussion, the creativity of manuscript should be noted compared with the similarity studies which published before.

Preparing Revision Guidelines

Please return the manuscript within 60 days; if you cannot complete the modification within this time period, please contact me. If you do not wish to modify the manuscript and prefer to submit it to another journal, please notify me of your decision immediately so that the manuscript may be formally withdrawn from consideration by Microbiology Spectrum.

**Manuscript No.: Spectrum 01651-22**

**Title: Probiotics (*Lactobacillus plantarum* HNU082) supplementation relieves**
**ulcerative colitis by affecting intestinal barrier functions, immunity-related genes**
**expression, gut microbiota, and metabolic pathways in mice.**

**Dear Dr. Xiaoyu Tang,**

I am very glad to receive your email again! On behalf of my co-authors, I thank
you very much for allowing us to revise our manuscript. We appreciate the time and
effort that you and the reviewers dedicated to providing feedback on our manuscript
and are grateful for the insightful comments on and valuable improvements to our
manuscript. We have discussed reviewer's comments carefully and revised the
manuscript taking all the comments positively. All revisions in the manuscript have
been highlighted in yellow. Please find the point-to-point responses to reviewers'
comments in the following text. We thoroughly double-checked the manuscript. In
addition, the revised manuscript with tracked changes is also uploaded as "Marked Up
Manuscript" files.

We sincerely hope that this revised manuscript will be published in "*Microbiology*
*Spectrum*." We deeply appreciate your consideration of our manuscript. If you have
any queries, please don't hesitate to contact us at the following e-mail address.

We would like to express our great appreciation again to you and the reviewers for
their comments on our paper. We are looking forward to hearing from you.

Sincerely,

Jiachao Zhang

Yours sincerely,

E-mail: Jiachao Zhang1*, zhjch321123@163.com

College of Food Science and Engineering, Hainan University, Haikou 570228, China

**Responds to the reviewer's comments**

Reviewer #2 (Comments for the Author):

The manuscript has been improved a lot, please fix the following.

**Response:** We appreciate the time and effort you dedicated to providing feedback on
our manuscript and are grateful for the insightful comments and valuable
improvements to our manuscript. We have discussed your comments carefully, and we
sincerely accept the suggestions. Your comments provided valuable insights to refine
its contents and analysis. In this document, we try to address the issues raised as best
as possible. All revisions in the manuscript have been highlighted in yellow. A list of
changes to the manuscript has been attached, and you can kindly find the
point-to-point responses to your comments in the following text.

1. In results, the title of each section should be same as the line 145 that show a
specific conclusion.

**Response:** We appreciate your valuable and helpful comment and we deeply agree
with the opinions of reviewer. According to your helpful suggestions, we have
rewritten the title of each section in results, and we have also improved the title of the
conclusion. We sincerely thank you again for pointing this out. It was very helpful.
The changes have been highlighted in the manuscript in yellow. And the revised
content is as follows.

The intake of Lp082 alleviated physiological lesions in DSS-induced colitis mice
(Page 6, line:145)

The intake of Lp082 up-regulated the anti-inflammatory cytokines and
down-regulated the pro-inflammatory cytokines in DSS-induced colitis mice
(Page 7, line:192-193)

The intake of Lp082 alleviated pathological lesions in DSS-induced colitis mice
(Page 8, line: 203)

The intake of Lp082 regulated the gut microbiota in DSS-induced colitis mice
(Page 9, line: 238)

The intake of Lp082 regulated the short chain fatty acid in DSS-induced colitis mice
(Page 10, line: 265-266)

The intake of Lp082 regulated the transcriptome of intestinal epithelial cells in
DSS-induced colitis mice
(Page 12, line: 328-329)

The potential mechanism of Lp082 alleviated the DSS-induced colitis
(Page 14, line: 398)

The intake of Lp082 improved the chemical barrier
(Page 16, line: 449)

The intake of Lp082 improved the mechanical barrier
(Page 17, line: 482)

The intake of Lp082 improved the immune barrier
(Page 18, line: 513)

The intake of Lp082 improved the biological barrier
(Page 20, line: 576)

2. Experiment details should not be appeared in "Result sections".

**Response:** We are grateful to the reviewer for pointing out this problem. We deeply
agree with the opinions of reviewer. We are very sorry for our negligence and we
sincerely apologize for the inconvenience caused to you. According to your helpful
suggestions, we have moved the contents of the experimental details appeared in
"Result" sections to the "Materials and methods" section, and we have rewritten the
relevant content in the results section. We have carefully checked and verified the
contents of the "Result" section again. The changes have been highlighted in the
manuscript in yellow. And the revised content is as follows. We sincerely thank you
again for pointing this out. It was very helpful.

To further evaluate colon injury, we quantified the pro-inflammatory cytokines
interleukin-1beta (IL-1 β), interleukin-6 (IL-6), interferon-gamma (IFN- γ), tumor
necrosis factor-alpha (TNF- α), and myeloperoxidase (MPO), and anti-inflammatory
cytokines interleukin-10 (IL-10) in serum of 6 mice in each group. The results showed
that compared with the control group, the pro-inflammatory cytokines TNF-, IL-1 β ,
IFN- α , IL-6, and MPO in DSS group were significantly increased ($p < 0.05$), while
the anti-inflammatory cytokines IL-10 were significantly decreased ($p < 0.05$), while
the opposite was observed in Lp082 and SASP groups (Fig. 1e). (Page 7, line:
194-201)

The results of Shotgun metagenomic data diversity analysis demonstrated the effect of
Lp082 on the diversity of intestinal microbiota in mice. The results of α diversity
analysis showed that on days 1 - 7 of the study, the Shannon index in DSS, Lp082,
and SASP groups were all significantly decreased (Fig. 3a) , but the Shannon index
was significantly increased after the intake of Lp082 ($p < 0.05$) (Fig. 3a). The results
of β diversity analysis showed that the DSS group, LP082 group and SASP group
(M_B, M_C, M_D) and control group (M_A) were significantly separated on day 7 (p
< 0.05) (Fig. 3b). However, on day 15, the DSS group was still significantly separated
from the control group (T_B), while the distance between Lp082 group (T_C), SASP
group (T_D), and control group (T_A) was significantly reduced (p values < 0.05),

and the distance between Lp082 group and control group was closer, the above results
were consistent with the principal co-ordinates analysis (PCoA) distance results (Fig.
3c). The above diversity analysis results showed that Lp082 increased the α
-diversity and optimized the β -diversity of cecal microbiota in mice. (Page 9, line:
239-252)

Gene distribution was analyzed using colonic transcriptome data, the volcano map the
results show that Lp082 significantly affected gene expression distribution (Fig. S5
a-f). To further explore the impact of these differentially expressed genes (DEGs), we
analyzed the pathways involved in DEGs. (Page 12, line: 330-333)

At the end of the experiment, we euthanized the mice , and the 1cm portion of the
distal colon of 6 mice in each group was randomly selected for HE staining, and
histopathological score and intestinal wall thickness were further measured (n=6).
(Page 23, line: 674-676)

Six mice were randomly selected at two time points for metagenomic sequencing of
feces. At the end of modeling (day 7 of the experiment), feces of 6 mice in each group
were randomly selected for metagenomic sequencing. At the end of treatment (day 15
of the experiment), feces of 6 mice in each group were randomly selected for
metagenomic sequencing, to observe the effects of DSS and Lp082 on the intestinal
microecology of mice. (Page 24, line: 706-711)

At the end of the experiment, 6 mice from each group were randomly selected for
colon transcriptome RNA sequencing, and the volcanic map was drawn based on the
preliminary gene distribution analysis results. The sequencing was performed by
Beijing Novogene Co., Ltd. (Beijing, China). The RNA extraction mini kit (Qiagen,
Hilden, Germany) was used for total RNA extraction from the mouse colon samples,
and NanoDrop 2000 was used for quantification. Then the library construction and the
quality control were carried on, and the raw RNA-seq data was filtered [1]. After

constructing the RNA library, Illumina Novaseq 6000 was used for sequencing, and
the FeatureCounts were used to estimate the gene expression [2]. (Page 26, line:
739-747)

**Reference**

1. Dobin A, Davis CA, Schlesinger F, Drenkow J, Zaleski C, Jha S, et al. STAR:
ultrafast universal RNA-seq aligner. *Bioinformatics*. 2013;29(1):15-21; doi:
10.1093/bioinformatics/bts635.

2. Liao Y, Smyth GK, Shi W. featureCounts: an efficient general purpose program for
assigning sequence reads to genomic features. *Bioinformatics*. 2014;30(7):923-30;
doi: 10.1093/bioinformatics/btt656.

3. In Results and Discussion, the author should be described the results more
concisely, rather than a repetitive description. For example, Fig.S1a should be a part
of the Disease Activity Index (DAI) score and so on. Please reorganize the description
in both sections.

**Response:** We appreciate your valuable and helpful comment. We apologize for the
language problems in the original manuscript. We sincerely apologize for the
confusion caused to you. The language presentation was improved with assistance
from a native English speaker with appropriate research background. We deeply and
sincerely agree with you that Fig. S1a should indeed be part of the Disease Activity
Index (DAI) score, we have put the two parts of the description together and
reorganize the description. In addition, according to your helpful suggestions, We
have rewritten the relevant content of the results and discussion section, and have
described the results in more concise language, deleted the repeated description, and
deepened the discussion. The changes have been highlighted in the manuscript in
yellow. And the revised content is as follows.

People with UC have a disorder of colon function, poor absorption, loss of appetite,
weight loss, diarrhea, and bloody stools [8]. Therefore, the lower the body weight, the

lower the amount of water and food intake, and the higher the disease activity index
(DAI) score (The scoring criteria is shown in TABLE S1), indicating the more severe
enteritis. (Page 6, line: 146-150)

From 1 to 7 days, the water intake, food intake, and body weight of the DSS group,
the Lp082 group, and the SASP group all showed a similar degree of gradual decrease,
which may be because these three groups were all under the same DSS modeling
conditions on days 0-7. Then on the 8th to 15th day, the water intake, food intake, and
body weight of the DSS group were still decreasing, but the water intake, food intake,
and body weight of Lp082 and SASP group gradually increased. However, the water
and food intake of the Lp082 combined SASP group increased significantly from day
9 ($p < 0.05$), and body weight increased significantly from day 12 ($p < 0.05$). (Page 6,
line: 151-158)

The DAI index of the DSS group, Lp082 group, and SASP group increased
significantly ($p < 0.05$) since the third day compared with the Control group. But after
stopping DSS gavage on the 8th day, the DAI index of the DSS self-healing group
still increased, while that of the Lp082 group and SASP group gradually decreased
from the 10th day. And the degree of decrease in the Lp082 group was greater than
that in the SASP group, indicating that Lp082 had a better improvement effect on DAI
index (Fig. 1b). In addition, we observe that the feces of the mice in the DSS group
were blood-red, but there was no blood in the feces after Lp082 and SASP ingestion
(Fig. S1 a). This phenomenon is consistent with the measurement results of DAI
index. (Page 6, line: 159-168)

An increase in immune organ index and a decrease in colon length indicate an
increase in inflammation [2]. The results showed that the immune organ index of the
DSS group was significantly increased ($p < 0.05$), but was significantly decreased
after Lp082 intake ($p < 0.05$) (Fig. 1c). And the colon length of the mice in the DSS
group was significantly decreased ($p < 0.05$), but was significantly increased after

Lp082 intake ($p < 0.05$) (Fig. 1d). (Page 6, line: 169-174)

Studies have shown that DSS-induced UC mice will have a worse mental state, even
abdominal pain, arch back, panic and other symptoms with the increase of disease
degree, and the spleen will also increase hyperemia and infection blackening [30].
After successful modeling of UC, we observed that the mice in the control group were
in a normal state, with normal urine and feces, shiny hair, active spirit, sensitive
reaction, and increased body size. However, mice in the DSS, Lp082 and SASP
groups had yellow and smelly urine, difficult defecation, bloody stool, dark and fried
hair, slow reaction and easy panic, arched back, and reduced body size (Fig. S1 b). On
the last day of treatment (Day 15), the mental state of the DSS mice was still poor, but
the mental state of mice in the Lp082 and SASP groups gradually returned to normal,
with an active spirit, no arched back, no hematochezia and shiny hair (Fig. S1 b). In
addition, we found that the spleens of mice in the DSS group were significantly larger
and darker than those of mice in the normal group, but the spleen gradually returned
to normal in size and color after the Lp082 and SASP intake. (Fig. S1 c). (Page 7, line:
175-188)

The results of Shotgun metagenomic data diversity analysis demonstrated the effect of
Lp082 on the diversity of intestinal microbiota in mice. The results of α diversity
analysis showed that on days 1 - 7 of the study, the Shannon index in DSS, Lp082,
and SASP groups were all significantly decreased (Fig. 3a) , but the Shannon index
was significantly increased after the intake of Lp082 ($p < 0.05$) (Fig. 3a). The results
of β diversity analysis showed that the DSS group, LP082 group and SASP group
(M_B, M_C, M_D) and control group (M_A) were significantly separated on day 7 (p
< 0.05) (Fig. 3b). However, on day 15, the DSS group was still significantly separated
from the control group (T_B), while the distance between Lp082 group (T_C), SASP
group (T_D), and control group (T_A) was significantly reduced (p values < 0.05),
and the distance between Lp082 group and control group was closer, the above results
were consistent with the principal co-ordinates analysis (PCoA) distance results (Fig.

3c). The above diversity analysis results showed that Lp082 increased the α
-diversity and optimized the β -diversity of cecal microbiota in mice. (Page 9, line:
239-252)

Gene distribution was analyzed using colonic transcriptome data, the volcano map the
results show that Lp082 significantly affected gene expression distribution (Fig. S5
a-f). To further explore the impact of these differentially expressed genes (DEGs), we
analyzed the pathways involved in DEGs. (Page 12, line: 330-333)

Reference

- 1. Costello SP, Hughes PA, Waters O, Bryant RV, Vincent AD, Blatchford P, et al.
Effect of Fecal Microbiota Transplantation on 8-Week Remission in Patients With
Ulcerative Colitis A Randomized Clinical Trial. *Jama-Journal of the American*
*Medical Association*. 2019;321(2):156-64; doi: 10.1001/jama.2018.20046.
- 2. Rodriguez-Nogales A, Algieri F, Garrido-Mesa J, Vezza T, Pilar Utrilla M, Chueca
251 N, et al. Differential intestinal anti-inflammatory effects of *Lactobacillus*
*fermentum* and *Lactobacillus salivarius* in DSS mouse colitis: impact on
microRNAs expression and microbiota composition. *Molecular Nutrition & Food*
*Research*. 2017;61(11); doi: 10.1002/mnfr.201700144.
- 3. Sun MY, Liu YJ, Song YL, Gao Y, Zhao FJZ, Luo YH, et al. The ameliorative
effect of *Lactobacillus plantarum*-12 on DSS-induced murine colitis. *Food &*
*Function*. 2020;11(6):5205-22; doi: 10.1039/d0fo00007h.

We sincerely thank you again for pointing this out. It was very helpful.

4. In Fig 5a, the data should be better presented regarding up-regulated genes and
down-regulated genes involved in metabolic pathway, respectively.

**Response:** We appreciate your valuable and helpful comment. We deeply and
sincerely understand the reviewer's idea. Fig. 5a is the results of Gene Ontology (GO)
enrichment analysis, GO can be divided into three categories, namely Biological

processes, Cellular Component and Molecular Function. In the initial analysis, I tried
to show the specific gene results and the up-regulation and down-regulation of
specific genes in the Gene Ontology pathway, but we did not do so in the end.
The reason we focus on the pathways in which genes are enriched, rather than the
genes in the pathways are as follows: By annotating the transcriptome data, we have a
volcanic map that reveals the distribution of gene expression and shows that the total
number of annotated genes is close to 20,000 (Fig. S5). There are so many genes that
it's too difficult for us to find rules among them. Through the investigation of
references [1], we found that a large number of disordered genes could be enriched
into a small number of pathways by gene enrichment analysis, so as to facilitate us to
explore the characteristics and rules between pathways. Gene enrichment analysis is a
common way to process a large amount of gene data, which can facilitate us to find
the rules among genes and GO enrichment analysis is one of the enrichment methods
[2]. The minimum value of GeneRatio of the GO term in Fig. 5a is 0.1, if the input
data used for enrichment analysis is assumed to be 1000 genes, then according to the
formula [3]: $\text{GeneRatio} = \frac{\text{the number of genes enriched to this GO term}}{\text{the number of all input genes used for enrichment analysis}}$, it can be concluded that the number of
genes enriched to the GO entry is 100 genes. There were 100 genes in one GO term,
1,000 genes in 10 GO terms. In fact, we calculated that the number of genes enriched
in a certain GO pathway was much greater than 100, because the number of
differentially expressed genes we input was much greater than 1000. That's why we
chose to analyze and present the pathway results, rather than listing every single gene
up-regulation and down-regulation in the pathway, because the amount of genetic data
is too large to find regular. Maza et al.[4] and Wang et al. [5] process a large number of
gene data through enrichment analysis, and finally find rules in pathway.

Our previous analysis idea was as follows: Since the preliminary analysis of
transcriptome data showed that the intake of Lp082 affects the gene expression
distribution (**Fig. S5**), in order to explore the relationship between a large number of
genes, we conducted GO pathway enrichment analysis and KEGG pathway
enrichment analysis for the differentially expressed genes (DEGs). Since the

differentially expressed genes (DEGs) were more enriched in the biological process
(BP) pathway among the three major GO pathway categories (**Fig. 5a-c**). And
compared with the DSS group, the number of significantly up-regulated genes in
Lp082 group is more than the down-regulated genes (**Fig. 5d**), so we performed
further GO-BP pathway enrichment analysis on the significantly up-regulated
differentially expressed genes (**Fig. 6d-6f**). Subsequently, we learned about some
genes that are abnormally expressed in inflammatory situations through literature,
analyzed the up-down regulation of these specific inflammatory genes, and found
similar rules in our data (**Fig. 6g-6i**). We have 6 biological replicates in each group,
and our data are realistic and objective enough to support our conclusion.

We appreciate your valuable and helpful comment again and we deeply agree
with the opinions of reviewer. We are deeply sorry for our not clear description.
According to your helpful suggestions, we have rewritten this part. The changes have
been highlighted in the manuscript in yellow. The rewritten content is more detailed,
and the details are as follows. (Page 12, line: 330-396)

Gene distribution was analyzed using colonic transcriptome data, the volcano
map the results show that Lp082 significantly affected gene expression distribution
(Fig. S5 a-f). To further explore the impact of these differentially expressed genes
(DEGs), we analyzed the pathways involved in DEGs.

Fig. 5a is the results of Gene Ontology (GO) enrichment analysis, GO can be
divided into three categories, namely Biological processes, Cellular Component and
Molecular Function. The results of gene ontology (GO) analysis (n=6) showed that
the DEGs of the DSS group and the control group were mainly involved in biological
processes such as the humoral immune response, activation of an immune response,
negative regulation of hemostasis; and cellular components such as blood
microparticle, membrane attack complex; and molecular functions such as lipid
binding, lipopolysaccharide-binding, thrombospondin receptor activity (Fig. 5a). On
the other hand, the DEG of the Lp082 and DSS groups was mainly involved in
biological processes such as blood coagulation, fibrin clot formation, regulation of
humoral immune markers, regulation of inflammatory cytokines; and cellular

components such as Golgi lumen, endoplasmic reticulum, and molecular functions
such as endopeptidase activity and peptidase activity (Fig. 5b).

Considering that in the Lp082, the up-regulated DEGs were far more than
down-regulated DEGs (Fig. S5 a-f), and the DEGs have the largest proportion of
participation in biological processes (Fig. 5a-5c), we further conducted GO-BP
analysis (n=6) on significantly up-regulated DEGs. The results of GO-BP analysis
showed that compared to control group, up-regulated DEGs in DSS group were
mainly enriched in the 6 inflammation-related GO-BP. Among those, the genes IL-1 β
and IL-1 α were both involved in the IL-1 β production and TNF production, the
oncogene Ereg were involved in the IL-1 β production, the genes IL-1 β and IL-1rn,
oncogene Fga were all involved in positive regulation of nuclear factor kappa-B
(NF- κ B) transcription factor activity, the oncogene Ldlr, Dgat2, and Mfsd2a were all
involved in the regulation of toll-like receptor 4 signaling pathway, the pro-oncogenes
Cdc7, Dbf4 were all involved in the acute inflammatory response, the anti-tumour
gene Syk and the inflammatory genes Nlrp3 as well as Syk were all involved in the
pro-inflammatory factor IL-6 production (Fig. 5d). Compared to DSS group, the
up-regulated genes in Lp082 group were mainly enriched in the 6
anti-inflammatory-related GO-BP. Among them, the gene Isg15, which exerted both
its antiviral and anti-inflammatory effects in innate immunity, and the gene Prg2,
which played an important role in wound healing, were involved in the
anti-inflammatory factors IL-10 production (Fig. 5e).

To further observe whether Lp082 treatment would suppress these inflammatory
and cancer genes enriched on inflammatory pathways in the DSS group, we
supplemented Fig. S6. As can be seen from Fig. S6, among the 13 inflammatory genes
or oncogenes that were up-regulated and enriched in the inflammatory pathway in the
DSS group, the following 10 genes were significantly down-regulated in the Lp082
group: IL-1 β , IL-1 α , Ereg, IL -1rn, Fga, Ldlr, Dgat2, Mfsd2a, Cdc7, Dbf4 (Fig. S6)

The results of kyoto encyclopedia of genes and genomes (KEGG) analysis (n=6)
showed that the DEGs in DSS and control groups were mainly enriched in systemic
lupus erythematosus, Staphylococcus aureus infection, Viral carcinogenesis, Pathways

in cancer, TNF signaling pathway, Cellular senescence, and mitogen-activated protein
kinase (MAPK) signaling pathway (Fig. S2a). However, the DEG in both Lp082 and
DSS groups, SASP and DSS groups, and SASP and Lp082 groups were mainly
enriched in the following five pathways: Complement and coagulation cascades,
Platelet activation, Autophagy - animal, Phagosome and N-Glycan biosynthesis (Fig.
S2b-S2d). Besides, the DEGs in Lp082 and DSS groups, as well as SASP and DSS
groups were involved in protein processing in the endoplasmic reticulum and
metabolic pathways (Fig. S2b-S2c).

The results of gut mucosal barrier analysis showed that gene expression of
MUC-2, ZO-1, ZO-2, occludin was significantly reduced in the DSS group but
significantly increased in the Lp082 and SASP groups (p values < 0.05), and the gene
expression of intercellular cell adhesion molecule-1 (ICAM-1), vascular cell adhesion
molecule (VCAM,) claudin-1, and claudin-2 increased significantly in the DSS group
but decreased significantly in the Lp082 and SASP groups (p values < 0.05)
(Fig.5g-5j). It is worth mentioning that MUC-2 is an essential component of gut
mucosa; ICAM-1 and VCAM induce gut mucosal lesions; ZO-1, ZO-2, and occludin
promote tight junctions of gut epithelial cells; claudin-1 and claudin-2 increase
intestinal permeability and aggravate inflammation.

Results of gene analysis related to NF- κ B pathway showed that Lp082 also
inhibited the mRNA expression of NF- κ B1, NF- κ B2, cyclooxygenase-2 (COX-2),
inducible nitric oxide synthase (iNOS), Toll-4, and RelA. These genes are signaling
molecules in the NF- κ B signaling pathway (Fig.5g-5j).

Reference

- 1. Y. Liao, G. K. Smyth and W. Shi, featureCounts: an efficient general purpose
program for assigning sequence reads to genomic features, *Bioinformatics*, 2014,
30, 923-930.
- 2. G. E. Hume, E. V. Fowler, D. Lincoln, R. Eri, D. Templeton, T. H. Florin, J. A.
Cavanaugh and G. L. Radford-Smith, Angiotensinogen and transforming growth
factor beta1: novel genes in the pathogenesis of Crohn's disease, *Journal of*

*medical genetics*, 2006, 43, e51.

- 3. Liao Y, Smyth GK, Shi W. featureCounts: an efficient general purpose program
for assigning sequence reads to genomic features. *Bioinformatics*.
2014;30(7):923-30; doi: 10.1093/bioinformatics/btt656.
- 4. Maza E. In Papyro Comparison of TMM (edgeR), RLE (DESeq2), and MRN
Normalization Methods for a Simple Two-Conditions-Without-Replicates
RNA-Seq Experimental Design. *Frontiers in Genetics*. 2016;7; doi:
10.3389/fgene.2016.00164.
- 5. Wang Y, li J, Ma C, Jiang S, Li C, Zhang L, et al. Lactiplantibacillus plantarum
HNU082 inhibited the growth of *Fusobacterium nucleatum* and alleviated the
inflammatory response introduced by *F. nucleatum* invasion. *Food & Function*.
2021;12(21):10728-40; doi: 10.1039/d1fo01388b.

5. In discussion, the creativity of manuscript should be noted compared with the
similarity studies which published before.

**Response:** We appreciate your valuable and helpful comment. We are very sorry for
our negligence of the creativity of manuscript. We sincerely apologize for the
confusion caused to you. According to your helpful suggestions, We have rewritten
the relevant content of the discussion section. The rewritten content focuses more on
creativity and innovation compared with similar studies published in the past. The
changes have been highlighted in the manuscript in yellow. And the revised content is
as follows.

Taniguchi et al. [1] found that ICAM-1 increases colonic mucosal damage. In our
study, we found that the Lp082 can not only decreased the mRNA expressions of
ICAM-1 and VCAM-1 but also can be effective in relieving intestinal mucosal lesions
(i.e., reduced ulceration and inflammatory cell infiltration caused by DSS). While the
adhesion molecules ICAM-1 and VCAM-1 are the key to the induction of intestinal
mucosal lesions[2]. This suggests that Lp082 may reduce intestinal mucosal lesions
by reducing mRNA expression of ICAM-1 and VCAM, thereby alleviating neutrophil
infiltration and ulceration. The above results showed that probiotic Lp082 increased

the MUC-2 content in the mucus layer by restoring the number of goblet cells, and
relieved the intestinal mucosal damage caused by ICAM-1 and VCAM-1, so as to
repaired the chemical barrier. (Page 17, line: 470-480)

Cordeiro et al. [6] found that the content of ZO-1 and ZO-2 were significantly
decreased in UC mice, but were increased after probiotic minas frescal cheese intake.
Because Lp082 excellently improved histopathology, we speculated that Lp082 also
has a regulatory effect on TJ molecules. To this end, we analyzed major TJ proteins,
including ZO-1, ZO-2, and occludin. As expected, the mRNA expression and
immunofluorescence protein content of ZO-1, the mRNA expression of ZO-2 and
occludin were significantly decreased in DSS-induced UC mice, but were
significantly improved in the Lp082 group, indicating that the improvement of the
mechanical barrier by regulating TJ may be one of the mechanisms by which
probiotic Lp082 exerts anti-UC. In addition, Icam-1 and VCAM-1, which are
abnormally expressed in UC patients, were increased in DSS group [7]. Adhesion
molecules ICAM-1 and VCAM-1 can not only induce intestinal mucosal injury [8],
but also increase the permeability of intestinal mucosa [1] while anti-ICAM-1
treatment can alleviate colonic mucosal injury [9]. Interestingly, the mRNA
expression of ICAM-1 and VCAM-1 was found to decrease after Lp082 ingestion.
Therefore, it can be thought that the alleviation of UC by Lp082 may be due to
down-regulation of ICAM-1, VCAM-1 and increase protein quantity and mRNA
expression of ZO-1, ZO-2 to reduce intestinal mucosal permeability, thereby
inhibiting the entry of harmful bacteria and undigested food and toxins into the body
and reducing inflammation. These results suggest that Lp082 repairs the intestinal
mechanical barrier by regulating TJ. (Page 17, line: 491-511)

Although the exact etiology of UC is complex and uncertain, studies suggest that the
NF- κ B pathway plays a vital role in the pathogenesis of UC [10]. Our study has
proved that Lp082 inhibits the NF- κ B pathway by down-regulating the mRNA
expression of NF- κ B2, NF- κ B1, COX-2, Rela, Toll4, iNOS, and that NF- κ B can also

regulate inflammation by regulating cytokines [11]. Therefore, it can be suggested
that Lp082 also has a specific regulatory effect on cytokines. To confirm this, we
analyzed the cytokines associated with NF- κ B. As expected, we observed that the
mRNA expression level of pro-inflammatory cytokines (TNF- α , IL-1 β , and IL-6) was
significantly increased in the DSS group but significantly decreased in the Lp082
group. It is interesting to note that the protein levels of TNF- α , IL-1 β , and IL-6
detected by ELISA kit were also increased in the DSS group and decreased after
Lp082 intake. Among them, TNF- α can promote the proliferation and differentiation
of T cells and increase intestinal inflammation [12]. The upregulation of IL-1 β is
involved in the recruitment and retention of leukocytes in inflamed tissues and can
activate innate immune lymphocytes [13]. IL-6 activates NF- κ B to regulate the
dextran sulfate sodium-induced colitis in mice [14]. The above results indicate that
Lp082 alleviates UC by inhibiting the levels of pro-inflammatory factors (TNF- α ,
IL-1 β , and IL-6). Interestingly, we also found that the mRNA expressions of
anti-inflammatory cytokines IL10, TGF-1, and TGF-2 were significantly decreased in
the DSS group but increased in the Lp082 group. IL-10 protein levels measured by
ELISA kit also decreased in the DSS group and increased in the Lp082 group.
Surprisingly, IL10, TGF-1, and TGF-2 were shown to activate Treg and
anti-inflammatory macrophages to alleviate UC [15]. And Sato et al. [16] also found
that the loss of IL-10 spontaneously gave rise to IBD, and Hume et al. [17] found that
TGF- β 1 and TGF- β 2 could dramatically relieve intestinal inflammation in
DSS-induced colitis mice. These results suggest that Lp082 alleviates UC by
increasing the levels of anti-inflammatory factors IL10, TGF-1, and TGF-2. We
further analyzed the specific regulatory effects of Lp082 on intestinal mucosal
immunity. In addition to inflammatory factors, we also noticed that a heme protein,
MPO, was significantly reduced in the Lp082 group. Trevisin et al. [18] found that
MPO caused UC by producing cytokines and hypochlorite and that MPO in the colon
of UC patients is mainly produced by neutrophil infiltration [19]. Interestingly, this is
consistent with the fact that the DSS group had a severe neutrophil infiltration in this
study. However, neutrophil infiltration and MPO content were significantly decreased

in the Lp082 group. This shows that Lp082 alleviates UC by reducing neutrophil
infiltration and its secreted MPO content. Thus, our results suggest that Lp082 may
play an anti-UC effect by inhibiting the NF- κ B pathway, down-regulating
pro-inflammatory cytokines, and up-regulating anti-inflammatory cytokines, reducing
MPO content, thereby maintaining immune balance and protecting the immune barrier.
(Page 18, line: 514-553)

**Reference**

- 1. Taniguchi T, Tsukada H, Nakamura H, Kodama M, Fukuda K, Saito T, et al. Effects
of the anti-ICAM-1 monoclonal antibody on dextran sodium sulphate-induced
colitis in rats. *Journal of gastroenterology and hepatology*. 1998;13(9):945-9; doi:
10.1111/j.1440-1746.1998.tb00766.x.
- 2. Philpott JR, Miner PB, Jr. Antisense inhibition of ICAM-1 expression as therapy
provides insight into basic inflammatory pathways through early experiences in
IBD. *Expert Opin Biol Ther*. 2008;8(10):1627-32; doi:
10.1517/14712598.8.10.1627.
- 3. Shi JL, Xie QG, Yue YX, Chen QX, Zhao LN, Evivie SE, et al. Gut microbiota
modulation and anti-inflammatory properties of mixed lactobacilli in dextran
sodium sulfate-induced colitis in mice. *Food & Function*. 2021;12(11):5130-43; doi:
10.1039/d1fo00317h.
- 4. Edelblum KL, Turner JR. The tight junction in inflammatory disease:
communication breakdown. *Current Opinion in Pharmacology*. 2009;9(6):715-20;
doi: 10.1016/j.coph.2009.06.022.
- 5. Stio M, Retico L, Annese V, Bonanomi AG. Vitamin D regulates the tight-junction
protein expression in active ulcerative colitis. *Scandinavian Journal of*
*Gastroenterology*. 2016;51(10):1193-9; doi: 10.1080/00365521.2016.1185463.
- 6. Cordeiro BF, Alves JL, Belo GA, Oliveira ER, Braga MP, da Silva SH, et al.
Therapeutic Effects of Probiotic Minas Frescal Cheese on the Attenuation of
Ulcerative Colitis in a Murine Model. *Frontiers in Microbiology*. 2021;12; doi:
10.3389/fmicb.2021.623920.

- 7. Nakamura S, Ohtani H, Watanabe Y, Fukushima K, Matsumoto T, Kitano A, et al.
In situ expression of the cell adhesion molecules in inflammatory bowel disease.
Evidence of immunologic activation of vascular endothelial cells. *Laboratory*
*investigation; a journal of technical methods and pathology*. 1993;69(1):77-85.
- 8. Mitselou A, Grammeniatis V, Varouksi A, Papadatos SS, Klaroudas A, Katsanos K,
et al. Immunohistochemical Study of Adhesion Molecules in Irritable Bowel
Syndrome: A Comparison to Inflammatory Bowel Diseases. *Advanced biomedical*
*research*. 2021;10:21; doi: 10.4103/abr.abr_2_20.
- 9. Ruco LP, de Laat PA, Matteucci C, Bernasconi S, Sciacca FM, van der Kwast TH,
et al. Expression of ICAM-1 and VCAM-1 in human malignant mesothelioma. *The*
*Journal of pathology*. 1996;179(3):266-71.
- 10. Hu LH, Liu JY, Yin JB. Eriodictyol attenuates TNBS-induced ulcerative colitis
through repressing TLR4/NF-kB signaling pathway in rats. *Kaohsiung Journal of*
*Medical Sciences*. 2021;37(9):812-8; doi: 10.1002/kjm2.12400.
- 11. Bauer J, Namineni S, Reisinger F, Zoller J, Yuan DT, Heikenwalder M.
Lymphotoxin, NF-kappa B, and Cancer: The Dark Side of Cytokines. *Digestive*
*Diseases*. 2012;30(5):453-68; doi: 10.1159/000341690.
- 12. Xiao B, Laroui H, Ayyadurai S, Viennois E, Charania MA, Zhang YC, et al.
Mannosylated bioreducible nanoparticle-mediated macrophage-specific TNF-alpha
RNA interference for IBD therapy. *Biomaterials*. 2013;34(30):7471-82; doi:
10.1016/j.biomaterials.2013.06.008.
- 13. 김연하, 김유림, 김성중, 황호근, Choi S-C, 김경숙, et al. Rebamipide Protects
Colonic Damage Induced by Trinitrobenzene Sulfonic Acid (TNBS) via
Down-Regulation of TNF- α , IL-1 β , and ICAM-1. *Anatomy and Cell Biology*.
2004;37(2):149-56.
- 14. Zhou X, Liu H, Zhang J, Mu J, Zalan Z, Hegyi F, et al. Protective effect of
*Lactobacillus fermentum* CQPC04 on dextran sulfate sodium-induced colitis in
mice is associated with modulation of the nuclear factor-kappa B signaling pathway.
*Journal of Dairy Science*. 2019;102(11):9570-85; doi: 10.3168/jds.2019-16840.
- 15. Mohammadnia-Afrouzi M, Hosseini AZ, Khalili A, Abediankenari S, Amari A,

Aghili B, et al. Altered microRNA Expression and Immunosuppressive Cytokine
Production by Regulatory T Cells of Ulcerative Colitis Patients. Immunological
Investigations. 2016;45(1):63-74; doi: 10.3109/08820139.2015.1103749.

16. Sato Y, Takahashi S, Kinouchi Y, Shiraki M, Endo K, Matsumura Y, et al. IL-10
deficiency leads to somatic mutations in a model of IBD. Carcinogenesis.
2006;27(5):1068-73.

17. Hume GE, Fowler EV, Lincoln D, Eri R, Templeton D, Florin TH, et al.
Angiotensinogen and transforming growth factor beta1: novel genes in the
pathogenesis of Crohn's disease. Journal of medical genetics. 2006;43(10):e51; doi:
10.1136/jmg.2005.040477.

18. Trevisin M, Pollock W, Dimech W, Savige J. Evaluation of a multiplex flow
cytometric immunoassay to detect PR3- and MPO-ANCA in active and treated
vasculitis, and in inflammatory bowel disease (IBD). Journal of immunological
methods. 2008;336(2):104-12; doi: 10.1016/j.jim.2008.03.012.

19. Chin AC, Parkos CA. Neutrophil transepithelial migration and epithelial barrier
function in IBD: potential targets for inhibiting neutrophil trafficking. Annals of the
New York Academy of Sciences. 2006;1072:276-87; doi: 10.1196/annals.1326.018.

Once again, we thank you for the time you put in reviewing our paper and we are very
grateful to your effort reviewing our paper and your positive feedback. The summary
of our work as written by you is precise. Since your inputs have been precious, in the
eventuality of a publication, we would like to acknowledge your contribution
explicitly.

October 14, 2022

Prof. Jiachao Zhang
Hainan University
Food Science
58 renmin road
Haikou, Hainan 570228
China

Re: Spectrum01651-22R2 (Probiotics (*Lactobacillus plantarum* HNU082) supplementation relieves ulcerative colitis by affecting intestinal barrier functions, immunity-related genes expression, gut microbiota, and metabolic pathways in mice)

Dear Prof. Jiachao Zhang:

Your manuscript has been accepted, and I am forwarding it to the ASM Journals Department for publication. You will be notified when your proofs are ready to be viewed.

Sincerely,

Xiaoyu Tang
Editor, Microbiology Spectrum
